# Broadband visual stimuli improve neuronal representation and sensory perception

Elisabeta Balla [1,2,3], Gerion Nabbefeld [1,3], Christopher Wiesbrock[1,3], Jenice Linde[1,3], Severin Graff[1,3,4], Simon Musall [1,3,4,5,6] ✉ & Björn M. Kampa [1,2,3,6] ✉

Natural scenes consist of complex feature distributions that shape neural responses and perception. However, in contrast to single features like stimulus orientations, the impact of broadband feature distributions remains unclear. We, therefore, presented visual stimuli with parametrically-controlled bandwidths of orientations and spatial frequencies to awake mice while recording neural activity in their primary visual cortex (V1). Increasing orientation but not spatial frequency bandwidth strongly increased the number and response amplitude of V1 neurons. This effect was not explained by single-cell orientation tuning but rather a broadband-specific relief from center-surround suppression. Moreover, neurons in deeper V1 and the superior colliculus responded much stronger to broadband stimuli, especially when mixing orientations and spatial frequencies. Lastly, broadband stimuli increased the separability of neural responses and improved the performance of mice in a visual discrimination task. Our results show that surround modulation increases neural responses to complex natural feature distributions to enhance sensory perception.

Receptive fields of neurons in the primary visual cortex (V1) are sparse representations of the visual scenery and selectively respond to prominent features, such as the orientation of elongated edges. Yet, edge orientations in natural scenes are conditionally dependent on each other and appear as specific combinations of common orientations, especially horizontal and vertical edges[1,2]. The visual system has, therefore, likely evolved to be well-tuned to the statistics of these features in the natural environment and shows highly variable and non-linear responses to complex visual stimuli[2–4]. Natural scenes have been shown to improve stimulus-specific V1 population responses[5] and have also been used to improve models of visual motion detectors[6,7]. In fact, even individual V1 neurons respond with high specificity to different natural images and exhibit complex tuning properties that go far beyond what could be inferred by their

responses to simple sine-wave gratings or Gabor patches[8,9]. The conventional approach of studying the neural encoding of visual information by using a limited set of simple visual stimuli is, therefore, inadequate to identify the key properties of neural responses in the visual system because it only covers a very small range of the possible visual feature distributions. An alternative approach is to use natural scenes or images to identify neural response features to complex stimuli that match the statistics of the environment[10–12]. However, since natural images are not parametrically well-defined[13,14], very large sets of images and complex models are required to identify meaningful neural response features. Moreover, the presentation of large sets of images is often prohibited by the limited number of neural responses that can be obtained for each stimulus[8,9,15].

[1]Systems Neurophysiology, Department of Neurobiology, RWTH Aachen University, Aachen, Germany. [2]JARA BRAIN Institute of Neuroscience and Medicine (INM-10), Forschungszentrum Jülich, Jülich, Germany. [3]Research Training Group 2416 MultiSenses – MultiScales, RWTH Aachen University, Aachen, Germany. [4]Institute of Biological Information Processing, Department for Bioelectronics, Forschungszentrum Jülich, Jülich, Germany. [5]Faculty of Medicine, Institute of Experimental Epileptology and Cognition Research, University of Bonn, Bonn, Germany. [6]These authors jointly supervised this work: Simon Musall, Björn M. Kampa. ✉e-mail: musall@bio2.rwth-aachen.de; kampa@brain.rwth-aachen.de

Different approaches have been employed to study the neuronal encoding of complex visual scenes. Quantification of natural images by using a bank of visual filters allows an analytical description based on different visual features, such as orientation, spatial frequencies, and their higher-order correlations[16]. These filters mimic the tuning properties and receptive fields of visual cortical neurons so that their outputs provide a simple model for cortical responses to natural images. The outputs from first and higher-order filters can then be used to synthesize new images with similar features to the original image and experimentally test neural responses and sensory perception. Experiments in humans[17], monkeys[18,19], and mice[20] have shown that these synthesized naturalistic textures can be perceived as similar to their natural counterpart, in particular when specific higher-order correlations of visual features are preserved. While this approach can break down natural images into a set of higher-order parameters, the entire parameter space is too large to be tested systematically and has mostly been linked to visual processing in higher visual areas (HVAs)[18–20]. How higher-order parameters affect neural responses to low-level features in V1, therefore, remains elusive[21]. In addition, the modulation of neural responses by stimuli extending to the area surrounding the receptive field of a visual neuron[22,23] is usually not considered in these algorithms.

An alternative approach focuses on identifying higher-order patterns in adjacent image pixels[24–26]. Studies in humans, monkeys, and rats have demonstrated perceptual sensitivity for different statistical orders of these pixel patterns with the psychophysics matching natural scene statistics[17,26,27]. Correspondingly, neurons in the primary and higher visual cortex are more sensitive to cardinal orientations, which are also overrepresented in natural scenes[2,28,29]. However, local pixel patterns only represent a limited aspect of natural scenes, and it remains unknown how responses to these patterns relate to more global distributions of orientations or spatial frequencies. This is particularly important in the context of surround modulation, where local stimuli within the receptive field of cortical neurons are strongly modulated by the surrounding context, both for simple grating stimuli but also for more complex natural scenes[30–37]. Correspondingly, spatial correlations have also been shown to play an important role in the perception of natural scenes[18–20].

Several studies also used visual noise stimuli to characterize neural responses to richer combinations of low-level stimulus features[38–40]. Interestingly, such broadband stimuli that contain a mixture of stimulus features increase neural responses, most likely due to the recruitment of additional neurons with diverse tuning preferences for stimulus orientations and spatial frequencies[29,38,39,41]. This suggests that broadening the bandwidth of any visual feature with specific tuning in cortical neurons, such as orientation or spatial frequency, might be beneficial to elicit stronger responses from a diverse population of visual neurons. Earlier modeling work also suggested that such a diverse population of responding neurons might provide a more accurate encoding of natural images and potentially improve perceptual acuity[39]. However, it remains unclear which low-level features are most effective in increasing cortical network activity and if the tuning of individual neurons or other factors, such as surround modulation, are the most important contributors. Moreover, the extent to which heightened neural responses to broadband stimuli can truly enhance visual perception remains an open question.

To assess the impact of broadband stimuli on neural responses and sensory perception, we therefore performed neural recordings and psychophysical experiments in mice. We used random phase textures, so-called motion clouds[42], to create broadband visual stimuli of different orientations or spatial frequency bandwidths. Motion clouds allow the generation of parametrically well-defined stimulus sets with specific feature distributions that can approximate the distributions of natural images. We then used two-photon imaging in awake mice to measure the responses of V1 in layer 2/3, exhibiting diverse visual tuning properties to visual gratings[43–45]. Increasing the orientation bandwidth activated more V1 neurons and also increased their neural response amplitude. In contrast, increasing the spatial frequency bandwidth modulated the activity of individual neurons but did not increase the overall neural population response. A Gabor filter model showed that this was not simply explained by the orientation tuning of individual neurons but also by center-surround suppression. Follow-up experiments confirmed this prediction, showing a clear relation between lowered surround suppression and increased broadband responses across individual neurons. Using high-density electrophysiology, we also recorded from deeper cortical layers and the superior colliculus (SC) and found that neurons were strongly driven by mixed stimuli with large orientation and spatial frequency bandwidth. Lastly, we trained mice to discriminate either the orientation or spatial frequency of motion cloud stimuli and found that expanding the orientation, but not spatial frequency, bandwidth improved perceptual performance. Our results demonstrate that surround modulation is an important driver of increased neural responses to complex visual stimuli, suggesting that cortical processing and visual perception are optimized for visual stimuli that match the statistics of natural sensory inputs.

## Results
### Broad orientation bandwidth increases the recruitment and response amplitude of V1 neurons

Natural scenes contain edges with a broad range of orientations that can strongly vary in different spatial locations (Fig. 1a). To test how neurons in the visual cortex respond to different ranges of orientation distributions (orientation bandwidth), we designed different sets of motion cloud stimuli to approximate a sensible range of orientation bandwidths that are also found in natural images[2,46,47] (Fig. 1b). Broadband stimuli had a bandwidth of 45° to match the dominant orientation bandwidths of natural images (Fig. 1a)[2] while mid-range stimuli (25°) were chosen to be between broad- and narrow-range stimuli with a 5° bandwidth. All motion clouds had a central orientation of 0° and a central spatial frequency of 0.04 cpd to match the response tuning of neurons in mouse V1 and HVAs[29,48,49]. We then presented 5-s long, rightward drifting motion cloud stimuli (1 Hz) in a pseudorandom order (with 5-s mean-gray inter-stimulus interval) to awake, head-fixed mice that were passively sitting in a tube.

Using 2-photon imaging, we then measured the neural responses of layer 2/3 neurons in V1, ubiquitously expressing the calcium indicator GCaMP6f (Fig. 1c, d). To evaluate the impact of different orientation bandwidths on the number of responding neurons, we first used a one-sided Mann-Whitney U test ($p < 0.05$) to identify all neurons with a significant response for each stimulus condition against baseline. Where possible, we used a linear mixed effects (LME) model, which accounts for nested data from multiple animals, to test for significant differences across conditions. Then, we followed with a Bonferroni correction for the number of performed tests (see "Methods").

Further, expanding the orientation bandwidth induced a more than two-fold increase in the number of visually responsive neurons across all sessions (Fig. 1e), demonstrating that broader orientation bandwidth stimuli recruit a much larger population of neurons (normalized responding cell count, both tested against the narrow stimulus: mid range (25°) = $1.38 \pm 0.09$, LME model test against narrow, $T = 4.64$, $p = 4.9 \times 10^{-5}$; broad range (45°) = $2.2 \pm 0.23$, LME model test against narrow, $T = 5.58$, $p = 3.04 \times 10^{-6}$; mean ± s.e.m, $n = 18$ sessions from 9 mice. See also per mouse comparison in Supplementary Fig. S1a. Percentage of visually responsive neurons over total neurons per session: narrow range = 11% ± 1%; mid-range = 15% ± 2%; broad range = 21% ± 2%). In addition to increasing the total number of responding neurons, expanding the orientation bandwidth also strongly increased neural responses, which were the largest for the broad orientation bandwidth (Fig. 1f, $\Delta F/F_{narrow} = 1.16\% \pm 0.04\%$;

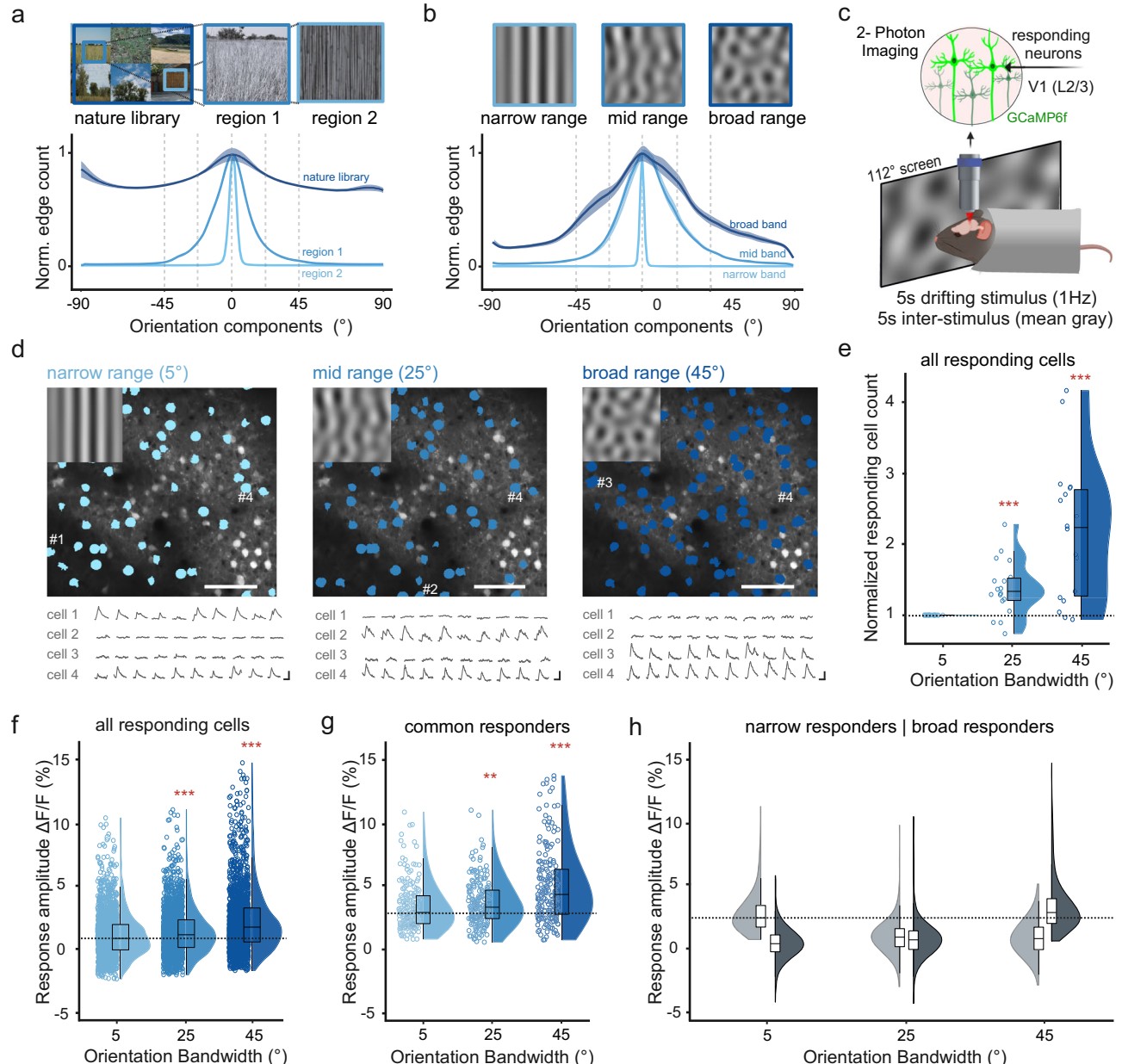

**Fig. 1 | V1 neurons are most responsive to broad orientation bandwidths.**
**a** Mean orientation distribution from natural images (example images adapted from Tkacik G et al., "Natural images from the birthplace of the human eye ", PLoS ONE 6: e20409 (2011), $n = 1077$ images, shading shows s.e.m.). Single-cell selectivity was tested at specific orientations (dashed lines, Fig. 3, Supplementary Fig. S2). Regions 1, and 2 show mid-range or narrow orientation bandwidth examples.
**b** Same as (**a**) for broadband orientation bandwidth stimuli (5°, 25°, and 45°, $n = 300$ frames each). **c** Awake, head-fixed mice viewed differing orientation bandwidth stimuli while 2Photon-imaging V1 neural activity in layer 2/3. **d** Example session, showing V1 neural responses to different orientation bandwidths (5°, 25°, 45°, blue masks show responsive cells, scale bar: 100 µm, corresponding stimuli at top left). Traces show single-trial responses of four example cells with different orientation bandwidth preferences (scale bar: 1 s, 20% ΔF/F). **e** Counts of responsive neurons to each orientation bandwidth, normalized by the number of cells responding to the 5° bandwidth ($n = 18$ sessions from 9 mice). **f** Difference between mean response

amplitude versus baseline for each orientation bandwidth. Shown are neurons with a significant response for at least one condition (1394 out of 4892 neurons, 29% ± 2% neurons per session; mean ± s.e.m.). The horizontal line shows the median narrow response amplitude. **g** Same as (**f**) for neurons that consistently responded to all bandwidths (12% of responding cells per session, 252 total neurons). The horizontal line shows the median narrow response amplitude. **h** Same as (**f**) for neurons that preferentially responded to narrow-band (left violin plots, 7% of responding cells per session, 152 in total), or to broadband (right violin plots, 23% of responding cells per session, 487 neurons in total). Box plots indicate the median (horizontal line), interquartile range (box bounds: 25th–75th percentiles), and whiskers (1.5 × interquartile range). The dotted line is the narrow-band median. Stars mark significant (Bonferroni correction for two tests, $\alpha = 0.025$) differences from two-sided tests against narrow condition: LME test (panel **e**) Wilcoxon signed rank (panels **f**, **g**) Wilcoxon ranked sum (panel **h**). Panel (**c**) was created in BioRender. Balla, E. https://BioRender.com/g05l789 (2025).

ΔF/F$_{mid}$ = 1.44% ± 0.04%, Wilcoxon signed rank test against narrow, $p = 5.21 \times 10^{-12}$; ΔF/F$_{broad}$ = 2.29% ± 0.06%, Wilcoxon signed rank test against narrow, $p = 5.25 \times 10^{-66}$, mean ± s.e.m., $n = 1394$ neurons from 18 sessions, 9 mice, see also per mouse comparison in Supplementary Fig. S1b).

Expanding the orientation bandwidth, therefore, increased both the number and amplitude of stimulus-responsive V1 neurons. Interestingly, this increase in response amplitude was also seen in neurons that consistently responded to all orientation bandwidths (Fig. 1g). These common responders significantly

responded to all bandwidths (one-sided Mann-Whitney U test, $p < 0.05$) but also strongly increased their response amplitude to broadband orientation stimuli (Fig. 1g, $\Delta F/F_{narrow} = 3.39\% \pm 0.14\%$; $\Delta F/F_{mid} = 3.67\% \pm 0.13\%$, Wilcoxon signed rank test against narrow, $p = 2.6 \times 10^{-3}$; $\Delta F/F_{broad} = 5.05\% \pm 0.22\%$, Wilcoxon signed rank test against narrow, $p = 2.1 \times 10^{-9}$, mean $\pm$ s.e.m., $n = 252$ neurons from 18 sessions, 9 mice). This indicates that the increase in neural response amplitude was not solely explained by the recruitment of additional neurons but rather by a general increase in neural responsiveness.

To further assess if increased neural responsiveness was due to the recruitment of broadband-selective neurons, we calculated a bandwidth selectivity index (calculated as the response difference between a preferred orientation bandwidth and the mean of the response to the other two bands, divided by their sum) and identified bandwidth-selective cells using a shuffle control (see "Methods"). Most bandwidth-selective neurons preferred broadband stimuli, while a smaller subpopulation preferred narrowband stimuli (Supplementary Fig. S1d). This could indicate that additional neurons were recruited due to their orientation tuning being further away from the narrow 5° range, while narrowband-selective neurons with close to 5° tuning would become unresponsive. However, surprisingly few neurons were significantly bandwidth-selective ($n = 25, 7, 76$ neurons for the narrow, mid, and broad range, respectively, versus 252 common responders), suggesting that increased responsiveness to broadband stimuli was not just driven by broadband-selective neurons.

Lastly, we compared the response amplitude of narrow- versus broadband-selective neurons to test if higher broadband response magnitude was driven by stronger responses of broadband-selective neurons or rather a general increase in responsiveness, as in the common responders (Fig. 1g). Here, we used a less conservative approach to identify bandwidth-selective neurons by selecting all cells that only significantly responded to either the narrow- or broadband condition (one-sided Mann-Whitney U test, $p < 0.05$). As expected, each group had their strongest responses to either narrow- or broadband stimuli but the response amplitudes in their respective preferred stimulus condition were almost similar (Fig. 1h, maximum response mean $\pm$ s.e.m.: $\Delta F/F_{narrow}$ 3.36% $\pm$ 0.28%, $n = 152$ neurons, $\Delta F/F_{broad} = 3.60\% \pm 0.17\%$, $n = 487$ neurons; Wilcoxon ranked sum test for narrow versus broad, $p = 0.089$).

Together, these results show that broadband orientation stimuli strongly increase the number and amplitude of neural responses. However, this effect appears to be due to a general increase in neural responses instead of the recruitment of a broadband-selective sub-population with a stronger visual response magnitude.

## Expanding spatial frequency bandwidth does not increase V1 responses

Similar to stimulus orientations, natural scenes also contain a large range of spatial frequencies. Spatial frequency distributions in natural scenes follow a power law, with higher power in the low-frequency bands for broad image features and lower power in the high-frequency bands for fine structural details[28,46] (Fig. 2a). V1 neurons have also been characterized for their spatial frequency tuning[2,29,49], so we tested if expanding the spatial frequency bandwidth of visual stimuli would affect neural responses similarly to orientation bandwidths. We, therefore, created motion cloud stimuli with different spatial frequency bandwidths (0.004 cpd, 0.04 cpd, and 0.4 cpd) with a vertical orientation, again approximating the distributions in different natural scenes (Fig. 2a, b). Visual stimuli were presented pseudo-randomly while measuring the activity of V1 neurons as described above (Fig. 2c, d).

In contrast to orientation bandwidth, expanding the spatial frequency bandwidth did not increase the number of visually responsive neurons across sessions (Fig. 2d, e; normalized responding cell count both tested against narrow condition: mid-range (0.04 cpd) = $1.03 \pm 0.1$,

LME model test against narrow: $T = 0.35$, $p = 0.72$; broad range (0.4 cpd) = $1.3 \pm 0.24$, LME model test against narrow, $T = 1.43$, $p = 0.16$; $n = 16$ recordings from 9 mice. See also per mouse comparison in Supplementary Fig. S1e. Percentage of responsive neurons over total neurons per session: narrow range = 7% $\pm$ 1%, mid-range = 7% $\pm$ 2%; broad range = 9% $\pm$ 1%, mean $\pm$ s.e.m.). Correspondingly, we found no significant increase in the response amplitude to the broad versus narrow spatial frequency bandwidth stimuli for all responsive neurons (Fig. 2f; $\Delta F/F_{narrow} = 1.7\% \pm 0.1\%$; $\Delta F/F_{mid} = 1.9\% \pm 0.1\%$; Wilcoxon signed rank test against narrow: $p = 0.09$; $\Delta F/F_{broad} = 1.9\% \pm 0.1\%$; Wilcoxon signed rank test against narrow, $p = 0.25$, mean $\pm$ s.e.m, $n = 888$ neurons from 16 sessions, 9 mice, see also per mouse comparison in Supplementary Fig. S1e). There were also no differences in response amplitude across stimulus conditions for common responders (Fig. 2g, $\Delta F/F_{narrow} = 4.9\% \pm 0.4\%$; $\Delta F/F_{mid} = 6\% \pm 0.5\%$, Wilcoxon signed rank test against narrow: $p = 0.11$; $\Delta F/F_{broad} = 4.6\% \pm 0.4\%$, Wilcoxon signed rank test against narrow: $p = 0.71$, mean $\pm$ s.e.m., $n = 68$ neurons).

A potential reason for the lack of increased responses to broadband spatial frequency stimuli could be that V1 neurons already exhibit broad spatial frequency tuning, resulting in overlapping activation of these neurons by different spatial frequency bandwidths[29]. This should result in a larger percentage of neurons that consistently respond to all spatial frequency bandwidth conditions. However, a smaller percentage of all stimulus-responsive neurons were common responders for broadband spatial frequency versus broadband orientation stimuli (11.2% versus 18.1%, respectively, Figs. 1g and 2g). Moreover, neurons that were significantly bandwidth-selective against a shuffle control were more common for broadband spatial frequency stimuli compared to broadband orientation stimuli (11.1% versus 7.7%, respectively). Lastly, an equally large amount of these bandwidth-selective neurons preferred either narrow- or broad-range stimuli, whereas mid-range selective neurons represented a smaller subpopulation (Supplementary Fig. S1h). Broadening spatial frequency bandwidth, therefore, induced an equal amount of up and down-modulation of neural responses in differently tuned neurons, but the total number and amplitude of responsive neurons remained unchanged. In line with this finding, we also found a similar response amplitude when selecting only neurons that significantly responded to either narrow- or broadband spatial frequency stimuli (Fig. 2h, maximum response mean $\pm$ s.e.m $\Delta F/F_{narrow} = 4.32\% \pm 0.33\%$, $n = 133$ neurons, $\Delta F/F_{broad} = 4.99\% \pm 0.45\%$, $n = 132$ neurons; Wilcoxon ranked sum test for narrow versus broad: $p = 0.52$).

To further test if the tuning of individual neurons differed between orientation and spatial frequency relative to the range that we used in the motion cloud stimuli, we performed additional recordings and presented simple gratings with different combinations of orientation and spatial frequency (Supplementary Fig. S2). We tested the feature tuning to a range of orientations and spatial frequencies that were most prominent in natural images and, therefore, used to create our visual stimuli (dashed lines in Figs. 1a, 2a). While we observed diverse neural tuning to the presented gratings, the tuning width of responding neurons covered a comparable range of orientations and spatial frequencies relative to the used orientation or spatial frequency bandwidths. This further argues against the hypothesis that the lack of increased neural responses to broadband spatial frequency stimuli was because of particularly broad spatial frequency tuning that prevented further recruitment of specific subpopulations. Instead, stronger responsiveness to broader orientation bandwidths appears to be a specific feature of V1 neurons that does not generalize to spatial frequency.

## A Gabor filter model with surround suppression accurately predicts broadband responses

A parsimonious explanation for the increased neural recruitment and responsiveness to broader orientation bandwidth would be the additional recruitment of neurons with orientation tuning that is outside of

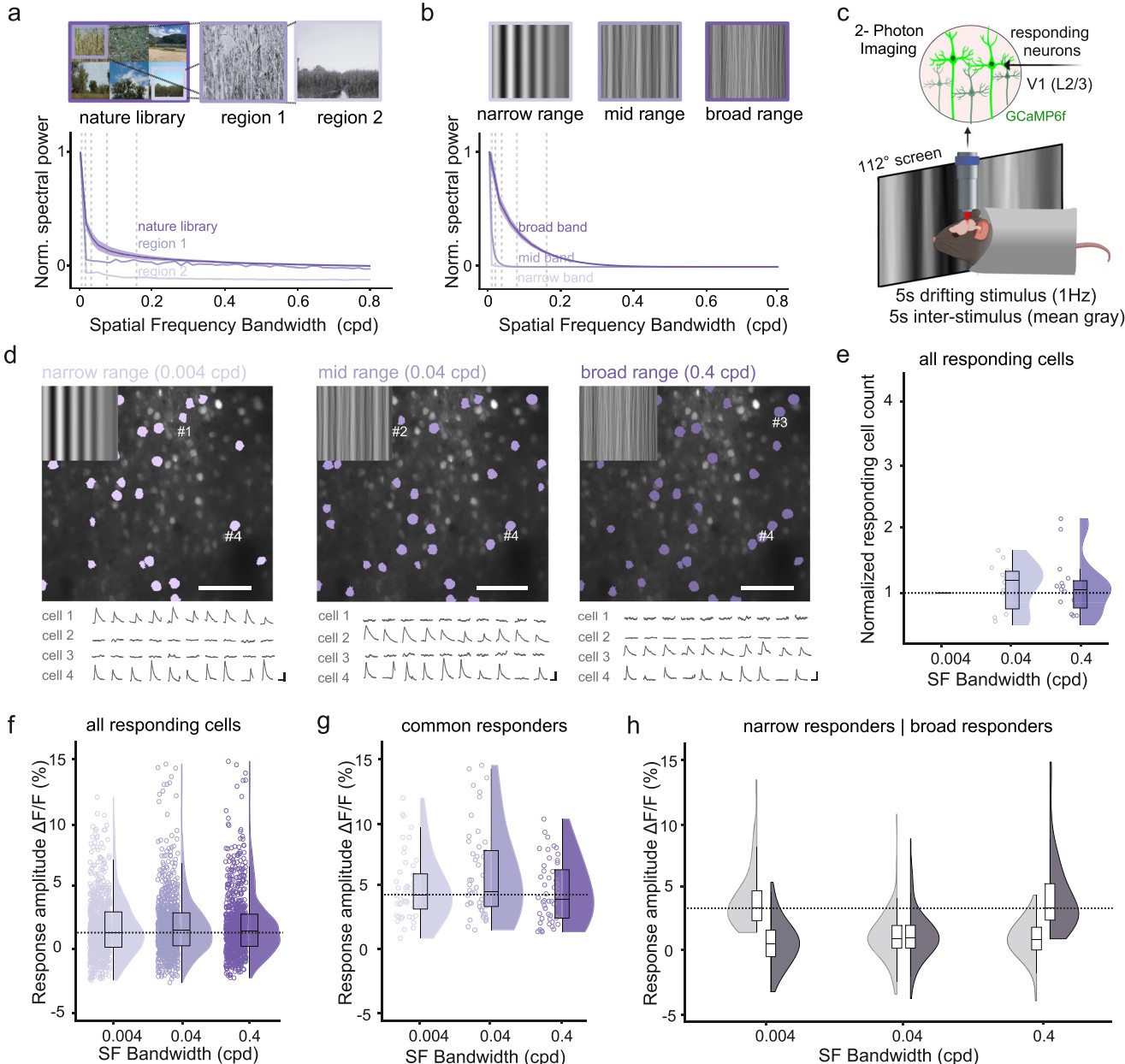

**Fig. 2 | V1 neurons are equally responsive to different spatial frequency bandwidths. a** Mean spatial frequency distribution from natural images (example images adapted from Tkacik G et al., "Natural images from the birthplace of the human eye ", PLoS ONE 6: e20409 (2011), $n = 1077$ images, shading shows s.e.m.). Single-cell selectivity was tested at specific spatial frequencies (dashed lines, Supplementary Fig. S2). **b** Same as (**a**) for broadband stimuli with different spatial frequency distributions (0.004 cpd, 0.04 cpd, and 0.4 cpd, $n = 300$ frames each). **c** Awake, head-fixed mice viewed differing orientation bandwidth stimuli while 2Photon-imaging V1 neural activity in layer 2/3. **d** Example session, showing V1 neural responses to different spatial frequency bandwidths (0.004 cpd, 0.04 cpd, 0.4 cpd, purple masks show responsive cells, scale bar: 100 μm, corresponding stimuli at top left). Traces show single-trial responses of 4 example cells with different spatial frequency bandwidth preferences (scale bar: 1 s, 20% ΔF/F). **e** Counts of responsive neurons to each spatial frequency bandwidth, normalized by the number of cells responding to 0.004 cpd bandwidth ($n = 16$ sessions). **f** Difference between mean response amplitude versus baseline for each orientation bandwidth.

Shown are neurons with a significant response for at least one condition (603 of 3185 neurons, 19% ± 2% neurons per session; mean ± s.e.m.). The horizontal line shows the median narrow response amplitude. **g** Same as (**f**) for neurons that consistently responded to all bandwidths (68 neurons, 9% of responding cells per session). The horizontal line shows the median narrow response amplitude. **h** Same as (**f**) for neurons that preferentially responded to the narrow-band stimulus (left violin plots, 24% of responding cells per session, 133 in total), or the broadband stimulus (right violin plots, 26% of responding cells per session, 132 neurons in total). Box plots indicate the median (horizontal line), interquartile range (box bounds: 25th–75th percentiles), and whiskers (1.5 × interquartile range). The dotted line is the narrow-band median. No significant (Bonferroni correction for two tests, $\alpha = 0.025$) differences from two-sided tests against the narrow condition: LME test (panel **e**) Wilcoxon signed rank (panels **f, g**) Wilcoxon ranked sum (panel **h**). Panel (**c**) was created in BioRender. Balla, E. https://BioRender.com/g05l789 (2025).

a narrow orientation bandwidth stimulus. However, our earlier findings, such as the increased response amplitudes in common responders when expanding the orientation bandwidth (Fig. 1h) and the lack of a corresponding effect for broadband spatial frequency stimuli,

appeared to be at odds with this hypothesis. To create a theoretical prediction on the relation between orientation tuning and the corresponding responses to broadband orientation stimuli, we designed a rich Gabor-filter-based model representing the receptive field

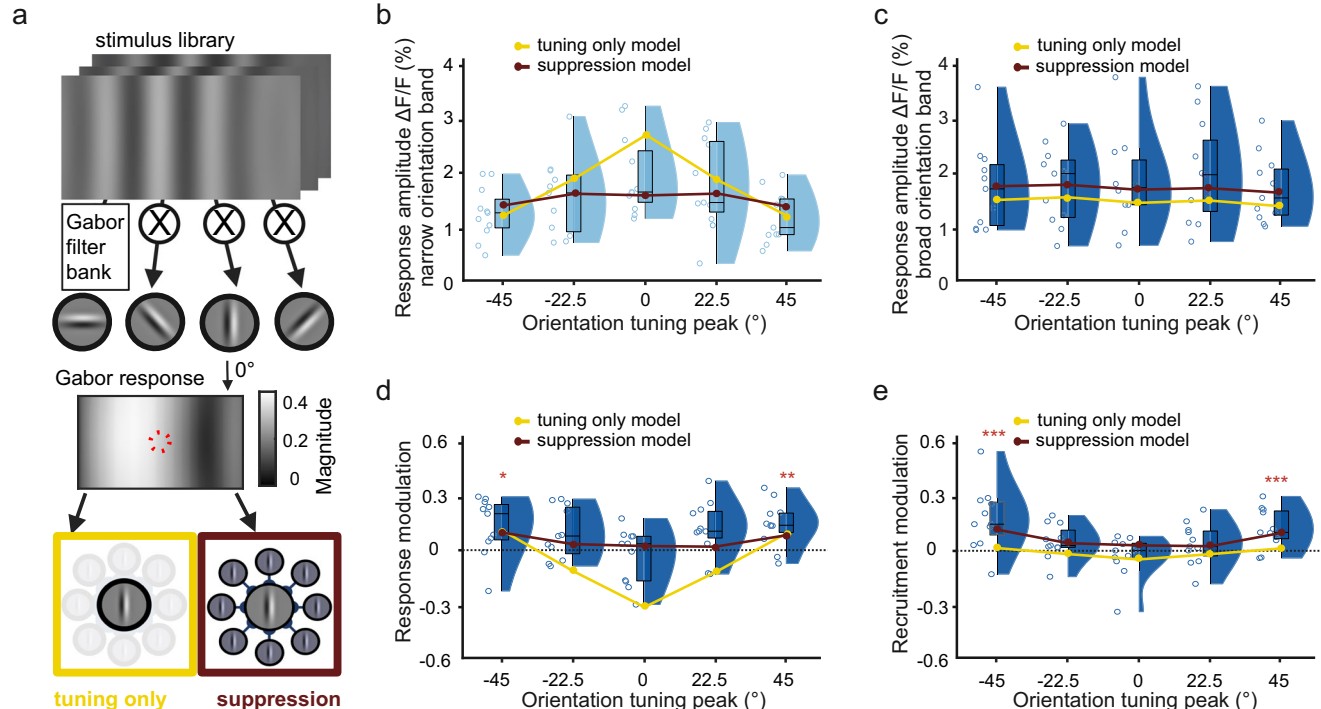

**Fig. 3 | Gabor filter model links orientation tuning and surround suppression to broadband responses. a** Schematic of a Gabor filter bank to model neural responses. Gabor filters of different orientations and spatial frequencies were used to jointly predict the measured responses to the presented broadband stimuli (see Supplementary Figs. S3 and S4 for details). Model predictions were either based on the responses of a central Gabor alone ("tuning only" model; bottom left, yellow box) or by additionally accounting for the impact of surround suppression ("suppression" model; bottom right, red box). **b** Mean response amplitude towards narrow bandwidth stimuli, depending on the orientation tuning of V1 neurons ($n = 11$ sessions in 4 mice). Neurons in each session were binned based on their orientation tuning to a set of full-contrast sinewave gratings. Yellow circles and lines show the best-fitted response of the tuning-only model. Red circles and lines show the best fit of the suppression model. (**c**) Same as (**b**) for broad orientation bandwidth stimuli. Both the tuning only and the suppression model were fit to narrow and broad orientation bandwidth stimuli at the same time. **d** Mean response modulation of orientation-tuned cells by expanding the orientation

bandwidth (derived from responses in **b** and **c**). Yellow circles and lines show the predicted response modulation from the tuning-only model. Red circles and lines show predicted response modulation from the suppression model. The horizontal black dotted line at 0 indicates no bandwidth modulation. **e** Same as panel d) but for recruitment modulation of orientation-tuned cells by expanding the orientation bandwidth. The ORI recruitment modulation index was calculated as the difference between the numbers of responsive neurons to narrow and broad orientation bandwidth stimuli divided by their sum. Yellow and red circles indicate fits of the tuning only and surround suppression model, respectively. Box plots indicate the median (horizontal line), interquartile range (box bounds: 25th–75th percentiles), and whiskers (1.5 × interquartile range). Stars in panels (**d** and **e**) mark significant differences against 0 based on an LME model (two-sided test). The significance threshold for each test was $\alpha = 0.01$ after Bonferroni correction for five individual tests. Panel a was created in BioRender. Balla, E. https://BioRender.com/g05l789 (2025).

properties of V1 neurons[38,50] (Fig. 3a and Supplementary Fig. S3 and see "Methods"). The Gabor filters covered the same range of orientations and spatial frequencies contained in our visual stimuli (Supplementary Fig. S3). In addition, we performed further recordings of responses to narrow and broad orientation bandwidth stimuli, for which we also measured the preferred orientation tuning. We then compared response amplitudes evoked by narrow and broad orientation stimuli, depending on the preferred orientation of V1 neurons (Fig. 3b, c).

As expected, responses to narrow bandwidth stimuli were related to orientation tuning, with neurons tuned to the 0° central orientation responding the strongest. In contrast, no clear tuning dependence was present for responses to the broadband orientation stimuli (Fig. 3b, c). The Gabor filter model also reproduced this general dependence of neural responses on orientation tuning (yellow lines show best-model fits). However, the simple "tuning only" model predicted a much stronger orientation dependence of response amplitudes than experimentally observed, especially for the narrow bandwidth responses of 0°-tuned neurons (Fig. 3b, c; $R^2_{\text{tuning only}} = -1.266$). Because narrow stimuli have a more regular spatial structure than broadband orientation stimuli, we hypothesized that this mismatch could be explained by the surround suppression of neural responses. We, therefore, added a modulation of the Gabor responses in the image center by responses of the spatially surrounding Gabors with

the same orientation tuning (Supplementary Fig. S4). This expanded "suppression" model accurately described the orientation dependence of neural responses to both narrow and broad orientation bandwidth stimuli and obtained a much better fit to the measured data (red lines show best-model fits; $R^2_{\text{suppression}} = 0.584$).

Based on the response amplitudes towards narrow and broad orientation bandwidth stimuli, we also computed an "orientation response modulation index" (ORI RMI) as the difference between narrow and broad orientation bandwidth responses, divided by their sum. As hypothesized, the increase in ORI RMI was strongest for neurons with orientation tuning further away from the 0° central orientation (Fig. 3d, ORI $\text{RMI}_{-45°} = 0.12 \pm 0.04$, $T = 2.79$, $p = 0.01$; ORI $\text{RMI}_{-22.5°} = 0.08 \pm 0.04$, $T = 2.11$, $p = 0.04$; ORI $\text{RMI}_{0°} = -0.03 \pm 0.04$, $T = -0.90$, $p = 0.37$; ORI $\text{RMI}_{22.5°} = 0.10 \pm 0.04$, $T = 2.4$, $p = 0.02$; ORI $\text{RMI}_{45°} = 0.13 \pm 0.04$, $T = 3.74$, $p = 10^{-3}$; mean ± s.e.m., LME model test against zero, $n = 11$ recordings from 5 mice).

In agreement with the better fit to response amplitudes (Fig. 3b, c), only the surround model accurately predicted the broadband modulation of neural response amplitudes across all orientation tunings (Fig. 3d, red line, $R^2_{\text{suppression}} = 0.395$). In contrast, the simple tuning-only model predicted a stronger decrease in neural response amplitude for neurons with 0° tuning that was not seen in the measured responses (Fig. 3d, yellow line $R^2_{\text{tuning only}} = -1.67$). Similar to neural

response modulation, we also observed a clear increase in the recruitment of neurons with orientation tuning that was further away from the central 0° orientation (Fig. 3e). To account for this modulation of neural recruitment in the model, we linearly rescaled the model fit to response modulation, assuming that increased response amplitudes should translate into observing a larger fraction of responding neurons. Again, the model that included surround suppression obtained a more accurate fit of the experimental data as the tuning-only model ($R^2_{tuning\ only} = 0.563$, $R^2_{suppresion} = 0.889$).

As in our earlier recordings (Fig. 2), expanding the spatial frequency bandwidth did not have an impact on neuronal response amplitudes, irrespective of their orientation tuning (Supplementary Fig. S5a, spatial frequency response modulation index (SF RMI): SF $RMI_{45°} = -0.06 \pm 0.06$, $T = -1.37$, $p = 0.19$; SF $RMI_{-22.5°} = -0.06 \pm 0.05$, $T = -1.38$, $p = 0.18$; SF $RMI_{0°} = -0.12 \pm 0.07$, $T = -1,37$, $p = 0.08$; SF $RMI_{22.5°} = -0.03 \pm 0.09$, $T = -0.41$, $p = 0.68$; SF $RMI_{45°} = -0.01 \pm -0.06$, $T = -0.17$, $p = 0.87$, mean ± s.e.m., LME model test against zero, $n = 11$ recordings from 5 mice). Correspondingly, we also found no changes in the recruitment of visually-responsive neurons or a clear dependence on their orientation tuning (Supplementary Fig. S5b).

Together, these results suggest that orientation tuning, as well as surround inhibition, are important factors in understanding broadband response modulation in V1. Importantly, the model predicted that, due to their more consistent spatial structure, surround suppression should be highest for narrow bandwidth stimuli, which could explain the corresponding increase in neural responses to broadband orientation stimuli as surround suppression is lifted.

## Surround modulation is crucial to predict broadband responses of individual neurons

To directly test the effect of orientation tuning and surround modulation on broadband responses of individual neurons, we performed additional experiments where we presented a set of narrow and broadband stimuli to awake 2NiellJ mice, expressing GCaMP6s in excitatory neurons (Fig. 4a). All stimuli consisted of motion clouds with a central spatial frequency of 0.04 cpd and a 0° central orientation (Supplementary Movies 1–4). The "narrow" condition consisted of motion clouds with narrow spatial frequency and orientation bandwidth (0.004 cpd and 5°, respectively). The "SF" condition consisted of broad spatial frequency and narrow orientation bandwidth stimuli (0.4 cpd and 5°), while the "ORI" condition consisted of narrow spatial frequency and broad orientation bandwidth stimuli (0.04 cpd and 45°). Lastly, the "mixed" condition consisted of both broad spatial frequency and orientation bandwidth (0.4 cpd and 45°) to assess their combined impact on neural responses (Fig. 4b). These stimuli were presented either full-field, covering the entire screen, or restricted to the center of the receptive field location for the majority of recorded neurons in each session. The size of the center stimulus was 15° to match the maximally stimulating receptive field size of V1 neurons in layer 2/3[37] (see also Supplementary Fig. S6 and "Methods"). The average RF size of the population across all imaging sessions was 30.45° ± 2.04°, ensuring that the center stimulus was placed within the receptive field of the center-responding cells.

To also identify differences in the higher order features of these different stimulus conditions, we quantified the salient orientations and stimulus regularity (linear predictability) in the center relative to the surround by computing the raw coefficient correlation[16] (Fig. 4c). While expanding the spatial frequency bandwidth still preserved the elongated structure of the underlying gratings, expanding orientation bandwidth resulted in the emergence of irregular lattice-like structures, which were less redundant than coherent elongated edges. Moreover, combining broad spatial frequency and orientation bandwidth in the mixed condition abolished most regular structures, leading to a strong decrease in stimulus predictability in the center from the surround. Furthermore, we found a clear reduction in higher-order image structure, computed as coefficient magnitude statistics[16], in the mixed condition (Supplementary Fig. S7). Interestingly, higher-order structures were increased in the ORI condition, potentially explaining its saliency to the V1 layer 2/3 population.

Similar to our earlier results, neural responses to broadband orientation, but not spatial frequency, stimuli were larger compared to the narrow condition. The same effect was seen for the mixed condition, which elicited similar responses to the ORI condition (Supplementary Fig. S8, all full-field responding neurons). This effect was also largely independent of the animal's behavioral state: while running generally increased visual responses, responses to the ORI condition remained consistently larger than responses to the narrow condition during both running and resting trials (Supplementary Fig. S9).

To isolate the impact of surround modulation on single-cell broadband responses, we selected neurons with a receptive field center that matched the location of the center stimulus and significantly responded to at least one of the center stimulus conditions (one-sided Mann-Whitney U test, $p < 0.05$, see "Methods"). In addition, we used pupil tracking to confirm that mice did not move their eyes, which could have changed the position of the neural receptive fields on the screen (Supplementary Fig. S6). For full-field stimuli, these neurons showed increased response amplitudes for the ORI condition and a similar trend for the mixed condition (Fig. 4d, $\Delta F/F_{narrow} = 3.27\% \pm 0.5\%$; $\Delta F/F_{SF} = 2.56\% \pm 0.35\%$, Wilcoxon signed rank against narrow, $p = 0.11$; $\Delta F/F_{ORI} = 3.76\% \pm 0.49$, $p = 5.9 \times 10^{-5}$; mixed = 2.98% ± 0.32, $p = 0.08$; mean ± s.e.m., $n = 378$ neurons from 5 mice). Importantly, this effect was not seen when stimuli were presented only to the receptive field centers of the recorded neurons without visual stimulation of the surround. Here, neural responses were similar between the narrow and ORI conditions, while responses to the SF and mixed conditions were lower than narrow responses (Fig. 4e, $\Delta F/F_{narrow} = 4.42\% \pm 0.37\%$; $\Delta F/F_{SF} = 4.39\% \pm 0.52\%$, Wilcoxon signed rank against narrow, $p = 7.8 \times 10^{-3}$; $\Delta F/F_{ORI} = 4.56\% \pm 0.45\%$, $p = 0.66$; mixed = 4.47% ± 0.51%, $p = 0.013$; response amplitude mean ± s.e.m., $n = 378$ neurons from 5 mice). This suggests that broadband stimulation may not primarily increase neural responses by increasing the feed-forward inputs to diversely tuned neurons but rather through a reduction of surround inhibition. To estimate the impact of surround modulation, we computed a surround modulation index (SMI) for each condition, defined as the difference between the full-field and center response divided by their sum. The more negative the SMI index, the stronger the surround inhibition for a given neuron. We found clear surround inhibition for the narrow stimulus condition that was still present for broad spatial frequency bandwidth but significantly lower for the broadband orientation and mixed conditions (Fig. 4f, $SMI_{narrow} = -0.04 \pm 0.01$; $SMI_{SF} = -0.04 \pm 0.01$, Wilcoxon signed rank test against narrow, $p = 0.06$; $SMI_{ORI} = -0.02 \pm 0.01$, $p = 2.28 \times 10^{-7}$; $SMI_{mixed} = -0.02 \pm 0.01$, $p = 1.42 \times 10^{-5}$, surround modulation index mean ± s.e.m., $n = 378$ neurons from 5 mice). Expanding the orientation bandwidth, therefore, reduces center-surround inhibition of cortical neurons, which could be due to the lower predictability of the receptive field center from the surround. The lower center responses for the SF condition also suggest that broadband spatial frequency stimuli provide weaker feedforward inputs to V1 neurons than narrow stimuli, which might also contribute to the lack of increased neural responsiveness to full-field SF stimuli.

To assess their respective importance, we then combined surround modulation, orientation tuning, and center responses to predict the broadband responses of individual neurons. We again used an LME model to account for mouse identity and included all three factors as regressors to predict broadband modulation for each stimulus condition (Fig. 4g; see "Methods"). Aside from orientation tuning, for each neuron, we also used $SMI_{narrow}$ to indicate their surround modulation. To also include changes in feedforward input, we computed the center response difference between a given modality and the narrow

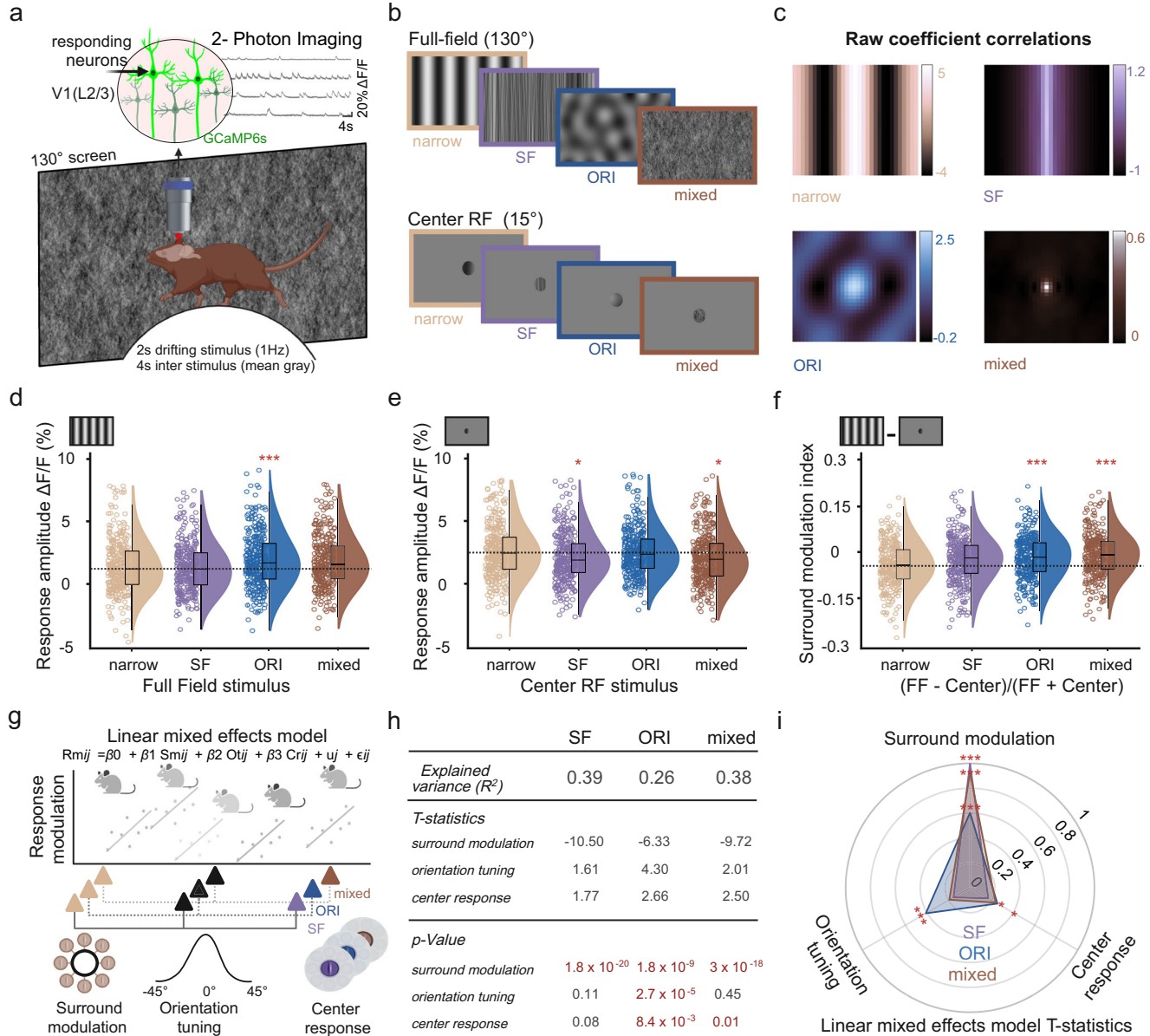

**Fig. 4 | Broadband visual stimuli modulate neural responses by reducing surround suppression. a** Visual stimulation and imaging setup. **b** Example frames from each stimulus category. Stimuli had a central spatial frequency of 0.04 cpd and orientation of 0° and shown full-field or through a 15° circular aperture, centered on the receptive field of imaged neurons. Frame colors indicate the condition identity throughout the figure. **c** Raw coefficient correlations were computed as the central samples of the spatial auto-correlation of lowpass filtered stimuli. **d** Mean response amplitude difference versus baseline for each full-field stimulus condition. Shown are all neurons with a receptive field at the center stimulus and significant responses to any of the center stimulus conditions (378 out of 8210 neurons). The horizontal line shows the median for the narrow stimulus. **e** Same as panel (**d**) but with responses to each center stimulus condition. **f** Surround modulation index for individual neurons, calculated as the difference between full-field responses in (**d**) minus center stimulus response in (**e**) divided by their sum. The horizontal line shows the median for the narrow stimulus condition. **g** Illustration of the linear mixed effects (LME) model with multiple regressors. Surround

modulation, orientation tuning, and center responses were used to predict single-cell broadband modulation for each condition. The model also considers the mouse identity for each neuron (see "Methods"). **h** Explained variance of the full LME model for each stimulus condition and T statistics and *p*-values for each regressor. Significant regressors are marked in red. **i** Polar plot to compare T statistics results for each stimulus condition. Each color-coded triangle shows the normalized *T*-values for each regressor, indicating its respective contribution to the model prediction. Across all conditions, surround modulation was the most important predictor. Box plots indicate the median (horizontal line), interquartile range (box bounds: 25th–75th percentiles), and whiskers (1.5 × interquartile range). Stars mark significant (Bonferroni correction for three tests, $\alpha = 0.0167$) differences from two-sided tests against the narrow condition: Wilcoxon signed rank (panels **d**–**f**), LME test (panel **i**). For visualization only, outliers were excluded from distributions. Panels (**a** and **g**) were created in BioRender. Balla, E. https://BioRender.com/g05l789 (2025).

condition. The model accurately predicted broadband modulation of individual neurons in all three conditions (Fig. 4h). While all regressors made different contributions in each stimulus condition, surround modulation showed the strongest and most consistent impact for all broadband stimuli. Orientation tuning had a significant impact on the

orientation bandwidth expansion, whereas the center enrichment had a smaller but more consistent impact for all conditions (Fig. 4h, i).

Together, these results show that surround modulation strongly contributes to neural responses to broadband stimuli. A potential mechanism for the increased responsiveness to broad orientation

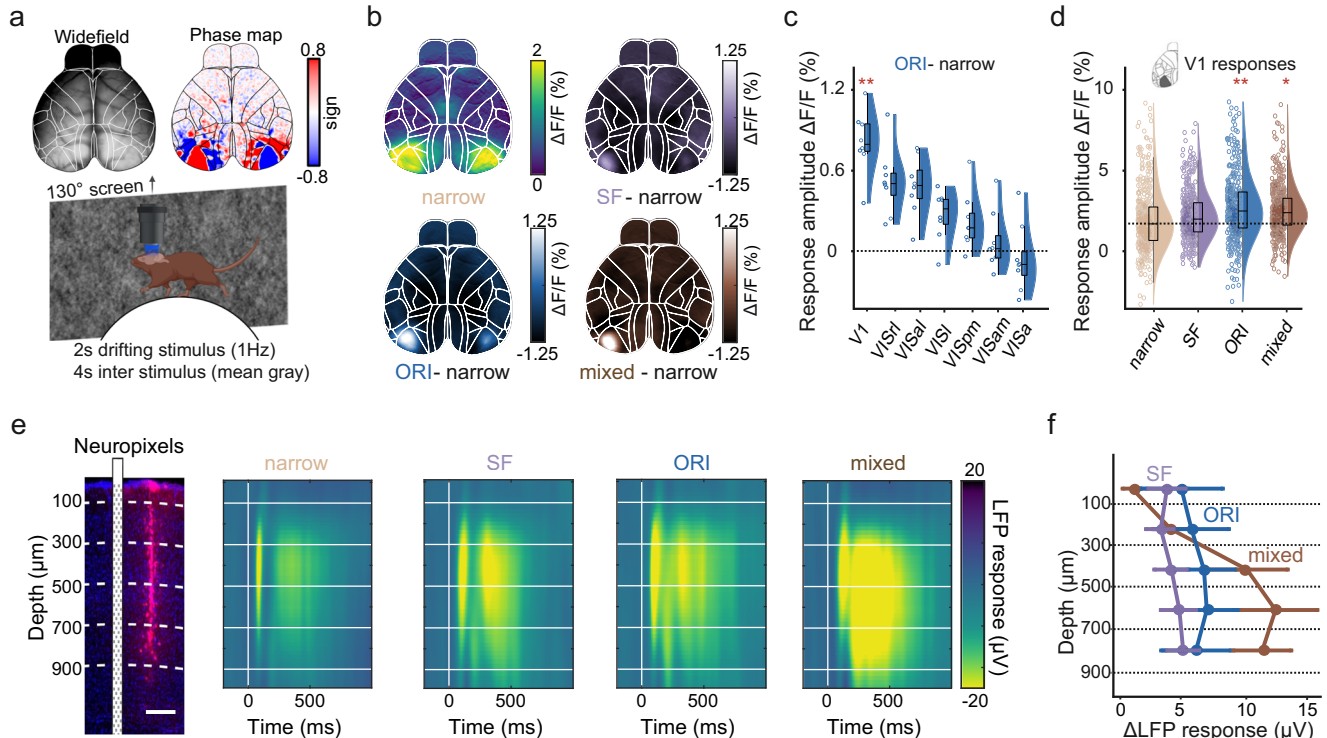

**Fig. 5 | Broadband visual stimuli increase sensory responses in cortical and subcortical neurons. a** Overview of the widefield setup and an example visual field sign map. Sign maps were used to confirm the anatomical alignment of recordings to Allen CCF (Wang, Q. et al. The Allen Mouse Brain Common Coordinate Framework: A 3D Reference Atlas. Cell 181, 936-953.e20 (2020)). Mice were head-fixed on a wheel and shown the same protocol as in Fig. 4 on the right-side screen. **b** Widefield responses averaged over the 2 s stimulus period. Mean responses to the narrow stimulus condition (top left) were subtracted from mean responses to broadband stimulus conditions. Shown are average response differences for the SF stimulus condition (top right), ORI stimulus condition (bottom left), and mixed stimulus condition (bottom right). **c** Average response differences for the ORI stimulus condition across different visual cortical areas ($n = 8$ sessions from 4 mice). **d** Individual trial response amplitudes to each stimulus integrated over the 2 s stimulus period ($n = 408$ trials from 4 mice). **e** Electrophysiological recordings. Left: Example brain slice from the V1 center, showing red fluorescence from the

Neuropixels probe positioning. A scheme of the probe is placed next to the fluorescent trace to visualize the probe position during the recording (scale bar: 200 μm). Different depths are marked by dashed white lines. Right: LFP responses to each stimulus condition across different cortical depths. Colors show voltage changes after stimulus onset relative to baseline. Horizontal white lines show depth, as in the histology panel on the left. **f** The difference in LFP responses for each stimulus condition (SF, purple; ORI, dark blue; mixed, brown) compared to the LFP response to the narrow stimulus across depths. Error bars show mean and s.e.m. ($n = 6$ recordings from 3 mice). Box plots indicate the median (horizontal line), interquartile range (box bounds: 25th–75th percentiles), and whiskers (1.5× interquartile range). Stars in all panels mark significant differences from a two-sided LME model test against narrow. The significance threshold was $\alpha = 0.0167$ after Bonferroni correction for performing three tests. For visualization only, outliers were excluded from distributions in panels (**c**, **d**). Panel (**a**) was created in BioRender. Balla, E. https://BioRender.com/g05l789 (2025).

bandwidths could, therefore, not only be the recruitment of neurons with diverse orientation tuning but also a release from surround inhibition due to the reduced predictability of the receptive field center.

### Broadband visual stimuli increase neural responses in cortical and subcortical areas

Earlier work suggested that broadband stimuli have distinct effects on neural responses in different HVAs[40]. We, therefore, used widefield calcium imaging to measure the impact of narrow, SF, ORI, and mixed stimuli on different visual areas. We again recorded from awake mice, head-fixed on a wheel, and presented full-field stimuli while imaging cortical activity through the cleared intact skull. All data was aligned to the Allen Common Coordinate Framework[51], and we additionally confirmed the location of V1 and HVAs using visual field sign mapping[49,52–54] (Fig. 5a). Presentation of the narrow stimulus resulted in strong activation of V1 and the surrounding HVAs (Fig.5b, top left). Visual responses were strongest in the left hemisphere, contralateral to the stimulus presentation, but we also observed some ipsilateral activity, especially in the binocular region of V1. Another cause for ipsilateral responses could also be reflections in the setup that reached the other eye. For our analysis, we focused on the visual areas in the contralateral hemisphere. Expanding either spatial frequency and/or

orientation bandwidth resulted in increased cortical activity that was mainly restricted to V1 (Fig. 5b). To verify this, we also computed the difference between responses towards ORI and narrow stimuli for V1 and the secondary visual areas (Fig. 5c). Here, V1 displayed the strongest increase in responses for ORI versus narrow stimuli with 0.81% ± 0.08%, confirming that broadband modulation was more pronounced in V1 compared to the surrounding HVAs (mean ± s.e.m.; ORI versus narrow: $p = 0.0048$, $T = 4.34$, LME model, $n = 8$ sessions from 4 mice). In particular, stimuli with broader orientation bandwidth evoked significantly higher activity in V1 compared to narrow or SF stimuli (Fig. 5d, mean V1 activity: $\Delta F/F_{narrow} = 1.90\% \pm 0.22\%$; $\Delta F/F_{SF} = 2.24\% \pm 0.14\%$, LME test against narrow: $T = 1.73$, $p = 0.11$; $\Delta F/F_{ORI} = 2.71\% \pm 0.25\%$, $T = 4.34$, $p = 7 \times 10^{-4}$; $\Delta F/F_{mixed} = 2.52\% \pm 0.17\%$, $T = 2.82$, $p = 0.014$, mean ± s.e.m., $n = 8$ recordings from 4 mice). However, responses to mixed stimuli were not stronger compared to broad orientation bandwidth alone (LME test for mixed versus ORI, $T = -0.77$, $p = 0.45$).

To further assess how expanding orientation and spatial frequency bandwidth modulate neuronal responses across all cortical layers, we then performed electrophysiological recordings in V1 using high-density Neuropixels probes[55] in awake mice (Fig. 5e, left). Mice were again moving on a wheel, and we presented the same

stimuli as described above. All stimulus conditions induced a clear modulation of local field potentials (LFPs), with significantly increased response modulation during broadband stimulation compared to the narrow condition (Fig. 5e, right; mean LFP response difference to narrow across layers: $\Delta LFP_{SF} = 4.28\ \mu V \pm 1.32\mu V$, $T = 3.71$, $p = 1.6 \times 10^{-3}$; $\Delta LFP_{ORI} = 6.44\ \mu V \pm 2.82\ \mu V$, $T = 2.91$, $p = 9.5 \times 10^{-3}$; $\Delta LFP_{mixed} = 9.72\ \mu V \pm 2.6\ \mu V$, $T = 3.78$, $p = 1.3 \times 10^{-3}$; mean ± s.e.m., LME model test against zero, $n = 6$ recordings from 3 mice). The earliest responses occurred in Layer 4 (300–500 μm), with the most pronounced subsequent activation in the deeper cortical layers. In agreement with our two-photon imaging results in superficial layers, broadband orientation stimuli induced stronger response modulation compared to broadband spatial frequency stimuli, which was also consistent across all layers (Fig. 5f). However, surprisingly, expanding both orientation and spatial frequency bandwidth resulted in much stronger responses in deeper cortical layers.

In agreement with our 2-photon results, we also observed a clear increase in neuronal responses to broadband orientation stimuli in the spiking activity of superficial layer 2/3 neurons (Supplementary Fig. S10a). To compare the spiking response strength across the four stimulus conditions, we calculated the response area under the receiver-operator characteristic curve. The AUC is a standardized measure for the overall separability between the baseline and stimulus period and, therefore, equally sensitive to more subtle sensory responses in sparsely active neurons. Unresponsive cells are represented by a value of 0.5, while values closer to 1 or 0 show reliable enhanced or suppressed responses, respectively. Neural responses to the ORI condition were significantly stronger than the narrow condition (Supplementary Fig. S10a, $AUC_{narrow} = 0.69 \pm 0.01$; $AUC_{SF} = 0.68 \pm 0.01$, LME model test against narrow, $T = -0.79$, $p = 0.43$; $AUC_{ORI} = 0.73 \pm 0.01$, $T = 2.72$, $p = 7 \times 10^{-3}$; $AUC_{mixed} = 0.70 \pm 0.01$, $T = 0.57$, $p = 0.57$, mean ± s.e.m., $n = 53$ neurons from 3 mice). In agreement with our LFP results, the mixed stimulus condition evoked larger spiking responses in deeper layers (500–1000 μm depth, Fig. 5f) with a corresponding shift in AUC values (Supplementary Fig. S10b, $AUC_{narrow} = 0.70 \pm 0.01$; $AUC_{SF} = 0.68 \pm 0.01$, LME model test against narrow, $T = -2.77$, $p = 5 \times 10^{-3}$; $AUC_{ORI} = 0.71 \pm 0.01$, $T = 0.94$, $p = 0.34$; $AUC_{mixed} = 0.73 \pm 0.01$, T = 2.96, $p = 3 \times 10^{-3}$; mean ± s.e.m., $n = 200$ neurons from 3 mice).

To also capture visual responses beyond the cortex, we further extended our experiments and recorded spiking activity in the SC, although we could not map the receptive fields of SC neurons (Supplementary Fig. S11). Interestingly, SC neurons responded much more strongly to the mixed stimulus condition and a smaller extent to broadband orientation and spatial frequency stimuli compared to narrow stimuli (Supplementary Fig. S11d, $AUC_{narrow} = 0.68 \pm 0.02$; $AUC_{SF} = 0.68 \pm 0.02$, LME model test against narrow, $T = 0.11$, $p = 0.91$; $AUC_{ORI} = 0.70 \pm 0.02$, T = 1.10, $p = 0.27$; $AUC_{mixed} = 0.82 \pm 0.02$, T = 5.44, $p = 4.46 \times 10^{-6}$, mean ± s.e.m., $n = 46$ neurons from 3 mice).

Together, these results show that increased cortical activation by broadband visual stimuli is mainly restricted to V1. While superficial layers show a clear preference for broadband orientation, deeper layers were also strongly activated by mixed stimuli with broadband orientation and spatial frequency. Interestingly, this is also the case for the subcortical superior colliculus, which receives direct input from the retina but also from deeper layers of V1.

### Neural discrimination capability is enhanced by broader orientation bandwidth

We next wondered if expanding orientation bandwidth also improves the neural representation of sensory information, thus increasing the discriminability of different broadband stimuli from each other. To test this hypothesis, we presented stimuli with a different central orientation (0° or 90°) or central spatial frequency (0.04 cpd or 0.16

cpd) and measured the impact of expanding orientation or spatial frequency bandwidth on the ability of neurons to discriminate between the central features, i.e., the spatial frequencies will be discriminated while orientation bandwidth is expanded and vice-versa. (Fig. 6a, e). To avoid any ambiguity, we did not change the bandwidth of stimulus orientations or spatial frequency together but instead compared neural responses to the central feature while changing the bandwidth of the other. Similar to our previous results, expanding the orientation bandwidth with a central spatial frequency of 0.16 cpd also increased the number of visually responsive neurons with a $1.36 \pm 0.14$ fold increase compared to the narrow condition (Fig. 6b, $T = 2.75$, $p = 9 \times 10^{-3}$; LME test against 0; percentage responsive neurons over total neurons per session: narrow range (5°): 11% ± 1%, mid-range (25°): 15% ± 2%, broad range (45°): 21% ± 2%, mean ± s.e.m, $n = 18$ sessions from 9 mice). The increase was lower with a central spatial frequency of 0.16 cpd compared to 0.04 cpd (LME test, $T = 3.91$, $p = 4.1 \times 10^{-5}$), potentially because higher spatial frequency stimuli were less efficient in driving neural responses which may have also reduced the impact of broadband response modulation (Fig. 4e). The amplitude of neural responses was also increased, demonstrating that expanding orientation bandwidth increases neural responsiveness, regardless of the central spatial frequency (Fig. 6c; orientation modulation index: ORI $RMI_{0.04cpd} = 0.17 \pm 0.01$, LME test against zero, $T = 14.3$, $p = 7.27 \times 10^{-45}$; ORI $RMI_{0.16cpd} = 0.06 \pm 0.01$, $T = 4.3$, $p = 1.53 \times 10^{-5}$; mean ± s.e.m., $n = 1345$ neurons from 18 recordings from 9 mice). Such increased neural responsiveness to broadband stimuli might enhance the discriminability of different spatial frequencies by driving stronger responses in a larger population of V1 neurons. Conversely, broadband stimulation may also drive the same broadband-selective neurons irrespective of the central stimulus feature. However, neuronal responses to broadband stimuli with different central orientations or spatial frequencies were not strongly correlated, suggesting that most neurons respond differently to broadband stimuli when changing the central orientation or spatial frequency (Supplementary Fig. S12).

To assess if expanding orientation bandwidth increases the neuronal discriminability between two central spatial frequencies, we computed a discrimination index ($AUC_{abs}$) as the absolute AUC between neural responses to stimuli with a central spatial frequency of either 0.04 cpd or 0.16 cpd (see also Supplementary Fig. S13). $AUC_{abs}$ was normalized between 0 and 1, with larger values indicating increased discriminability of the central spatial frequency and we computed the discriminability for either narrow or broad orientation bandwidth stimuli (Fig. 6d, $AUC_{abs,\ 5°} = 0.30 \pm 0.01$; $AUC_{abs,\ 45°} = 0.36 \pm 0.01$, LME model test against narrow condition, $T = 8.13$, $p = 5.44 \times 10^{-16}$; mean ± s.e.m., $n = 2096$ neurons from 18 sessions, 9 mice). The discriminability between central spatial frequencies was significantly increased with broader orientation bandwidth, demonstrating that broadband orientation stimuli increase cortical responses, which enhances the neural representation of other stimulus features.

In contrast, broadening the spatial frequency bandwidth did not increase the number or magnitude of neural responses to stimuli with a central orientation of either 0° or 90° (Fig. 6f, number of visually-responsive neurons compared to the narrow condition: broad SF bandwidth (90° central orientation) = $1.09 \pm 0.21$, $T = 0.42$, $p = 0.67$, LME model test against zero, mean ± s.e.m; Fig. 6g, spatial frequency modulation index: SF $RMI_{0°} = -0.004 \pm 0.02$, $T = -0.17$, $p = 0.86$; SF $RMI_{90°} = 0.014 \pm 0.02$, T = 0.6, $p = 0.53$, mean ± s.e.m., $n = 16$ recordings from 9 mice). Consequently, we found no significant change in the neural discriminability of stimuli with different central orientations when expanding spatial frequency bandwidth (Fig. 6h, orientation discrimination index: $AUC_{abs,\ 0.004cpd} = 0.30 \pm 0.01$; $AUC_{abs,\ 0.4cpd} = 0.32 \pm 0.01$, LME model test against the narrow condition, $T = 1.8$, $p = 0.06$, mean ± s.e.m., $n = 888$ neurons from 9 mice).

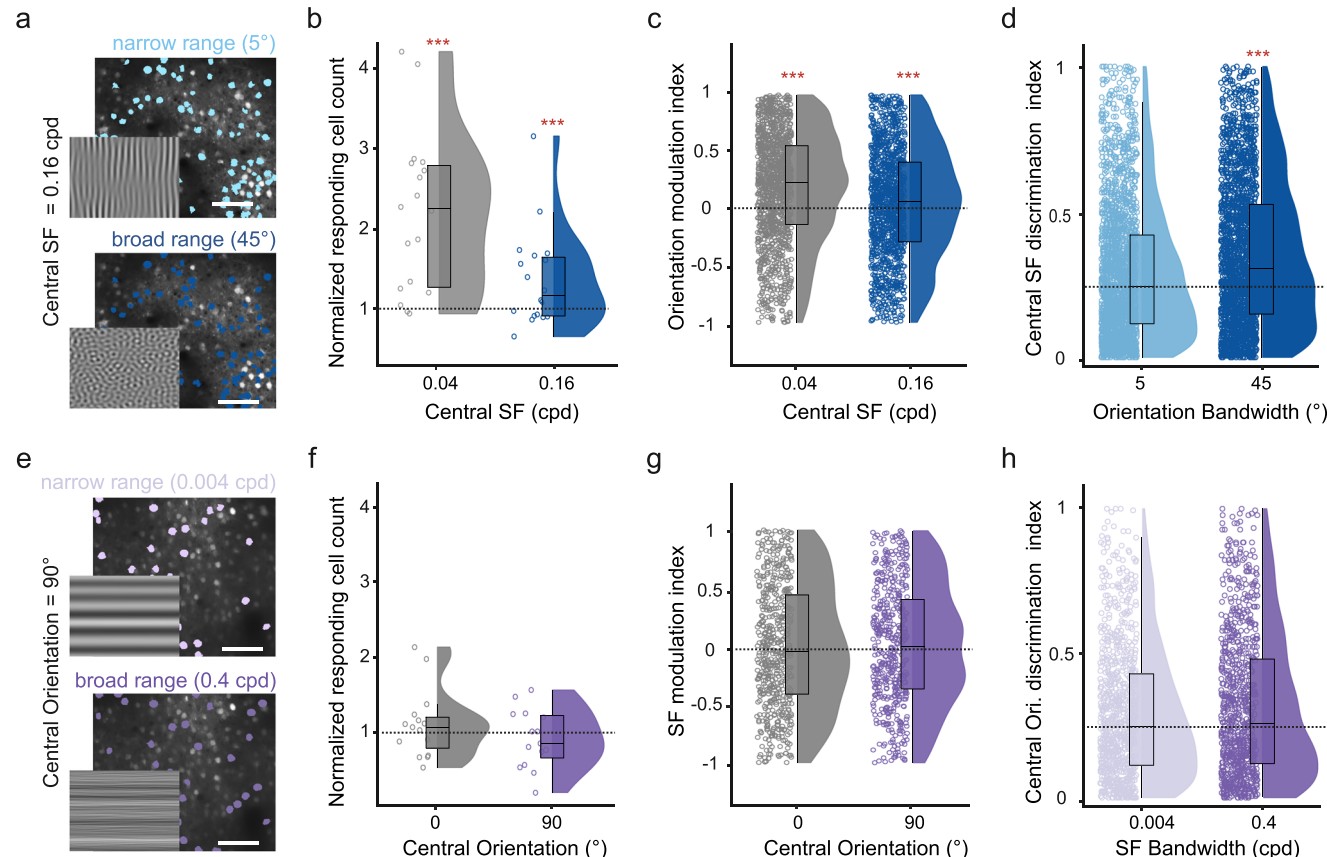

**Fig. 6 | Expanding orientation bandwidth improves stimulus discriminability in V1 neurons. a** Example field of view of an experiment with the central spatial frequency of 0.16 cpd and narrow or broad orientation bandwidth. Stimuli are shown at the bottom left of the plane, scale bar: 100 μm. **b** Number of responsive cells to broad orientation bandwidth (45°) with either 0.04 cpd or 0.16 cpd central spatial frequency, normalized to the respective number of cells responding to the narrow condition (5°). Data for 0.04 cpd are the same as in Fig. 1 and shown here for reference in gray. **c** Orientation modulation index, calculated as mean broad minus narrow orientation bandwidth responses, divided by their sum. Shown are all positively narrow and broad responding neurons for each central spatial frequency ($n = 1333$ responsive neurons for central SF = 0.04 cpd in gray, similar to data in Supplementary Fig. S1c, $n = 1345$ responsive neurons for central SF = 0.16 cpd in blue. $n = 4892$ neurons in total across 18 sessions from 9 mice). **d** Discriminability between neural responses to stimuli with a central spatial frequency of either 0.04

cpd or 0.16 cpd. Shown is the discrimination index for narrow and broad orientation bandwidth in light and dark blue, respectively ($n = 2096$ neurons from 18 sessions). Shown are the results for all cells that responded to any of the presented stimuli. The horizontal black dotted line shows the median discriminability for the narrow band. **e** Example field of view of an experiment with 90° central orientation and narrow or broad frequency bandwidth, scale bar: 100 μm. **f–h** Same as panels (**b–d**) but for narrow and broad frequency bandwidth and 0° and 90° central orientation. Box plots indicate the median (horizontal line), interquartile range (box bounds: 25th–75th percentiles), and whiskers (1.5 × interquartile range). Stars in all panels mark significant (Bonferroni correction for two tests, $\alpha = 0.025$) differences from a two-sided LME test against 1 (panels **c**, **g**) or the narrow bandwidth (panels **b**, **d**, **f**, **h**). For visualization only, outliers were excluded from distributions in panels (**b–d**, **f–h**).

## Visual perception is improved by expanded orientation bandwidth

Our results show that neural responses in V1 are enhanced when expanding orientation bandwidth, which also increases the neuronal discriminability of these broadband visual stimuli. To assess if increased neuronal discriminability also results in enhanced visual perception, we trained mice to perform a visual discrimination task while freely moving in a custom-built touchscreen chamber[28] (Fig. 7a). Here, mice had to discriminate between motion clouds by touching one of the two presented stimuli on a touch-sensitive screen and were rewarded when choosing the target stimulus. For two groups of mice (6 mice each), motion clouds either differed in their central spatial frequency (0.16 cpd target versus 0.04 cpd non-target; Fig. 7b, left, central orientation was 0° for both) or central orientation (0° target versus 90° non-target; Fig. 7b, right, central spatial frequency was 0.04cpd for both). In addition, we again varied the orientation or frequency bandwidth of the presented stimuli to test their impact on the animals' discrimination performance. As in our earlier experiments, the bandwidth was only altered for the stimulus parameter that

was not tested in the discrimination task to avoid any ambiguity. In the orientation discrimination task, we thus varied spatial frequency bandwidth while varying orientation bandwidth during spatial frequency discrimination (Fig. 7b).

Animals were split into two groups and either trained on orientation or spatial frequency discrimination. Both groups reliably learned to perform their respective task and achieved comparable expert discrimination performance. Expanding the orientation bandwidth also improved the spatial frequency discrimination performance, demonstrating that broadband orientation stimuli not only improved neural response discriminability but also enhanced spatial frequency perception (Fig. 7c, percentage of correct trials: narrow (5°) = 77.0% ± 1.93%; mid-range (25°) = 77.63% ± 1.62%, Wilcoxon signed rank against narrow condition: $p = 0.92$; broad (45°) = 82.77% ± 1.41%, $p = 4.9 \times 10^{-3}$, mean ± s.e.m., $n = 46$ sessions from 6 mice). Conversely, expanding spatial frequency bandwidth did not further improve orientation discrimination performance and even showed a trend towards reduced performance with higher spatial frequency bandwidth (Fig. 7d, percentage of correct trials: narrow (0.004 cpd) =

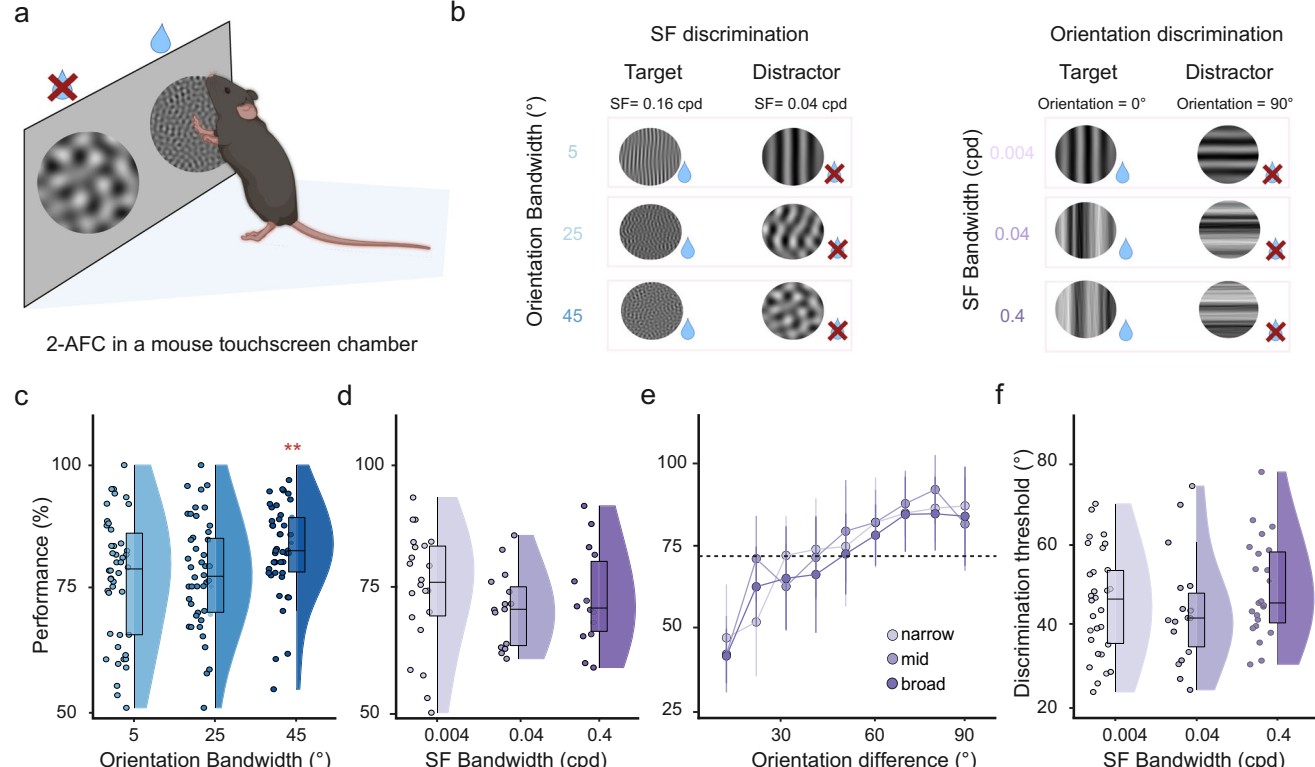

**Fig. 7 | Expanding orientation bandwidth improves visual perception.**
**a** Illustration of the visual discrimination task. The mouse initialized the trial by triggering the lick detection, followed by a 500 ms long ITI with a gray screen, followed by a visual cue (white cross) for 700 ms. Two stimuli were then shown next to each other, and the mouse had to touch one stimulus to report a choice. Mice then received a water reward at a central lick spout after touching the target stimulus. Triggering the lick detection (even if no reward was given) initiated the next trial. **b** Example stimuli for the spatial frequency discrimination task with different orientation bandwidths (left) and the orientation discrimination task with different spatial frequency bandwidths (right). **c** Spatial frequency discrimination performance for different orientation bandwidths. Spatial frequency discrimination performance was significantly higher for larger orientation bandwidth ($n = 46$ sessions from 6 mice). **d** Same as in panel **c**) but for the orientation discrimination performance with different spatial frequency bandwidth stimuli. ($n = 25,17,19$ sessions respectively.) **e** Psychometric curves for orientation

discrimination performance at different target-distractor orientation differences. Colors show psychometric curves for different spatial frequency bandwidths. Expanding the spatial frequency bandwidth did not affect the maximal discrimination performance or discrimination thresholds. The horizontal dashed line shows the 72.7% discrimination threshold. Error bars are centered at the colored circles and show standard deviation ($n = 30,18$ and 24 sessions, respectively). **f** Discrimination thresholds for the three spatial frequency bandwidths showed no significant differences ($n = 30, 18$, and 24 sessions, respectively). Box plots indicate the median (horizontal line), interquartile range (box bounds: 25th–75th percentiles), and whiskers (1.5 × interquartile range). Stars in panels (**c**, **d**, and **f**) show the significance of a two-sided Wilcoxon signed rank test against the narrow condition. The significance threshold was $\alpha = 0.025$ after Bonferroni correction for performing 2 tests. Panel (**a**) was created in BioRender. Balla, E. https://BioRender.com/g05l789 (2025).

72.53% ± 2.85%; mid-range (0.04 cpd) = 69.18% ± 2.12%, Wilcoxon signed rank test against narrow percentage of correct trials, $p = 0.79$; broad (0.4 cpd): 65.34% ± 3.60%, $p = 0.29$, mean ± s.e.m., $n = 24, 16$ and 18 sessions for each condition from 6 mice).

These findings strongly suggest that the enhanced neural discriminability of V1 responses translates into corresponding changes in visual perception, with the greatest discrimination performance observed for broad orientation bandwidths. However, a potential reason for the unchanged discrimination performance across spatial frequency bandwidths could be that perceptual changes are only visible near the frequency discrimination threshold, where even small perceptual variations can effectively impact behavior. We, therefore, further tested different spatial frequency bandwidths over a larger range of orientation differences (Fig. 7e). Using a staircase procedure, we dynamically changed the orientation difference between target and non-target stimuli to identify the orientation discrimination threshold for each mouse within individual sessions. Consistent with our earlier results, the obtained discrimination thresholds and psychometric curves were similar across the tested spatial frequency bandwidths, again arguing against a notable impact of spatial frequency bandwidth on orientation discrimination (Fig. 7f, discrimination threshold at

72.7% orientation discrimination threshold from staircase procedure: narrow (0.004 cpd) = 46.54° ± 2.33°; mid-range (0.04 cpd) = 49.93° ± 2.53°, Wilcoxon signed rank test against narrow, $p = 0.61$; broad range (0.4 cpd) = 49.34° ± 2.24°, $p = 0.28$, mean ± s.e.m, $n = 18, 23$ and 30 sessions respectively). Together, these findings show that broadband orientation, but not spatial frequency, stimuli can not only increase the responsiveness and discriminability of neural responses but also enhance visual perception.

## Discussion
We used motion cloud stimuli with comparable bandwidths as in natural scenes to test the impact of broad orientation and spatial frequency bandwidth on neural responses and visual perception. Broadband stimuli elicited diverse V1 responses in awake mice, but only expanding orientation bandwidth increased V1 responses, reflected in more responsive neurons and larger response amplitudes (Figs. 1, 2). Aside from recruiting additional neurons with diverse orientation tuning, our modeling and experimental results showed that a key contributor to this effect is a broadband-specific reduction in surround inhibition (Figs. 3, 4). Moreover, our electrophysiological recordings show that mixed stimuli drive particularly strong responses in deeper

cortical layers and the SC (Fig. 5), potentially due to their lower stimulus predictability. This prominent increase in response strength could be an adaptive mechanism of the visual system to enhance the perception of naturalistic broadband stimuli. Indeed, expanding the orientation bandwidth of visual stimuli also improved the discriminability of neural responses to visual stimuli with different spatial frequencies (Fig. 6). Lastly, we tested if such improved neural encoding is translated into enhanced visual perception and found that mice were more accurate in discriminating visual stimuli with broad compared to narrow orientation bandwidth (Fig. 7). Together, these results demonstrate that broadband stimuli engage V1 neurons by reducing surround inhibition, with orientation broadband stimuli increasing neural response strength, stimulus discriminability, and visual perception.

Our results of increased neural responsiveness to broadband orientation stimuli are in agreement with earlier work, showing that noise stimuli with broadband spatiotemporal frequencies and orientations can increase V1 responses[38,39]. An intuitive explanation for this effect is the additional recruitment of orientation-tuned neurons by stimuli that cover a broader range of orientations. To test this hypothesis, we used a computational model and found that orientation tuning is indeed a likely contributor to increased broadband responses. However, to accurately fit the measured response amplitude of orientation-tuned neurons, we also had to include response modulation by the stimulus surround. While the suppressive effect of narrow bandwidth grating stimuli that extend the receptive field size of cortical neurons has been well described[31,34,37,56], we found that broadband orientation stimuli release neurons from this surround suppression and thus enhance responses across a large fraction of the V1 population.

This difference in surround suppression could be due to a reduction in the predictability of the receptive field center. When presenting natural images to awake monkeys, center-surround interactions in V1 are sensitive to higher-order structures, such as image contours[57]. In contrast, randomizing the phases of the Fourier spectrum of the image surround diminishes center-surround interaction, depending on the inferred redundancies in the natural image[58]. Similarly, expanding the orientation bandwidth in random phase motion clouds resulted in visual stimuli with a low raw coefficient correlation and, therefore, a low predictability of the stimulus center by its surround[16]. In line with the predictive coding theory[59], stating that responses in the receptive field center should be suppressed if they can be predicted by the surround, we found that such reduced center predictability is indeed related to a reduction in surround inhibition. Aside from feature tuning of individual neurons, center-surround modulation is, therefore, a major driver of increased neural responses. Moreover, our results suggest that changes in surround modulation are also likely to promote recently described response features of V1 neurons, such as preferred responses to isotropic stimuli with shorter edges[40].

In contrast to orientation bandwidth, expanding the spatial frequency bandwidth had less impact on raw coefficient correlation and surround suppression. Together with lower center responses, this difference in stimulus predictability could also explain the selective increase in neural responses to broadband orientation but not spatial frequency stimuli. Broader neural tuning to spatial frequencies compared to orientations[29,48,49] might also result in an unselective activation of broadly responding neurons with different spatial frequency bandwidths. If neurons were broadly tuned to spatial frequency, they would be expected to respond to all spatial frequency bandwidth stimuli. However, our data show that they are not tuned broadly relative to the tested spatial frequencies. When we directly compared the tuning of V1 neurons to the same range of orientations and spatial frequencies that we used in our motion cloud stimuli, the responses of individual neurons covered a largely similar range of stimulus bandwidths for both features (Supplementary Fig. S2). We also observed a

larger proportion of cells responding to all orientation bandwidth stimuli than to the full range of spatial frequency bandwidth motion clouds (Figs. 1g, 2g), further suggesting that differences in neural tuning cannot explain the response preference to broadband orientation stimuli.

Previous studies have shown that HVAs can be tuned to different spatial frequencies, to different speeds, or show stronger orientation tuning[29,40]. Several areas are also more tuned to natural textures than scrambled images or narrow bandwidth gratings[6,18]. We, therefore, used widefield imaging to explore if areas outside of V1 show similar or stronger tuning to broadband visual stimuli. However, broadband response modulation was strongest in V1, demonstrating its importance for processing broadband stimuli. Our electrophysiological measures from deeper cortical layers and the SC also showed that expanding orientation and spatial frequency bandwidths together can further increase neural responses. The lower impact of spatial frequency bandwidth might, therefore, be specific to the supragranular V1 layers. The stronger response modulation to the mixed stimulus condition in the deeper LFP but also the widefield recordings could be explained by dendritic signals of layer 5 neurons, which also contribute to widefield signals[60]. We also found stronger responses to broadband spatial frequency compared to narrow stimuli in the LFP and widefield, which were not seen in the 2-photon and spiking recordings. A potential reason for this difference could be that LFP and widefield are population measures that also represent dendritic and axonal signals that may not be reflected in somatic spiking. It is, therefore, possible that broadband spatial frequency stimuli have a more subtle impact on V1 inputs that was not detected in our single-cell measures of V1 neural activity.

Superficial V1 layers have also been implicated with processing prediction errors during motor-visual mismatch[61–64], stimulus sequences[65], and between center and surround[30,36,37]. In contrast to the preponderance of prediction-error neurons in layer 2/3, the deeper layer 5 has been implicated with the representation of the internal predictions[64,66]. Furthermore, V1 neurons have an additional receptive field in the surround with different visual tuning properties compared to the center receptive field[37]. The interaction of center and surround is thought to be tuned to the characteristics of natural scenes and might enhance natural pattern completion[67], which could explain the observed layer-dependence in responses to broadband visual stimuli. Co-stimulation of the center and the surrounding receptive field also leads to sparser neuronal responses and increased information coding[32–34,68] due to local network interactions that sharpen recurrent excitation to produce specific and reliable visual responses[33]. Together, increased neural responses to broadband orientations appear to be a specific feature of the visual cortex and could be an adaptation to the orientation distributions in natural scenes.

Superposition of differently oriented gratings can suppress cortical responses in anesthetized cats[69–73] and monkeys[39], while facilitated responses have been reported in awake monkeys and mice[40,74–79]. Surround suppression has been shown to be strongly reduced under anesthesia[31,37]; therefore, a release from surround suppression by broadband visual stimuli might be less effective. The large variety in cortical responses to multiple superimposed gratings could also be explained by orientation-specific horizontal interactions within V1[74,75,80] or a thalamocortical feedforward mechanism[77]. Furthermore, motion cloud stimuli have been used in human studies to test how visual speed processing depends on spatial frequency bandwidth[81]. While visuomotor reflexes were enhanced, perceptual speed discrimination was impaired for stimuli with large bandwidths. Since mice also possess speed-tuned neurons in the visual cortex[29,82], it would, therefore, be interesting to also test speed perception in mice using different bandwidth-enriched stimuli. Moreover, visual flow patterns of small moving bars evoke strong cortical responses that are different from those evoked by gratings and are also evoked at much higher spatial frequencies[83]. This suggests an additional mode of perception

for global motion patterns, which might be related to our observation of increased responses to broad bandwidth motion clouds[80,83].

Theoretical studies using convolutional neural networks (CNNs) also showed that including surround suppression can considerably improve both the performance and training speed of traditional CNNs in visual tasks, showing a superior generalization capability in different lighting conditions[84]. Interestingly, we found that broadband orientation stimuli also improved spatial frequency discrimination, arguing that lower stimulus predictability can improve the sensory perception of specific visual features. Heightened sensory perception also depends on the saliency of a visual stimulus, which can be influenced by its surroundings, allowing it to merge or pop out from the rest of the visual scene[30,36,85–87]. This pop-out effect is even stronger in the SC[88] which is known to encode a saliency map of visual scenes primates[89]. Our finding that SC neurons strongly respond to heterogeneous broadband visual stimuli with low spatial correlations could suggest that less-predictable stimuli are also of higher saliency and, therefore, recruit SC circuits more strongly than the visual cortex. An interesting topic for future studies would, therefore, be a more detailed comparison of cortical and subcortical processing of broadband stimuli.

Natural vision is broadband vision, and gratings of a single orientation and spatial frequency are rarely present in a natural environment. The visual system is tuned to the statistics of natural inputs[2,3], which is represented in the ability of mice and humans to discriminate fine differences in visual features that are most common in the environment[2,29,90]. Natural scenes also consist of distributions of largely overlapping orientations around areas of interest[1] (Fig. 1a), leading to complex receptive fields of cortical neurons that respond more strongly to visual features when they are embedded in natural scenes instead of random or artificial backgrounds[91]. Revealing the computational principles of natural scene processing is therefore crucial in order to understand visual processing at large. Our results with broadband orientation motion clouds demonstrate that rich but parametrically controlled visual stimuli are a promising tool to achieve this goal and reveal how the visual cortex has evolved to perceive our natural environment.

## Methods

### Animals

All experiments were carried out in accordance with the German animal protection law and local ethics committee (LANUV, NRW) under the study protocols 84-08.04.2016.A357 and 81-02.04.2021.A021. All mice (*Mus Musculus*) were between 14–30 weeks old. For the two-photon imaging experiments, six male and three female C57BL/6 J (Charles River, Germany) mice (Figs. 1 and 2 show data from the same 9 mice, but each protocol was run separately on different days in randomized order) and five 2Niell/J (B6; DBA-Tg(tetO-GCaMP6s)2Niell/J, Charles River, Germany) females (Figs. 3 and 4) were used. For the widefield imaging experiments, two male and two female 2Niell/J mice were used. For the electrophysiological recordings, three 2NiellJ males were used. For the behavioral discrimination tasks, three female and three male C57BL/6 J mice were used for the spatial frequency discrimination task; for the orientation discrimination task, six male C57BL/6 J mice were used. All mice were kept on a reverse light cycle (light/dark cycle: 12/12 h) with regulated temperature and humidity conditions (23 ± 2 °C, 55 ± 5 % air humidity). Mice in the behavioral task were water-restricted throughout the experiments. They received water ad libitum on the weekend when no experiments were performed and received at least 1.5 ml of water on experimental days. Mice had access to food ad libitum and were weighed and checked for their health status before the start of each behavioral session.

### Surgical procedures

For the cranial window implantation and viral injection, the mice were weighed and given a subcutaneous injection of Carprofen (4 mg/kg,

Rimadyl, Zoetis GmbH) and Buprenorphine (0.1 mg/kg, Buprenovet sine, Bayer Vital GMBH) for analgesia, 20 min before the surgical procedure. They were then anesthetized using (1%–2.5%) isoflurane in oxygen and placed in a stereotaxic frame with stabilized and monitored body temperature (37 °C). Eye ointment (Bepanthen, Bayer Vital GmbH) was placed on the ipsilateral eye and ophthalmic gel in the contralateral to keep them moisturized but also allow intrinsic imaging to identify the location of V1[29,48,52,54]. Following a local injection of Bupivacaine (0.08 ml of 0.25% Bucain 7.5 mg/ml, Puren Pharma GmbH), the skin was incised and pushed aside to reach the skull around the position of V1, based on anatomical coordinates (3.5 mm posterior and 2.5 mm lateral from bregma). The skull was then carefully thinned with a dental drill to gain optical access to the cortex for intrinsic imaging (see section 'Intra-surgical intrinsic imaging'). A 4-mm-wide circular craniotomy was then performed over the center of V1. In the C57BL/6 J mice for two-photon imaging, a viral vector (AAV9.Syn.GCaMP6f.WPRE.SV40; Addgene, viral titer ~$10^{13}$ vg/ml, diluted 1:10 in phosphate-buffered saline (PBS)) was then injected through a thin glass pipette with a micropump (UMP-3, World Precision Instruments). In each mouse, we injected 250 µl at 100 nl/s in two separate V1 locations (~ 500 µm apart from each other) to obtain an even expression GCaMP6f throughout the area (We waited around three weeks for stable expression before imaging). The dura was kept moist with PBS solution throughout the injection. After viral injections, a 4 mm-wide round coverslip was placed inside the craniotomy and fixed with light-curable dental cement (DE Flowable composite, Nordenta). For electrophysiology experiments, we used coverslips with a small opening (~ 0.2 mm) that was covered with silicon to allow access of the Neuropixels probes to the cortex. A metal head holder was then positioned and attached with dental cement to allow subsequent head fixation and imaging in the two-photon microscope setup. In the 2Niell/J mice for the two-photon experiments, the head-bar placement followed immediately after the window placement, and the location of V1 was identified later with widefield field sign mapping (see section 'Widefield imaging'). The custom stainless steel headbar piece was held in place with quick adhesive cement (Superbond C&B, Sun Medical CO., LTD) and fast-curing acrylic resin (Jet Denture repair, Lang Dental INC). For widefield imaging, 2Niell/J mice were similarly prepared, but a skull clearing procedure was performed instead of the cranial window[92]: the skull was thoroughly cleaned and covered with a thin, transparent layer of cyanoacrylate (Zap-A-Gap CA +, Pacer technology) to obtain optical access to the cortical surface[93].

After the surgery, all animals received the same injections of Carpofen and Buprenorphine as in the beginning and were kept over a heating blanket during their immediate recovery. They also received Buprenorphine (0.009 mg/ml, Buprenovet sine, Bayer Vital GmbH) and Enrofloxacin (0.0227 mg/ml, Baytril 5%, Bayer Vital GMBH) in their drinking water for three days. The recovery time was at least one week before starting animal handling.

### Intra-surgical intrinsic imaging

During cranial window surgeries, retinotopic mapping was performed using intrinsic signal optical imaging to functionally confirm the location of V1. Here, we presented continuous periodically drifting bars with a high contrast flickering checker pattern[49,52]. Stimuli were created using Psychtoolbox[94,95] in MATLAB (MATLAB R2019b, MathWorks) and composed of a static checker pattern with a patch size of 20° and contrast reversal at 3 Hz. This pattern was masked and only visible through a 15°-wide bar aperture, moving over the screen with a temporal period of 18 s in all four cardinal directions in randomized order. Stimuli covered roughly 120° horizontally and 80° vertically and were pre-rendered in advance. A spherical correction was applied to compensate for distortions due to the presentation on a flat monitor[49]. The monitor (BenQ XL2420T) was positioned at an angle of 20° to the midline and tilted 10° above the animal. The eye was positioned at the

horizontal center of the monitor, 30 mm above the lower edge of the monitor, and the stimulus presentation was adjusted accordingly. A vessel image for alignment was captured under green-light illumination (peak ~ 525 nm, green power LED, 15506, LUMITRONIX). Hemodynamic signals were acquired under red-light illumination (peak ~ 630 nm, red power LED, 15607, LUMITRONIX). Images were captured using a tandem-lens macroscope composed of a 25 mm objective (DO-2595, Navitar) above the subject and a 50 mm objective in front of a CMOS camera (acA1920-155um, Basler). In between, a removable bandpass filter was used to isolate reflected red light (630 nm – 760 nm, G384082035, Qioptiq). Phase maps of horizontal and vertical retinotopy were generated online using custom software based on previously published analysis[54]. Visual field sign maps, delineating the precise borders of V1 to higher visual areas, were then computed after imaging was completed[52]. During intrinsic imaging, anesthesia was kept light, just above 1% isoflurane in oxygen. Throughout retinotopic mapping, the state of anesthesia was closely monitored using an additional camera viewing the animal, as well as screening for global drifts in the hemodynamic signal.

## Two-photon imaging
Two-photon imaging was done in a custom-built setup with a resonant-scanning two-photon microscope (CRS 8KHz, Cambridge Technology) and a femtosecond-pulsed TI: Sapphire laser (Mai Tai DeepSee, Spectra-Physics) tuned to 920 nm. Data acquisition and controls were performed with the MATLAB-based Scanimage package (Scanimage 2015)[96]. Settings and laser power were kept constant across all mice and experiments. References from the vessel image, visual field sign maps, and/or injection site referencing were used to define the center of the primary visual cortex. Recordings were made from a single plane in layer 2/3 of V1 at a depth of 250–300 μm below the pial surface with a frame rate of 30 Hz. We used a 16x, 0.8 NA Nikon objective lens (CFI75 LWD 16X W) at a pixel resolution of 512 × 512 pixels (575 μm × 575 μm). Motion Cloud stimuli from the orientation bandwidth expanding and the spatial frequency bandwidth expanding protocol (Figs. 1, 2, and 6, see 'Visual stimulation' section below) were presented at 17 cm distance from the right eye on a gamma-corrected LED-backlit LCD monitor (BenQ XL2420T, 24", 60 Hz refresh rate, or Viewsonic VX3276-2K-MHD-2, 32", 60 Hz refresh rate). The larger screen was used to increase the visual field coverage for center-surround stimuli, where we also used a sparse noise stimulus[97] to map the preferred RF locations of the individual neurons in the field of view of each recording. Two behavior cameras (Firefly DL Teledyne FLIR) recorded the animal from different angles throughout the sessions for movement and eye tracking.

## Widefield imaging
Widefield Ca²⁺-imaging was performed using a custom tandem-lens macroscope[98], consisting of two 85 mm objectives (Walimex Pro 85 mm f/1.4 IF; Walimex) using a sCMOS camera (Edge 4.2, PCO) through the cleared intact skull. Frames were acquired using the Python-based software package Labcams (Version 0.2 https://github.com/jcouto/labcams, by Joao Couto) at a framerate of 30 Hz with an effective resolution of 512 × 512 pixels with 4 × 4 spatial binning. Frames were acquired under alternating illumination using a blue LED (470 nm, M470L3, Thorlabs) and a violet LED (405 nm, M405L3, Thorlabs) with a 405 nm excitation filter (#65-133, Edmund optics). The excitation light was collimated using adjustable collimator lenses (SM2E, Thorlabs). Using a dichroic mirror (no. 87–063, Edmund optics), both excitation light paths were merged and then reflected onto the brain surface using a second dichroic mirror (495 nm long-pass, T495lpxr, Chroma). GCaMP6s fluorescence signals were isolated using a 525 nm emission filter (86–963, Edmund optics) in front of the camera. Images acquired under violet illumination captured calcium-independent fluorescence at the isosbestic point of GCaMP[99].

Therefore, by subtracting the linearly-rescaled calcium-independent signal from the calcium-dependent signal acquired under blue illumination, we could remove the intrinsic signal due to hemodynamic fluctuations.

Widefield imaging data were motion corrected using a subpixel image registration routine[100] was performed separately for frames acquired under blue or violet illumination. Then, a singular value decomposition (SVD) was used to reduce the dimensionality of the data and computational cost associated with the analysis[101]. All subsequent analysis across time was performed on the 500 components that described the highest variance in the imaging data. Furthermore, a zero-phase, second-order Butterworth filter was used to remove slow signal drifts below 0.1 Hz before performing the hemodynamic correction. All subsequent analyses were based on this low-dimensional and hemodynamic-corrected signal.

All recordings were aligned to the Allen CCF based on anatomical landmarks, and the location of V1 and higher visual areas (HVAs) was additionally confirmed using retinotopic mapping as described above[49,53]. Due to the faster neural signals with GCaMP-related widefield imaging, we used a faster movement of the bar aperture (2 s instead of 18 s per period with 20 repetitions for each direction). The same retinotopic mapping procedure was also used for mice in the two-photon and electrophysiology experiments, where the imaging was done through the cranial window (Supplementary Fig. S14a).

## Electrophysiological recordings
Electrophysiological recordings were done with Neuropixels 1.0 Probes[55] in head-fixed awake mice freely running on a wheel. For each mouse, we performed four recordings on subsequent days. To later recover the probe position in each recording, the probes were painted with DiD cell labeling solution (Invitrogen V22887) before each recording. To record neural responses to motion cloud stimuli across all cortical layers, we targeted the center of V1 (4 mm posterior and 2.5 mm lateral from bregma) and inserted the probe perpendicular to the cortical surface (6 recordings in 3 mice) to a depth of 2 mm. To simultaneously record from V1 and SC, the probes were angled to 35° elevation and inserted to a depth of 3.6 mm. Neuropixels were positioned with a high-precision micromanipulator (uMp-4, Sensapex) at a speed of 10 μm/s. In each recording, the insertion point was moved by ~100 um in either mediolaterally or rostral-caudal direction from the target coordinate to ensure that fluorescent electrode tracts could be later related to the probe position in individual recording sessions (Supplementary Fig. S11a and S14b). To determine the surface of the SC, we took advantage of the visible gap in visual responses between the cortex and SC and determined the SC surface at the depth at which we could detect visual LFPs after this gap. In agreement with our histological results, this was usually at a depth of 1.63 ± 0.16 mm.

All visual stimuli were presented at a 17 cm distance from the right eye on a gamma-corrected LED-backlit LCD monitor (Viewsonic VX3276-2K-MHD-2, 32", 60 Hz refresh rate). Similar to the two-photon experiments, we then presented visual stimuli from the orientation band expanding and spatial frequency band expanding protocols (see 'Visual stimulation' section below). Data acquisition from the Neuropixels probe was done using SpikeGLX v3.6 (https://github.com/billkarsh/SpikeGLX, Bill Karsh). High-pass filtered signals were recorded from 384 channels at 30 kHz, and low-pass filtered signals were recorded at 2.5 kHz. The data was then time-shifted and median-subtracted with CatGT 3.9. Channels that were broken or outside the brain were detected using SpikeInterface[102] and removed from further analysis. The data was then spike-sorted with Kilosort 2.5 and manually curated using phy 2.0b1 (https://github.com/kwikteam/phy). The criteria for manual curation were consistent spike waveforms across at least three recording channels, stable spike amplitudes across the recording, and a visible gap in the center of the auto-correlogram. In addition, we imposed a cutoff for neurons with inter-spike-violations

**Table 1 | Specific Parameters for the orientation and spatial frequency band expanding protocols**

| | Central Orientation (°) | Orientation bandwidth (°) | Central Spatial Frequency (cpd) | Spatial Frequency bandwidth (cpd) |
|---|---|---|---|---|
| **Orientation band expanding** | 0 | 5, 25, 45 | 0.04, 0.16 | 0.004 |
| **Spatial frequency band expanding** | 0, 90 | 5 | 0.04 | 0.004, 0.04, 0.4 |

To investigate the effect of the orientation bandwidth band expanding (orientation band enrichment protocol, top row), the central orientation (first column) and spatial frequency bandwidth (last column) were kept constant (0° and 0.04cpd, respectively), while different combinations of orientation bandwidth (second column: 5°, 25°, 45°) and central spatial frequency (third column: 0.04 cpd and 0.16 cpd) were presented. This resulted in a total of 6 different stimulus parameter combinations. The same principle was used for the spatial frequency band expanding protocol (bottom row).

above 0.5, using the contamination rate method[103]. Sorted data were analyzed using custom MATLAB code.

After all electrophysiological recordings were complete, the mice were first perfused with cooled PBS and subsequently with 10% formalin. After storing the brains in sucrose solution with increasing concentration from 10-30%, 50 μm-wide brain slices were cut using a Cryostat (CM3050 S, Leica). Images of the brain slices were made using a fluorescent microscope (BZ x810, Keyence, Germany).

### Visual stimulation

Motion cloud stimuli were generated using the Motion Cloud Package[42] and adapted for display via custom MATLAB code (MATLAB R2019b, MathWorks). The package follows a linear generative model for creating a dense mixing of localized moving gratings with random positions based on predefined parameters. To ensure that neural responses were not strongly driven by the individual properties of these random phase motion textures, 10 renderings (300 frames each) of each stimulus were generated for each parameter combination. 20 repetitions of each rendering were then presented in a pseudorandom sequence movie where stimulus presentation order was shuffled with a restriction that prevents consecutive presentation of the same stimulus category. Motion cloud stimuli were either enriched in orientation bandwidth (narrow: 5°, mid: 22.5°, broad: 45°; around a central orientation of 0°; see also Fig. 1) or in spatial frequency bandwidth (narrow: 0.004 cpd, mid: 0.04 cpd, broad: 0.4 cpd; around a central spatial frequency of 0.04 cpd; see also Fig. 2). To test the discrimination of different central orientations or spatial frequencies (Figs. 6, 7) the central orientation was either 0° or 90° and the central spatial frequency either 0.04 cpd or 0.16 cpd. All motion clouds were presented on the screen, drifting from nasal to temporal (or orthogonal to the central orientation) with a temporal frequency of 1 Hz. For the orientation and spatial frequency bandwidth protocols, the stimulus duration was 5 seconds, followed by a 5 s inter-stimulus interval during which a mean gray screen was shown. An overview of all parameter combinations and protocols is shown in Table 1. To combine changes in both orientation and spatial frequency bandwidth (Figs. 4, 5; 'mixed' condition), we used motion cloud stimuli with a central spatial frequency of 0.04 cpd, a central orientation of 0° and a temporal frequency of 1 Hz, and then increased the bandwidth of either spatial frequency, orientation or both. The 'narrow' case consisted of 0.004 cpd spatial frequency bandwidth and 5° orientation bandwidth and served as the baseline for the narrow bandwidth of both features. In the 'SF' case, the spatial frequency was increased to 0.4 cpd while the orientation bandwidth remained at 5°. In the 'ORI' case, the spatial frequency bandwidth remained at 0.004 cpd, and the orientation bandwidth was increased to 45°. Lastly, in the 'mixed' condition, both spatial frequency and orientation bandwidth were increased to 0.4 cpd and 45°, respectively. For each stimulus condition, we generated eight separate stimulus renderings to reduce the risk of overemphasizing neural responses to complex features that could occur in a specific rendering. Each rendering had a total of 7 presentations as full-field stimulus (56 presentations for each full-field stimulus).

To test for the impact of center surround suppression, we first used sparse noise stimulation to identify the receptive field locations of the neurons in each recording session. The stimulus in each frame was composed of four non-adjacent 12° × 12° white squares over a black background[97]. The grouping of the squares was pseudo-randomized to ensure that each location would be presented 20 times in total. Each presentation lasted for 250 ms with an inter-stimulus interval of 750 ms. For each field of view, we then performed an imaging session with sparse noise stimuli to identify the receptive fields (RF) of all imaged neurons (see section 'Data analysis and statistics'). After the imaging session, we then created a new stimulus set where all motion clouds were covered by a gray mask, and a circular, 15° transparent aperture was placed in the center of the mean RF of the imaged population (Fig. 4b, 'Center RF,' see also Supplementary Fig. S6). In a second recording session, we then returned to the same field of view (based on the vessel pattern and visually comparing the location of the imaged cortical neurons) and repeated the sparse noise stimulation. Subsequently, we presented the full-field and center RF stimuli for the narrow, SF, ORI, and mixed conditions, as described above. For both the full-field and center RF conditions, we presented 56 stimuli, respectively, in pseudo-randomized order. The stimulus duration was 2 s, followed by a 4 s gray screen.

To test for orientation tuning (Fig. 4), we showed full-field drifting square wave gratings at five different orientations (−45°, 22.5°, 0°, 25°, 45°; 0.04 cpd spatial frequency; 1 Hz temporal frequency) in pseudo-randomized order. The stimulus duration was 1 s, followed by a 2 s inter-stimulus interval where a gray screen was shown, and each stimulus was repeated 20 times.

### Touchscreen chamber experiments

Behavioral experiments were performed in a custom-built operant conditioning touchscreen chamber, using a 10" screen for visual stimulation (Elo Touch 1002 L monitor, running at 60 Hz)[90]. All animals performed the experiments under a water restriction regime and received at least 1.5 ml of water per day to maintain their general well-being. They received water *ad libitum* on the weekends and had access to food *ad libitum* throughout the experiment. All mice were weighed daily and monitored before and during the behavioral experiments. In the first phases of pre-training, the animals were habituated to the touch screen chamber and learned to associate a green light with the availability of a reward. After the animals were able to collect at least one reward per minute, they proceeded to the next phase, where they had to first touch the screen to receive a reward, which was still indicated by a green light. Again, after the animals performed the sequence of touching the screen and collecting a reward at least once a minute, they were moved to a more specific training regime to perform either the orientation or spatial frequency discrimination task. All trained mice passed the initial criteria and could be trained on the discrimination tasks (see Supplementary Movie 5).

The *orientation discrimination task* started with the training for easy discrimination, where the animals had to discriminate a horizontal (target) from a vertical (distractor) sine-wave grating with a spatial frequency of 0.04 cpd. Each stimulus was shown through a

circular aperture with a diameter of 7.5 cm, and stimuli were presented left and right from the screen center, with a distance of 10 cm between the stimulus and the screen center. After the animals reached a performance of at least 80% correct responses, we proceeded with a staircase procedure. In this staircase procedure, the difficulty level was dynamic. Every correct response was followed by a decrease in orientation difference of 3°, while an incorrect response was followed by an increase in orientation difference of 8°, leading to a target performance of 72.7%. After the animals could perform the staircase procedure, we obtained one orientation discrimination threshold per session, revealing the minimal orientation difference between the target and distractor stimulus that was required for robust task performance. In randomized sessions, motion clouds with different spatial frequency bandwidths were used. Only one spatial frequency bandwidth was used per session to test how the discrimination threshold would be affected by a given spatial frequency bandwidth. To later compare the perceptual impact of different spatial frequency bandwidths, all trials with an orientation difference between 40° and 50° were combined and averaged over all sessions.

The *spatial frequency discrimination task* followed a similar training regime. After the animals performed sufficiently in the pre-training (Supplementary Fig. S15), they were trained to discriminate between two (0° central orientation) motion clouds of different spatial frequencies. Throughout pre-training, the animals had to discriminate different spatial frequency pairs (0.04 cpd vs 0.06, 0.12 or 0.16 cpd; Supplementary Fig. S15). The animals showed a stable discrimination between 0.04 and 0.16 cpd, after which different orientation bandwidths were introduced. The different orientation bandwidths were used within the same session because no adaptive staircase procedure was used in these experiments.

To ensure that animals did not develop an intrinsic side bias during the discrimination tasks, we used a bias correction routine[28]. In the initial ten trials, the position of the correct stimulus was randomized, and the bias correction was subsequently used to correct for direction and repetition bias. In case a bias was detected, the stimulus position was chosen to increase the probability of showing the target stimulus on the non-preferred side. If a repetition bias was detected, the probability of changing the target stimulus position from one trial to the next was increased. When no bias was detected, the position of the target and distractor stimulus was chosen randomly.

## Data analysis and statistics
All data analysis was done in MATLAB (MATLAB R2022a, MathWorks).

**Widefield.** When computing V1 responses in the widefield, we first determined the 50% region of V1 that responded the strongest across all presented stimuli per mouse. This was needed to prevent obtaining artificially lower response magnitudes in V1 due to integrating too many pixels from medial V1 that showed much weaker visual responses. For Fig. 5c, V1 activity was averaged over this subregion of V1, and individual trial responses were pooled across mice.

**Electrophysiology.** To quantify the electrophysiological spiking responses of individual neurons to broadband stimulation (Fig. 5f–h), we computed the area under the receiver-operator characteristic curve (AUC) between the baseline and the stimulus period. This allows us to obtain a measure of how reliably a given neuron responded to each stimulus, regardless of its overall firing rates. As with the two-photon data, we first computed the mean firing rate during the stimulus and a 1 s baseline period for each stimulus presentation. To identify stimulus-responsive neurons, we then used a one-sided Mann-Whitney-U test to assess if the mean firing rate across all stimulus presentations was significantly larger than zero (*p*-value < 0.05). For all stimulus-responsive neurons, we then computed the AUC between the baseline and stimulus periods to quantify how reliably they responded to

each stimulus condition. An AUC of 1 indicates a perfect separation of the activity in the baseline and the stimulus period, and an AUC of 0.5 suggests complete overlap (see also Supplementary Fig. S8).

*Two-photon imaging.* After each recording, two-photon imaging data were motion corrected, and individual neurons were isolated using the Suite2P package with model-based background subtraction[104]. Suite2p was used to perform rigid motion correction on the image stack, identify neurons, extract their fluorescence, and correct for neuropil contamination. ΔF/F traces for each neuron were then produced using the method of Jia et al.[105], skipping the final filtering step. To compute the stimulus-response amplitude (ΔF/F) of individual neurons in the two-photon data, we computed the median across all trials for the difference between the fluorescence during the first 2-3 seconds of the stimulus and a preceding 1 s-long baseline window for each stimulus presentation.

To identify neurons that were responsive to a specific visual stimulus, we used a one-sided Mann-Whitney-U test to assess if ΔF/F across all stimulus presentations was significantly larger than baseline (*p*-value < 0.05).

To identify the receptive field of neurons from the sparse noise paradigm, we also computed the ΔF/F for each neuron to all sparse noise stimuli that contained a white square in a specific part of the screen. This resulted in a set of neural responses for each 12° × 12° square on the screen, and we used a one-sided Mann-Whitney-U test to identify significant responses to visual stimulation on a specific part of the screen. For significantly responding neurons, we then inferred the center of the receptive field as the stimulus location that resulted in the strongest neural responses.

We used DeepLabCut[106] V2.3.9 to analyze the pupil position (see Supplementary Fig. S6). Eight reference points around the lids and one pint at the center of the pupil were labeled and tracked. We binned the pupil movement range area (4 pixels/bin) and then calculated the mean frequency at which the center was located within a bin during the stimulus period.

To compute the orientation tuning for different neurons, we computed the ΔF/F in response to different grating orientations and used a one-sided Mann-Whitney-U test to identify neurons that significantly responded to any grating stimulus. Besides this, we did a linear interpolation of the tuning curves and inferred the preferred grating orientation of responsive neurons as the orientation that induced the strongest neural response. Only neurons with a clear orientation tuning, i.e., a difference between peaks and troughs above the median of the peak and trough distribution of the whole population, were selected for the multiple regressor model (Fig. 4h, i).

**Statistics.** Throughout the results section, we have reported mean ± s.e.m. When reporting the normalized responding cell count for each stimulus per session, we normalized the cell count per session by the number of neurons that responded to the lowest respective bandwidth. We then pooled the cell counts from all recordings to test for significant differences in the number of responsive cells for different feature bandwidths. Since recordings from the same animals are not fully independent variables, we used a linear mixed-effects (LME) model (using the fitlme function in MATLAB) to determine significant differences between groups wherever possible (normally distributed data). Here, we included the stimulus and animal identity for the fixed and random effect coefficients, respectively. This allowed us to test if the stimulus identity significantly explains differences between two stimulus conditions while controlling for potential differences between animals. To test for significance, we performed a likelihood ratio test between a mixed-effects model that only contained animal identity versus a full mixed-effect model that contained both stimulus and animal identity. The reported p values and T statistics, therefore, indicate whether the addition of the stimulus regressor added significant information beyond the animal identity to explain differences

between the tested conditions. We used this approach to test for significant differences whenever samples (individual neurons or recordings) were pooled across multiple animals (Figs. 1e, 2e, 3b–e, 5c–h, 6b–d, f–h). In addition, for testing the effect of orientation tuning versus surround modulation in Fig. 4g–i, we upgraded the model to a multiple-regressor linear mixed effect model, whereby the orientation tuning, the center response, and the surround modulation effect were used as regressors to determine the impact of each regressor on the resulting changes.

The LME model formula was:

$$Rm_{ij} = \beta_0 + \beta_1 Sm_{ij} + \beta_2 Ot_{ij} + \beta_3 Cr_{ij} + u_j + \epsilon_{ij} \quad (1)$$

where $Rm_{ij}$ is the response modulation for the $i$th observation within the $j$th animal, $\beta_0$ is the fixed intercept (the total mean response modulation across all animals and conditions), $\beta_1$ $\beta_2$ and $\beta_3$ are the fixed effect coefficients for the predictors: Sm (Surround modulation), Ot (Orientation tuning) and Cr (Center response). $u_j$ is the random intercept for the $j$th animal, accounting for the variation between different animals, and $\epsilon_{ij}$ is the residual error term for the $i$th observation within the $j$th animal, which accounts for the variability not explained by the fixed effects or the random intercepts.

In cases where using the LME model for statistics was not possible due to the data being not normally distributed, such as for ΔF/F amplitudes of pooled cells (Figs. 1f–h, 2f–h, 4d–f, 7c–f), we used a Wilcoxon signed rank test and additionally did a mouse-wise comparison in Supplementary Fig. S1.

If multiple comparisons were performed, for example, when comparing the number of responsive cells in the mid and broad-bandwidth to the narrow-band condition (Fig. 1e), we performed a Bonferroni-correction for the number of performed tests to adjust for a multiple-comparison bias. Here, we adjusted the α level to reach significance to $\alpha = 0.05 / n$, where n was the number of performed tests. In the above example, two tests were performed, so a significant difference required a $p$-value below $\alpha = 0.05 / n = 0.025$.

To further test if individual neurons selectively responded to the narrow-, mid-, or broad-range of either the orientation or spatial frequency bandwidth (Supplementary Fig. S1d, h), we computed a bandwidth selectivity index BW_SI that was computed as the response difference to a given bandwidth (e.g., narrow-range, $Rn$) and the mean of the other two (e.g., mid- and broad-range, $Rm$ and $Rb$), normalized by their sum.

$$BW\_SI = \frac{Rn - \text{mean}(Rm, Rb)}{Rn + \text{mean}(Rm, Rb)} \quad (2)$$

The BW_SI is bounded between − 1 and 1, with values > 0 indicating selectivity for a given bandwidth, such as the narrow range, while negative values indicate selectively weaker responses to that bandwidth. To test if the BW_SI for a given neuron could not be observed by chance, we performed a shuffle control. Here, we shuffled the labels for three bandwidth conditions and then re-computed the BW_SI to obtain a measure of possible BW_SI values that can occur due to random response fluctuations. For each neuron, we computed 1000 shuffled BW_SIs to obtain a random distribution and then tested if the real BW-SI was above the 95th percentile of the shuffle distribution. Neurons for which this was the case were counted as selectively responding to narrow, mid, or broad-range stimuli (outer paddle of the Euler diagrams in Supplementary Fig. S1d, h). Neurons with negative BW_SI below the 5th percentile were counted as being mixed selective for the two bandwidths aside from the non-preferred bandwidth. For example, a neuron with a negative BW_SI below the shuffle control for the narrow-range bandwidth was counted as being selective for both the mid- and broad-range bandwidths (overlapping paddles of the Euler diagrams in Fig. 1d, h). Lastly, neurons without any selectivity beyond

the shuffle control were counted as non-bandwidth-selective (the center region of the Euler diagrams in Supplementary Fig. S1d, h).

Using the same approach, we computed an orientation modulation index OMI to quantify the difference in neural responses to broad versus narrow feature bandwidth (Fig. 6c, g).

$$OMI = \frac{R_{broad} - R_{narrow}}{R_{broad} + R_{narrow}} \quad (3)$$

where $R_{broad}$ and $R_{narrow}$ are mean responses to broad- and narrowband stimuli, respectively. The OMI is bounded between −1 and 1, with positive values indicating increased responses to broadband stimulation and negative values indicating reduced responses to broadband stimulation.

To determine the ability of each neuron to discriminate between the two different central spatial frequencies or between two central orientations, we also computed the AUC between these stimulus conditions (Fig. 6c, g). We computed the median response amplitude to all stimulus presentations for a given central feature (0.04 versus 0.16 cpd for spatial frequency and 0° versus 90° for orientation) and computed the AUC as a normalized measure of response discriminability. Here, we used the absolute AUC as a measure of discriminability, regardless of the response preference for a given stimulus feature for a given neuron. This was done by computing the absolute AUC as

$$AUC_{abs} = |AUC - 0.5|*2 \quad (4)$$

$AUC_{abs}$ is bound between 0 and 1, with positive values indicating increased discriminability of neural responses between the two stimulus conditions. This is referred to as spatial frequency or orientation discrimination index in the Results section.

### Visualization

All data figures were generated in MATLAB (MATLAB R2022a, MathWorks). Raincloud plots were generated with the Raincloud tool[107], and figures (Figs. 1c, 2c, 3a, 4a, d, 5a, 7a, Supplementary Figs. S4c, d, g, h, S9a) were created in BioRender[108]. For better visualization of the raincloud plots, outlier elements above or below3 median absolute deviations were excluded from the plots using the *rmoutliers* function in MATLAB. All values were included in the statistical analyses. Stars mark p-value ranges, whereby one to three stars respectively signify p-values smaller than 0.05, 0.01, and 0.001. If multiple tests were performed, significance thresholds were corrected for the number of comparisons (e.g., 0.05/n for one star).

### Statistical analysis of natural images and motion cloud stimuli

**Orientation distribution analysis.** The ImageJ[109] Directionality[110] plugin was used on a library[47] of nature photos (Fig. 1a, $n = 1077$ images) or a complete rendering of the motion cloud stimuli (Fig. 1b, $n = 300$ frames) to derive the distribution of directional changes in color intensity within a local gradient (5 × 5 Sobel filter is applied). Each image was loaded individually, and 180 bins of 1° each (range 1° – 180°) were selected. For each image, a data table of measured orientation contents in the given bin of orientations was saved. Finally, we calculated the mean and s.e.m. across all the individual images and plotted the respective curves (a light Gaussian fit, $\sigma = 2$ was applied) via custom MATLAB (MATLAB R2022a, MathWorks) scripts.

**Spatial frequency spectrum analysis.** We analyzed the distribution of spatial frequencies for a set of images from either a library of nature photos[47] (Fig. 2a, $n = 1077$ images) or a complete rendering of the motion cloud stimuli (Fig. 2b, $n = 300$ frames). Here, we computed the 2D FFT spectrum for each image individually. To describe the spatial frequency content of images independent of their orientation, we

shifted the zero-frequency component to the center and averaged energies in the frequency domain depending on their Euclidian distance to the zero frequency. Units of the spatial frequency spectra were converted from cycles per image to cpd based on their presentation or acquisition. For motion cloud stimuli, we computed their angular size based on their presentation:

$$angular\ size = 2\ x\ arctan\left(\frac{monitor\ width}{2\ x\ distance\ to\ the\ eye}\right) \quad (5)$$

For the natural images, the angular size was derived based on the individual camera field angle of view of each album:

$$FOV = 2\ x\ arctan\left(\frac{sensor\ dimension}{2\ x\ crop\ factor\ x\ focal\ length}\right) \quad (6)$$

Finally, we calculated the mean and s.e.m. across all individual images and plotted the respective spectra.

**Stimulus predictability analysis.** Spatial autocorrelations were computed for individual stimulus frames using the analysis provided by (Portilla and Simoncelli, 2000)[16]. For the use of the Portilla-Simoncelli (PS) analysis, stimulus frames were cropped to a size of 256 by 512 pixels and spatially down-sampled by a factor of 2. Raw coefficient correlations presented in Fig. 4c were computed as the central sample of the real autocorrelation for a spatial neighborhood of 51 pixels. Maps of spatial autocorrelations were averaged over all frames in the stimulus library of a given condition and presented with the corresponding colormaps.

**Stimulus separability.** To estimate how variable different realizations of the same stimulus condition are, we performed a principal component analysis across all texture parameters generated by the PS analysis (including pixelStats, pixelLPStats, autoCorrReal, autoCorrMag, magMeans, cousinMagCorr, parentMagCorr, cousinRealCorr, parentRealCorr, varianceHPR). We computed the principal components for individual frames and then averaged coefficients across frames within individual stimulus realizations. This resulted in average coefficients for each of the eight realizations per stimulus condition, of which we present a projection onto the first two principal components (Supplementary Fig. S7a).

**Coefficient magnitude statistics.** We investigated if expanding orientation- and SF bandwidths result in the emergence of higher-order structures in the stimuli by computing the cross-orientation- and cross-scale energies[16,18] (Supplementary Fig. S7b, c). For this purpose, we computed the PS analysis using four orientations over six spatial scales for the steerable image pyramid decomposition. Cross-orientation energies are based on the "cousinMagCorr" parameter of the PS analysis, representing the correlations between sub-band magnitudes of different orientations within the same spatial scale[16]. Here, we computed the average cross-orientation energy on the coarsest spatial scale between the 0° and the ±45° sub-band magnitudes, as these represent the dominant features in our stimuli. Cross-scale energies, given by the "parentMagCorr" parameter of the PS analysis, represent correlations of sub-band magnitudes with those of all orientations at the next coarser spatial scale[16]. Here, we focused on the cross-scale energy of the 0° magnitude sub-band of the steerable image pyramid between the two coarsest spatial scales. For both cross-energy measures, we tested for statistical differences between the stimulus conditions, using pair-wise Wilcoxon–Mann–Whitney U tests with subsequent Bonferroni correction, comparing energies across stimulus realizations. For this, we averaged cross energies over frames belonging to the same stimulus realization. For visualization purposes, we displayed the cross energies of individual stimulus frames after removing outliers.

## Model for neural responses of V1 neurons

To explore if the increased neural responses and recruitment for larger orientation bandwidth stimuli can be explained by different orientation tuning of V1 neurons, we created a simple bank of Gabor filters model. To predict neuronal responses, we compute the response magnitude of individual Gabor filters using the imgaborfilt-function in MATLAB (MATLAB R2022a, MathWorks). The response magnitude of complex Gabor filter kernels represents the response of an ideal-phase filter; the model, therefore, assumes a uniform distribution of phase representation in V1. We also designed the bank of Gabors to match the known tuning properties of mouse V1. For this, all Gabor filters were generated with a spatial frequency bandwidth of 2.38 octaves, matching previous reports[38,50]. Further, we adjusted the spatial aspect ratio of the Gabor filters, determining the envelope ellipticity of the filter kernels, thereby affecting the orientation tuning width of a given filter. We estimated this parameter for Gabor filters with a spatial frequency of 0.04 cpd, presented with a corresponding full contrast, 0.04 cpd sine wave grating. To match the tuning width of ~ 22° half-width half-maximum previously reported for mouse V1 neurons[38,50], we used Gabor filters with a spatial aspect ratio of 0.55 (Supplementary Fig. S3b). This represents an elongation of the supporting envelope in the direction of the parallel stripes of the filter. Furthermore, to capture a broad range of responses, we used Gabor filters tuned to 5 different spatial frequencies (0.01 cpd, 0.02 cpd, 0.04 cpd, 0.06 cpd, 0.08 cpd; Supplementary Fig. S3c) within the predominantly represented range of spatial frequencies of mouse V1[2]. For each spatial frequency, we used filter kernels with 180 different orientations in 1° increments, resulting in a bank of 900 individual complex Gabor filter kernels that covered the full range of orientations. To compare and combine responses across Gabor filters with different spatial frequencies, we normalized Gabor filter responses to their optimal stimulus (a full contrast sine-wave grating of matching spatial frequency and orientation).

We then simulated V1 population responses to our motion cloud stimuli. To reduce the computational cost, we spatially downsampled stimulus frames by a factor of 4 and computed Gabor filter responses for each individual "location" within the central 30° by 30° patch (Supplementary Fig. S4) around the stimulus center (37 × 37 sites in total) in the following denoted as such:

$$m(o, f, b, x, y)\,|\,o \in O, f \in F, b \in B, x \in X, y \in Y$$

$$O = \left(\{-45°, -22.5°, 0°, 22.5°, 45°\}; n_O = 5\right)$$

$$F = \left(\{0.01\,cpd, 0.02\,cpd, 0.04\,cpd, 0.06\,cpd, 0.08\,cpd\}; n_F = 5\right)$$

$$B = \left(\{narrow\ orientation\ bandwidth, broad\ orientation\ bandwidth\}; n_B = 2\right)$$

$$X = Y = \{1, 2, \ldots, 37\} \quad (7)$$

Here, m represents the response magnitude of a Gabor filter with orientation tuning o and spatial frequency tuning f in response to a stimulus with orientation bandwidth b at the location (x, y) within the center of the stimulus frame. x and y denote indices of the 37 × 37 grid of locations corresponding to the 30° × 30° patch around the stimulus center. Subsequent model fits were generated based on the response at the stimulus center, either on its own or modulated by the average surrounding filter responses (Fig. 3 and Supplementary Figs. S3, and S4). The Gabor responses at the stimulus center were obtained using the center position and averaging over the 5 Gabors with different

spatial frequency tuning:

$$c(o, b) = \frac{1}{n_F} \sum_{f \epsilon F}^{n_F} m(o, f, b, x_c, y_c) \qquad (8)$$

To translate these normalized Gabor filter response magnitudes into ΔF/F values that matched the measured neuronal response amplitudes, we computed a scaling factor $w_t$ (Supplementary Fig. S4). To determine this factor, we first measured the median responses of V1 neurons with different orientation tuning to narrow and ORI stimuli (− 45 °, − 22.5°, 0°, 22.5°, 45°; Fig. 3b, c) and then calculated the mean across all bins to equally weigh responses from differently tuned neurons. The average neural response amplitude was 1.622% ΔF/F. Second, we selected response magnitudes from Gabor filters with matching orientation tuning contained in set O and averaged responses across filters with these five orientation tunings, five spatial frequencies, and across stimulus conditions. This yielded an average normalized Gabor response magnitude of 0.1627 arbitrary units. From these two measures, we then derived the scaling factor:

$$w_t = \frac{1}{n_O n_B} \times \frac{\sum_{b \in B}^{n_B} \left( \sum_{o \in O}^{n_O} a(o, b) \right)}{\sum_{b \in B}^{n_B} \left( \sum_{o \in O}^{n_O} c(o, b) \right)} = \frac{0.01622 \, \Delta F/F}{0.1627 \, AU.} = 0.0997 \qquad (9)$$

Here, $w_t$ represents the scaling factor from Gabor response magnitudes to match the measured neural responses to different stimulus conditions based on their orientation tuning (a (o, b); analogous to c (o, b)) in units of ΔF/F (Fig. 3b, c). We termed these scaled Gabor responses the "Tuning only model" as they are solely based on the scaled, orientation-dependent Gabor filter responses at the central stimulus location (Fig. 3b, c and Supplementary Fig. S4). The tuning-only model responses were computed as follows:

$$tuning \ only \ responses(o, b) = w_t \times c(o, b) \qquad (10)$$

The "Tuning only model" fit showed the strongest deviation from the experimental data in the narrow condition. Since narrow stimuli also have a more regular spatial structure compared to ORI stimuli (Fig. 4c), we hypothesized that the difference between the modeled and experimental data could be explained by surround modulation. We, therefore, included an additional interaction between the center and spatially surrounding Gabor responses within the 30° by 30° patch into the model. This "Suppression model" used divisive surround suppression to normalize center responses:

$$suppression \ model \ responses(o, b) = \frac{1}{n_F} \sum_{f \epsilon F}^{n_F} \frac{w_c \times m(o, f, b, x_c, y_c)}{1 + w_s \times \max(0, s(o, f, b) + t)} \qquad (11)$$

Here, $w_c$, $w_s$, and t represent fitted model parameters, and s was defined as the average response magnitude of Gabor filters with the same orientation and spatial frequency tuning as in the center but from the surround region.

$$s(o, f, b) = \frac{1}{|P|} \sum_{x, y \epsilon P}^{n = |P|} m(o, f, b, x, y) \qquad (12)$$

With:

$$P = \left\{ (x, y) | x \in X, y \in Y \ and \ d(x, y) > \lim_i and \ d(x, y) < \lim_o \right\}$$

$$d(x, y) = \sqrt{(x - x_c)^2 + (y - y_c)^2} \qquad (13)$$

Here, P defines the set of "valid" 2D (x,y) coordinates within the 37 × 37 grid of Gabor responses that define the surround region. |P| denotes the cardinality of the set P. P depends on the model parameters $\lim_i$ and $\lim_o$, which define the minimum and maximum distance from the center Gabor response to qualify as part of the surround (converted from degrees to corresponding units).

We then fitted the suppression model using these five parameters: the minimum and maximum distance from the center at which responses were considered as contributing to the surround modulation ($\lim_i$, $\lim_o$), the weight of the center and surround responses ($w_c$, $w_s$) and the surround response threshold (t). This surround suppression acted on the level of individual frame responses and was fit to the response over the entire stimulus dataset using the fminsearchbnd-function (John D'Errico, 2006; Matlab file-exchange). This allowed us to limit the surround region from 7.5° to 15° displacement from the central Gabor, as well as limit $w_c$ and $w_s$ to positive values. For the presented model, we used the parameters of the best fit over 100 individual model fits with random parameter initializations. The presented, best fitting model used the following parameters: $\lim_i = 11.19°$, $\lim_o = 12.19°$, $w_c = 11.9$, $w_s = 1295.1$ and $t = − 0.2$ arbitrary units. The resulting orientation-tuning dependent fit is shown in Fig. 3b, c. From these fitted narrow and ORI responses, we then derived the "response modulation" shown in Fig. 3d.

To generate a fit for the expected recruitment, based on the "tuning only model" as well as the "suppression model", we used a simple linear regression without an intercept term to translate the fitted ORI response modulation to orientation-dependent neuronal recruitment (Fig. 3e). Our underlying assumption was that a higher response amplitude should be reflected in a higher fraction of neurons reaching the response criterion, resulting in larger recruitment. The fitted recruitment is derived from the Gabor response magnitude:

$$recruitment \ modulation(o) = \beta \times response \ modulation(o) \qquad (14)$$

The fit for translating response to recruitment modulation was $\beta_{tuning \ only \ model} = 0.134$ for the Tuning only model and $\beta_{suppression \ model} = 1.141$ for the Suppression model. Lastly, we computed the mean-corrected coefficient of determination ($R^2$) to evaluate how well our models described orientation-dependent differences in the observed broadband response and recruitment modulation.

### Reporting summary
Further information on research design is available in the Nature Portfolio Reporting Summary linked to this article.

## Data availability
The processed data have been deposited in Zenodo [https://doi.org/10.5281/zenodo.14605878]. The natural image library[47] used for natural feature distribution analysis is accessible at [http://tofu.psych.upenn.edu/~upennidb]. Source data are provided in this paper.

## Code availability
Image analysis, data analysis as well as the Gabor filter bank models can be found in GitHub [https://github.com/AachenBrainLab/Eyes_Wide_Bandwidth].

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

## Acknowledgements

We would like to thank Laurent Perrinet, Hugo Ladret, Ivo Vanzetta, and Frédéric Chavanne for helpful insights on the motion cloud stimuli and fruitful discussions on the project and manuscript; Magdalena Robacha for help with the mouse preparation and imaging, Fatemeh Yousefi for help with the pupil tracking. This work was supported by the Deutsche Forschungsgemeinschaft (DFG, German Research Foundation) - 368482240/GRK2416 (B.K.), 424556709/GRK2610 (B.K.) and 430156848/SPP2205 (B.K.), the Helmholtz association VH-NG–1611 (S.M.) and the state of North Rhine-Westphalia through the iBehave initiative (B.K., S.M.).

## Author contributions

B.M.K., C.W., and E.B. conceived and planned the experiments. E.B. designed the two-photon experiments, collected the neural data, and performed the analysis with input from B.M.K. and S.M. G.N. generated the stimuli, collected and analyzed the intrinsic- and widefield imaging data, and built the model of V1. C.W. and J.L. designed the 2AFC task, collected the behavioral data, and performed the analysis. S.G. and S.M. collected and analyzed the electrophysiological data. E.B. wrote the initial draft and generated the figures with input from B.M.K and S.M. The manuscript was revised and completed by B.M.K and S.M. All authors read and approved the final manuscript.

## Funding

## Competing interests

The authors declare no competing interests.
