## [Transparent Peer Review file · Nature Communications]

Broadband visual stimuli improve neuronal representation and sensory perception

Corresponding Author: Professor Bjoern Kampa

Version 0:

Reviewer comments:

Reviewer #1

(Remarks to the Author)

Summary

The authors employ visual stimuli of medium complexity, i.e., containing a larger spectrum of orientation and spatial frequency features than commonly used stimuli, to test neural response statistics in mouse primary visual cortex. They find that stimuli made up of a broader range of orientations drive more neurons and increase neural response amplitude, while stimuli made up of a broader range of spatial frequencies did not result in an overall change in neural responses. V1 population responses to different spatial frequencies could be discriminated better when stimuli contained a larger range of orientations. Similarly, mice could discriminate spatial frequencies better in broadband orientation stimuli.

Major comments

Overall, the study reveals new insights into how V1 represents visual stimuli and links these insights to behavioral performance. In particular, the increased neural and behavioral discrimination with broadband orientation stimuli are intriguing. However, we feel that the study would greatly benefit from a more detailed comparison between single neuron responses to the motion cloud stimuli versus the classic grating stimuli (more details below). Secondly, it would help the reader if the authors go into more detail comparing their study to other studies using similar approaches or having similar aims to reveal their unique contribution better. In our opinion, an attempt to bridge the insights from using the classic stimuli versus the more complex stimuli may help in that respect.

More detailed comments

1. A number of studies used similar approaches or had similar aims. Here are just some examples, some of which the authors already cite:

Walker EY, Sinz FH, Cobos E, Muhammad T, Froudarakis E, Fahey PG, Ecker AS, Reimer J, Pitkow X, Tolias AS. 2019. Inception loops discover what excites neurons most using deep predictive models. *Nature Neuroscience* 22:2060–2065. doi:10.1038/s41593-019-0517-x

Yu Y, Stirman JN, Dorsett CR, Smith SL. 2022. Selective representations of texture and motion in mouse higher visual areas. *Current Biology* 32:2810-2820.e5. doi:10.1016/j.cub.2022.04.091

Han X, Vermaercke B, Bonin V. 2022. Diversity of spatiotemporal coding reveals specialized visual processing streams in the mouse cortex. *Nat Commun* 13:3249. doi:10.1038/s41467-022-29656-z

Bolaños F, Orlandi JG, Aoki R, Jagadeesh AV, Gardner JL, Benucci A. 2022. Efficient coding of natural images in the mouse visual cortex. *bioRxiv* 2022.09.14.507893. doi:10.1101/2022.09.14.507893

It would help to put the current study into the context of the findings of these (and possibly other) publications. What were the insights from those studies, what is the contribution of the current study?

Instead it is not clear how relevant the comparison to the human study (1.297) is, given that it is about speed perception, which is not tested here.

2. It would help to have a more detailed description of motion cloud stimuli. Maybe a simplified description of how they are generated. From the results section alone, it wasn't clear that the stimuli are moving. Although this is not an important feature for the study, it seems like important information for the reader.

3. A more detailed description of how AUCs were used to determine responsive neurons would help. Even better to include an example curve.

4. Figure 1.

f, h, i. It would help to see how single neuron responses change across conditions, i.e., how many increase responses, how

many decrease responses, and by how much? You could introduce a change index, like $(R1-R2)/(R1+R2)$ where R1 and R2 are responses to 2 conditions, then do the same for R1 and R3.

g. The diagram doesn't help to visualize quantities. Is there a way to match area to percentages? (same for Fig 2g).

h. Figure caption does not match statement in text (l. 119).

5. The use of the term "tuning" is confusing. Presumably, the authors mean tuning to specific orientations or spatial frequencies but that is not obvious. E.g., l. 116: "Instead of an overall recruitment of neurons with narrow tuning, ... tuning is ... selective ...". What is the difference between narrow and selective tuning? Or l. 317: "... neural tuning cannot explain the response preference to broadband orientation stimuli."

6. Figure 2.

d. Do the calcium traces show responses to different repetitions of the exact same stimulus, or to different stimuli of the same class? Single trial or averages?

7. Regarding the specificity for certain bandwidth stimuli (e.g., l. 174), it seems possible that it is caused by the specific instances of 10 stimuli of the same class, rather than the general stimulus class. It would help to see response variation of single cells to stimuli of the same class compared to stimuli from different classes.

8. To get a better understanding of how the responses to the broadband stimuli are related to responses to more classic gratings, preferences for the broadband stimuli should be compared to orientation/spatial frequency preferences and tuning width of single neurons. E.g., it may be expected that the broadband orientation stimulus (with center orientation of 0 degrees) drives more neurons with preferences away from 0 degrees compared to the more narrowband orientation stimulus. As mentioned earlier, a detailed understanding of this relationship could really boost the impact of this paper. It may be one step towards showing in how far responses to the motion cloud stimuli can be predicted from responses to classic stimuli.

9. L. 326: It is not clear what the authors mean by "spatial consistency" and how it is related to stimulus predictability. The argument in the following sentence (l. 329) then seems very weak as the relation to predictability is unclear, and it is not known which aspect actually improves stimulus representation.

10. To understand the qualitative difference in results between orientation and spatial frequency broadband stimuli, it may be useful to quantify further stimulus statistics. When one compares examples between Figure 1b and Figure 2b, our impression is that stimulus texture changes in Figure 1 but not Figure 2. To determine whether other image statistics change, we suggest to quantify texture parameters as presented in:

Portilla, J., Simoncelli, E.P. A Parametric Texture Model Based on Joint Statistics of Complex Wavelet Coefficients. *International Journal of Computer Vision* 40, 49–70 (2000). <https://doi.org/10.1023/A:102655361998>,

In particular:

'Raw coefficient correlation' which represents the regularity of the image. This might help the point about spatial predictability.

'Coefficient magnitude statistics', which deal with structures in the image, and again might show a difference between the orientation broadband and the spatial broadband.

The suggested statistics might add and strengthen to the message of the paper. For examples of analysis refer to :

Okazawa, G., Tajima, S., & Komatsu, H. (2015). Image statistics underlying natural texture selectivity of neurons in macaque V4. *Proceedings of the National Academy of Sciences*. doi.org/10.1073/pnas.141514611

Bolaños F, Orlandi JG, Aoki R, Jagadeesh AV, Gardner JL, Benucci A. 2022. Efficient coding of natural images in the mouse visual cortex. *bioRxiv* 2022.09.14.507893. doi:10.1101/2022.09.14.507893

11. Unexpectedly, Fig. 3b,f seem to suggest that fewer neurons responded to the higher spatial frequency and the 90 deg orientation stimuli. Could the authors discuss why there is a difference (if significant)?

12. It would be useful to have a more detailed description of the temporal trial structure of the behavioral task (inter-trial interval, maximum time period between stimulus onset and response) and the "mechanics" of the task (how was the reward delivered, where was the reward delivery in relation to the screen, where exactly did the mouse have to touch the screen, etc.). Was the low spatial frequencies and the 0 degree orientation stimuli always the target or was the design balanced (The following statements may be related but were unclear: "In randomized sessions, different motion clouds with different spatial frequency bandwidths were used. Per session, there was only one spatial frequency bandwidth used to obtain the characteristics of the adaptive procedure."). What is the definition of "performance"?

13. Figure 4.

c,d. The absolute differences between performance levels in c and d look comparable. Is it possible that the higher number of samples in c underlie the significant outcome, in contrast to d?

f. How was the discrimination threshold determined? We assume that the psychometric curve was smoothed or fit?

Minor comments

14. L. 16 "improved feature discrimination performance": it is not clear whether this refers to neural or behavioral performance

15. L. 210: "broadband stimuli" should be changed to "orientation broadband stimuli" as this statement is not true for spatial frequency broadband stimuli. The statement in l.296 is similarly vague.

16. L. 395: "90 deg angle" is ambiguous. In relation to what? In which axis?

17. L. 440: Why is the p-value divided by 12 (rather than any other number)?

18. Minor typo: Line 234 should read custom-built

Reviewer #2

(Remarks to the Author)

This work explores the relationship between statistical properties of naturalistic visual stimuli (motion clouds), population-level responses in the mouse primary visual cortex and perceptual discrimination. There are two, related central claims of this article: i) increasing the bandwidth of stimulus orientation engages V1 population more strongly than increasing the

frequency bandwidth, and ii) perceptual discrimination improves with increasing orientation bandwidth and not with increasing frequency bandwidth.

The paper addresses an interesting and relevant question about the sensitivity of V1 populations and perception to higher order statistics of natural scenes. It simultaneously links neural coding and behavior - which is its definite strength. The writing is quite lucid, and I particularly appreciate very clear figures with a nice color scheme (blue-orientation, purple-frequency), which greatly helps parsing the results.

At the same time, I have some major concerns about the relevance of the study, its relationship to previously published results and the methodology. It is my opinion that these concerns should be addressed prior to the publication. I'm, however, not certain if they can be readily addressed at this stage of the project.

Major concerns:

1. The article neglects a large field devoted to developing parametrizable, naturalistic stimuli which interpolate between simple, artificial signals and fully natural images. To mention some examples: Portilla and Simoncelli 2000, Freeman et al. 2013, Hermundstad et al. 2014, Yu, Schmid and Victor. 2015, Clark, Fitzgerald et al, 2014, Banyai et al. 2019 (and many more). The motion-cloud approach is presented as a singular solution to stimulus design, which is not the case. It is therefore hard to judge which specific aspects of neural coding will be addressed by choosing this specific stimulus parametrization over others.

2. Related to the point above - overall it is not quite clear what is the central, specific motivation of the article beyond application of motion-clouds. The Introduction begins with a very broad claim that the visual system is adapted to stimulus statistics. It then switches the tone and starts to read almost like an accompanying study of the motion cloud paper [20], aiming to test sensitivity of V1 to properties of this particular stimulus (lines 50-51 and 59-61). It is true that V1 is adapted to stimulus statistics, but this has been a subject of research for a long time now. The article should be much more precise in stating what specific question it is addressing, except for exposing the animal to this specific class of stimuli, even matched in some sense to natural statistics.

3. A key claim is that ranges of motion cloud parameters used in experiments are matched to natural stimulus statistics. Such analysis is however not presented anywhere in the paper, and in my view this claim is weakly qualitative at best, and not substantiated at worst. Figures 1A and 2A present single images - this is not enough to justify the choice of parameters for the textures. Analyses should be performed on corpora of natural images (or one could use ranges reported in the literature, but this needs to be stated). Furthermore I was not able to find any description of how orientation distributions or spectra were computed to generate plots in Fig 1A and 2A.

4. The experiment studying sensitivity to frequency bandwidth is quite puzzling. The Author's chose to use a single, vertical orientation of the stimulus. Given that orientation is the primary factor for which V1 cells are selective, isn't that a trivial result? One would naturally expect that limiting orientation range decreases response strengths, since only a fraction of the population responds. In fact comparing distributions in Fig 2F to the leftmost distribution in Fig 1F seems to reflect that - is that right? One potential suggestion would be to use a "null model" of V1 population tuning i.e. a bank of Gabor filters (either parametric, or learned from natural image statistics with ICA or other algorithm). One could then simulate responses of such a filter bank to the same set of stimuli and compare to those obtained in the experiment. Any deviation of the data from such a null-model would be a signature of something non-trivial. Without analysis of this kind I find it hard to interpret the results.

Minor concerns:

1. I believe much more space should be devoted to explaining details of the analyses in the Methods section. I already mentioned lack of explanation of natural stimulus statistics analysis. Similarly no explanation is given how the area under the ROC curve (AUC) has been computed. What kind of classifier has been used? What was the range of threshold parameters? These details need to be included.

2. I strongly suggest including the psychometric curve for spatial frequency discrimination (as in Fig 4E). Only then one can have a full picture required to understand perceptual differences between conditions.

3. It is not technically correct to say that "edge orientations in natural scenes are not statistically independent from each other" (lines 28-29). As revealed by algorithms such as ICA or sparse coding, edge filters are approximately marginally independent. They become *conditionally* dependent in presence of high-order structures (e.g. long-range contours, textures etc).

Reviewer #3

(Remarks to the Author)

This manuscript describes an increase in responsivity of mouse upper layer primary visual cortical neurons when the orientation bandwidth of the stimuli is increased. Surprisingly, the same is not the case for increases in the spatial frequency bandwidth used.

Also surprising is the finding that some neurons appear to show bandpass tuning in the orientation bandwidth and spatial-frequency bandwidth domain, with some neurons responding preferentially to intermediate bandwidths.

The authors show convincingly that the observed increases in responses to broader orientation bandwidth stimuli also lead to better perceptual discriminability of these stimuli.

These results advance our understanding of primary visual cortical processing and should be of interest to visual- systems- and computational neuroscientists.

In general the results are solid and well supported by the data. However some points could perhaps have been strengthened by additional analysis or better experimental design.

The methods need to be more specific and lack details.

My main issues and suggestions, roughly in the order of decreasing importance are the following:

There are some missed opportunities:

I really like their idea of increased surround suppression for the far more predictable SF bandwidth stimuli to be the reason for the lack of responsivity increases for these conditions. But it would have been very nice and straightforward to actually show it by varying SF bandwidth on wider orientation bandwidth stimuli. This could either, according to their argumentation, increase responsivity also in the SF BW domain because of even lower predictability of these stimuli. Alternatively it could reveal that SF BW never influences responsivity of V1 populations.

One of the main points is that they find tuning to specific orientation and SF bandwidths.

Showing the percentages of cells responding relatively specifically to one or two bandwidths is good to see, however, this point of BW selectivity could perhaps be made stronger.

The authors could for example consider showing the distribution of some selectivity measure. For example $(r_s - r_n)/(r_s + r_n)$ with r_s being the response to the preferred orientation bandwidth (or SF BW) and r_n the average of the other two. This would allow judging the overall level of selectivity for particular bandwidths across the population much better than the percentages (1 and 2 g) which could come from very weak differences in dF/F. If there are many units with strong tuning to intermediate bandwidths, this would make that point stronger.

Similarly, with the data from supplemental figures 2 and 3, there might be an opportunity to strengthen this point:

The authors could directly relate the bandwidth tuning of each neuron in the two conditions. If, for example, a neuron prefers the intermediate bandwidth for both orientations, this would be a much stronger point for this neuron being tuned to a specific orientation bandwidth, independent of the actual orientation. This could also give hints to potential mechanisms for this sort of tuning of V1 cells.

To do this, the authors could, for example, simply calculate a correlation between the dF/F of each bandwidth condition in the two orientation conditions and show the distribution of the r-values. If many are high, this would point to true, strong bandwidth selectivity, independent of absolute orientation or SF. If, however, many neurons switch bandwidth-preference between the different orientations, e.g. low for vertical but broad for horizontal, this would argue more for the simpler recruitment theory. All of this would of course also be helped by a broader range of bandwidths being used.

Line 176ff: The entire attempt to directly compare tuning widths between spatial frequency and orientation appears flawed. I understand this attempts to show the tuning widths between orientation and SF are not different and that the differences in increased responsiveness are not due to broader tuning to SF. However,

1. it is entirely unclear how the five SFs and five orientations and especially the step size were chosen for this control experiment and how they relate to the bandwidth experiment. These choices seem somewhat arbitrary and they are not as line 177 states the orientations and SFs they used in the motion cloud stimuli. They are an arbitrary subsample of them.
2. Importantly, there is no correspondence between an octave different spatial frequency and a 22.5° orientation shift. These are simply parameter choices. SF and orientation are entirely different measures and cannot be compared directly to say the tuning to one is "broader" than the other (regarding Figure S1).

It is also unclear what the Wilcoxon signed rank test compares. The tuning width for each unit? What is n? Also, if they are different, why does line 180 and the discussion state they were similar?

There is great value in testing the orientation and SF tuning of each unit. Especially if they would be the same units measured for the bandwidth experiment (it is unclear if they are). That way there could be a direct comparison of each unit's tuning to bandwidth stimuli, as well as their tuning width for orientation and spatial frequency. However the experiment as is, especially if performed on different units/sessions, does not prove or disprove "broader" tuning for orientation and thus their argument of broader SF tuning not being the reason for the absent increase in responsive units for broader SF bandwidths is futile.

The methods need some work. A lot of details are missing. One important thing would be to know whether the spatial frequency and orientation bandwidth tuning was performed in the same nine mice or whether this was separate cohorts of mice. Was it the same sessions or different sessions? In other words, do we know both the SF and orientation bandwidth tuning for the same set of cells or not? Why is there different numbers of sessions for the two stimulus sets - 16 and 18 sessions?

The description of the visual stimuli is lacking the size of the stimulus/screen. Was the monitor run at 30Hz? 150 frames for 5 seconds?

For the touchscreen task, there is no mention of how the patterns were arranged, at what size they were presented, were there gray ISIs, how was the task timed, was the position randomised, what was the make and model of the screen, what its size?

Why were the target SFs and orientations not randomized but the same for all mice? Did all mice pass the initial criteria? Perhaps a video would help to imagine what the stimuli actually looked like, including the temporal component. If the TF is always 1Hz, then the different spatial frequencies would all have different speeds. Or was the pattern kept fixed and simply moved at 1Hz of the basis SF?

The figures are aesthetically very nice! However the single dots in the lightest purple are very low contrast and fairly hard to see (especially figure 4d and f). On the spectra (1&2 a&b) some x-ticks would have been nice. Especially at the spatial frequencies tested in S1. The example images in figure 3 are too small. You have to zoom to see anything and in the printed version this will only look gray.

Dyballa/Stryker PNAS 2018 has a similar point to more complex stimuli showing a more varied response landscape in V1 and mixed and non-linear selectivity of neurons. This paper should probably be discussed.

Figure 1i shows responses well up to 16-17dF/F, but these responses are somehow not plotted in 1f, where responses max out at about 8dF/F. Which values are actually plotted and which omitted and is 1h and i not just a subset of the cells in f? Same applies to figure 2.

line 65 increasing frequency bandwidth only weakly affected neural responses - this is not quite true, no? It did affect single cells quite dramatically. Just the overall responsivity wasn't enhanced as in the case for orientation bandwidth increases.

line 66 it should say "spatial frequency" and not simply "frequency". Twice this line!

line 69: "narrow gratings" could be misinterpreted

line 70: maybe should say "more closely match", as these stimuli definitely also do not fully match the statistics of natural sensory inputs.

line 83 it is unclear here how AUC determines which neurons respond, so the shuffling procedure should be mentioned here.

line 87: should just say "neurons" as it is never tested whether these neurons are orientation tuned.

Figure 1b it is a bit confusing that 0° in a) seems to be horizontal but vertical in b)

While discussing Figure 1 it would be good to also know about the motion component of the motion cloud stimuli. Is it drifting? homogeneously? drifting left or right? This should be clear and not have to be looked up in the methods (where it is also not reported except for the 1Hz).

line 85/86 and 140/141, 195 it might be better to report the percentage of visually responsive units per session and not absolute numbers which will depend on FOV, expression level, etc. and thus be better comparable from animal to animal.

There seems to be a problem with the responsive neurons masks in 1d. I am assuming they should all be solid and of the same color per panel as in figure 2. Instead multiple different shades of blue show up in at least the middle panel and the masks are not solid but sometimes just an outline or parts of the cell body are coloured. Or do different levels of blue signify a difference?

111ff with increased neural responsiveness here, do they mean increased number of responsive cells or larger response amplitude (or both)? The wording implies more increased amplitude, but the argumentation following sounds more like increased number of cells.

Perhaps some bootstrap/shuffling procedure would be good, to show that these distributions of overlapping responsiveness are not to be expected by chance alone.

117 they only test for bandwidth tuning and not orientation tuning so this should just read "tuning", without orientation.

136 to say that the two distributions in 2a and b are matching is a vast overstatement. They are more similar than the narrow perhaps, but far from matching. Perhaps the poorer match of the SF spectrum as compared to the orientation bandwidth distribution should be discussed.

Which spatial frequency do they use to increase the bandwidth from? Is it 0.04 as in the table in the methods? In figure 2b it looks as if the center is 0 cpd (but hard to tell without x-ticks). Or is this the difference to 0.04? Why then is it one-sided?

figure 1c and 2c, the stimulus does not appear vertical but slanted. But the text states it was vertical.

Is the percent responsiveness for 1 and 2d similar? Or in other words, how do all the normalized to 1 values from 1 and 2e compare? It appears to respond less to the SF BW variant. Is it because orientation bandwidth is so narrow for these stimuli? This also relates to the earlier point of perhaps testing whether altering SF on more broad band orientation stimuli would change the lack of increase with broader distributions?

186 specify orientation bandwidth

234 and 404 should be custom-built

why is the target-performance for the staircase 72.7% correct?

293ff it should be made clear already here that only increasing orientation bandwidths increased responses while the other had no effect on response strength or responsive population. Again in 296: only enrichment of orientation bandwidth. This distinction is discussed, but only at the end of the next paragraph.

304 should be "revealed"
305 increased sensory perception - rephrase
375 positioned and? attached
379 "with" missing
389 should be "driven" not "drive"
391 should be "were" not "when"
395 90° from what angle?
408 should be "weighed daily"
425 should be "sufficiently"
426 "to" should be "two"

Supplement

figure 1a the y axis should probably be reversed
line 8: n = XX should be a number
line 13 should be "signed rank test"

Reviewer #4

(Remarks to the Author)

The study by Balla et al. addresses a very interesting question that is surprisingly understudied in the mouse (despite huge literatures in other species) - what types of stimuli best drive V1 neurons? They use a motion cloud tuned to different parameter spaces and measure responses in V1 neurons via 2p imaging. While the question is of broad interest to the community, a number of methodological issues substantially reduce the interpretability of the findings (see below). The overall consequence is that the results do not present a compelling, novel set of insights into stimulus representations by visual cortex.

Major concerns:

1. The stimulus size is not specified in the methods, but appears to be whole-screen from Figure 1. V1 neurons exhibit robust surround suppression whose magnitude may be strongly dependent on the stimulus properties of the center and surround regions. Thus, it is impossible with the present data to understand which response properties are due directly to the features of the stimulus present in the classical receptive field of each cell versus the stimulation of its surround.
2. There is no description at all of the animal's position during imaging. Figure 1 suggests the animal is in a tube, though awake or anesthetized is not specified. If anesthetized, it is hard to know if the present results generalize to awake mice. If awake, there is no discussion of how behavioral state or arousal level is monitored and included in analyses. As arousal can have profound impact on visual responsiveness, it would be necessary at a minimum to ensure this variable did not change across or within sessions. Overall, it is hard to interpret the present data without this critical information.
3. The statistical analyses are highly problematic and appear (with only minimal description) as being carried out with a "by-cell" comparison. Even with "Bonferroni Correction" (exactly what is being corrected or what the magnitude of the correction is does not appear in the text), cells within a single animal are not independent variables and cannot be analyzed as such. Most simply, analyses should be carried out by-animal. However, a mixed-effects model where both animal and stimulus are considered could also be constructed. Again, without proper statistics, it is impossible to fully appreciate the results for which many effect sizes appear to be quite small.
4. Broadly speaking, GCaMP population imaging (6 for sure, but probably true for all) does not robustly or reliably capture small changes in firing rate (this has been clearly shown in recent Allen Institute publications). For many studies, this is only a minor issue. However, at present where the goal is to say something about the sensitivity of cells to stimulus features, it may be a major difficulty. For example, if GCaMP imaging only reports bursts of spikes, the data here may show that certain stimulus parameters increase bursting. However, much of the information in neuronal activity might be carried in the addition or subtraction of a single AP (particularly in sparsely firing Layer 2/3 cells). I think any claim of "coding principles" by these cells must include some amount of ground truth electrophysiology data.

Less major concerns:

5. Please report responsive cells as a fraction of total cells imaged, not only the absolute number.
6. There are no numeric data presented anywhere in the text (means, SEMs, quartiles, stat values, etc. should all be present in the Results text).
7. The volume and titer of virus injected and the time after injection for imaging sessions are not specified.

Version 1:

Reviewer comments:

Reviewer #1

(Remarks to the Author)

The current revision of the manuscript has greatly improved the impact of the study. The authors have placed their specific research question and their results in context of previous studies on similar topics. By using a model of V1 neural responses, by relating responses for complex (motion cloud) and simple (grating) stimuli, and by recording responses to small versus fullfield stimuli, the manuscript now provides much clearer insights into circuit mechanisms behind their findings. The additional electrophysiology data from V1 adds a strong confirmation of the two-photon data and provides new insights on response differences across layers of V1. Finally, the provided movies showing the stimuli are very helpful.

The authors have also added new data on higher visual areas (HVAs) and the superior colliculus (SC). Although we think that the data are of high quality, it is our impression that they distract from the main story of this manuscript, which focuses on V1 and provides a comprehensive narrative on how neurons in V1 respond to narrow versus broadband stimuli. The additional data on other visual areas is very limited in comparison and raised several questions (see Major comments). Although they provide an interesting incentive to study broadband responses in other visual areas, we would recommend to leave these to a separate publication and/or put the current data into Supplementary figures. However, we just wanted to share our impression. The decision should be made by the authors.

Major comments

- More details about the Gabor model need to be provided, together with a better explanation of Figure S4a. Please describe which Gabors contribute to the surround modulation of one cell. What is their size, their preferred orientation, their spatial frequency/spatial aspect ratio? A cartoon showing the spatial layout of all Gabors relative to the stimulus would be helpful as well.

Which data were used to fit the Gabor models (tuning only, and surround suppression)? Specifically for the fit mentioned in I. 1225?

L. 1198 and Fig S4b-e: does the spatial frequency of the response curves (title of plots) refer to the stimulus spatial frequency or the spatial frequency of the Gabors?

I. 1209: Is this statement reflected in the presented data/plots? Where?

- Figure S3: more details need to be added.

Do the curves in the 2nd row of plot b show the same curve as in a but for different spatial aspect ratios? If yes, please state this in the figure caption. Is the spatial frequency of the stimulus the same as in a?

Are response curves in c for the tuning only or the surround suppression model?

- Figure S4:

b,d: legends "narrow" versus "ori" are confusing. Better "narrow" versus "broad"

- Figure 3:

To better compare the model predictions (Fig S4b), which show response amplitudes depending on preferred orientation, it would be very useful to have a similar plot as 3b but showing response magnitude (y-axis). These are the data used to rescale the Gabor responses (I. 1204-06).

The model predictions in 3b (yellow versus red lines) look like they could be scaled versions of each other. We were wondering how dependent the scaling of these curves is on the model parameters. Are there model parameters that would lead to a downscaling of the tuning only model (yellow curve), which would lead to a better fit for the tuning only compared to the suppression model?

Could the authors add the model predictions for plots d and e? Did the authors expect any dependence of SF response modulation and recruitment modulation on orientation preference?

We assume that the preferred spatial frequencies of the neurons were not measured. However, if the data exist it would be very interesting to have plots and model predictions as in plots b and c but clustering neurons according to their preferred spatial frequency.

- Figure 4:

Again, to evaluate the suitability of the Gabor model, it would be very informative to include the model predictions for the data shown in plots d-f. Specifically, we would be interested in seeing whether the model predicts the population responses for the small stimuli (in e) showing reduced responses to SF and mixed stimuli.

Although the illustrations of model parameters in plot g are nice, we didn't find the plot too helpful to understand the LME. It would help to add the formula that was used. Also, the term "response enrichment" is not defined; it should probably read "response modulation".

- Effect of surround modulation: The authors state that they matched the small stimulus (without surround) to the RFs of the neurons. As surround modulation is highly dependent on the placement of the stimulus on the RF center, it is crucial that we can see the RFs of the neurons in relationship to the stimulus. A related point is that eye movements could easily move the neural RFs away from the small stimulus. Can the authors exclude this possibility?

Please, state the size of the center stimulus in the main text and compare it to the average size of the neural RFs.

- Currently, it is not clear what the widefield data in Fig 5 a-c add to the manuscript. The widefield imaging was introduced to study HVAs, but no quantitative data (like for V1 in Fig 5c) are presented. The authors state that (based Fig 5b) broadband stimuli decrease responses in HVAs, which seems surprising but is not further discussed.

In addition, it is surprising that the V1 activity ipsilateral to visual stimulation is comparable in size to the contralateral activity although the stimulus was only presented to one visual hemifield. On the other hand, broadband modulation is much smaller on the ipsilateral side. Can the authors explain/discuss this?

- The widefield and the LFP data (Fig 5c-e) show that responses to the narrow stimuli are smallest compared to other stimuli, which is different from the single-neuron data in Fig 4d. Furthermore, mixed stimuli evoke large activity in the widefield data (V1), but relatively low activity in the LFP data in superficial layers of V1, while for single-neuron data mixed stimuli evoked responses comparable to narrow and SF data. The single-neuron spike data then is very similar to the single-neuron 2P data, which is a great confirmation. However, we think that the above mentioned differences across datasets and recording modalities should be discussed.

- Spike sorting: please add the criteria that were used to sort the units and evaluate commonly used quality metrics of the chosen units (if these were not used for sorting).
- Regarding the spiking data from the SC, more information needs to be added. Specifically: where are the RFs of these neurons? How was the border between cortex and the SC determined and in which layers were the neurons recorded? Are there differences across the depth of the SC as observed in V1?

Minor comments

• Figure 1:

The use of color in panel h is misleading. In all other plots, color reflects the stimulus feature while in h color reflects neural tuning.

L. 156: "(a)" is missing

• L. 252-255: Sentence is difficult to understand. Consider shortening and separating comparison between spatial frequency and orientation results from narrow and broad-range selectivity. As there are many parameters/features discussed, please specify each time which features are compared. For example, "... more common for broadband spatial frequency stimuli compared to broadband orientation stimuli".

• L. 255-57: Again, difficult to read. Not clear how this statement follows from the previous sentence. What is "diverse response modulation" (up and down modulation?) and where do we see that?

• L. 500: better terms for "excitatory or inhibitory responses" may be "enhanced or suppressed responses"

• L. 523-25: Description could be clarified in this sentence stating that spatial frequencies will be discriminated while orientation bandwidth is increased, and vice versa.

• L. 569: delta AUC was not defined. Should it read absolute AUC?

• L. 603ff.: Please add to the main text that stimuli of different bandwidths were grouped into blocks, not intermixed within a block.

• L. 694-705: The argument could be clarified by stating the assumption: if neurons were broadly tuned to spatial frequency, they would be expected to respond to all spatial frequency bandwidth stimuli. The data show that they are not tuned broadly relative to the tested spatial frequencies.

• L. 1233: Fig S4g,h don't exist

• Figure S8b: an explanation/cartoon of how the curves are constructed from the data in a would help readers unfamiliar with ROCs; our impression is that the green curve is not consistent with the data in a (right side) maybe because of the smoothing by the violin plot.

• Figure S9a: Correlation measure is not clear. Is each correlation for a single neuron? But in response to how many different stimuli and how many trials? This correlation seems to measure how similar the response of a neuron is to different stimuli, not the overlap of populations responding to different stimuli.

Dr Sylvia Schroeder, Dr Liad J. Baruchin
University of Sussex, UK

Reviewer #2

(Remarks to the Author)

First, I would like to thank the authors for their effort in addressing mine and other reviewers' comments. The new manuscript has been substantially modified - just the the rebuttal letter is an impressive 30 pages.

I believe that the manuscript has much improved. Comparisons to the baseline Gabor filter model show which effects are expected by default and which reveal more complex properties of V1 populations. Below I list suggestions that I believe should be incorporated prior to publication. I leave the final assessment of the fit of this work to Nature Communications to the editors. I can say, however, that I find this manuscript a thorough, rigorous and systematic contribution to our understanding of population coding in the rodent visual cortex.

Minor comments:

1. Please add the equations describing the surround suppression model. I can't find it in the main text. The mechanics of the model should be clearly explained and motivated. In particular please explain why did you choose subtractive (?) and not divisive normalization.

2. Parameters of the model were fitted to data - while this does not affect the interpretation, it should be perhaps explained whether it is a prediction or a fit.

Typos:

l. 1176 - envelop -> envelope

Reviewer #3

(Remarks to the Author)

The authors have significantly improved on the original version of the manuscript by adding new data, new analyses, and a model to substantiate their findings.

They have improved the writing and methods and completely clarified my previous questions.

The only remaining very minor issue for me is in the results for Figures 1 and 2, they report the results of the Wilcoxon signed-rank tests. They test the mid vs. the narrow bandwidth, but it is unclear what the test is for for the broad bandwidth (against mid or narrow). This is the case for both orientation and SF bandwidths, as well as for amplitude and percentage of responsive cells. Perhaps a Kruskal-Wallis ANOVA would be more appropriate here.

Reviewer #4

(Remarks to the Author)

The authors have done a great job responding to both my concerns and those of the other reviewers. The addition of copious new data sets and additional modifications to the statistical approaches greatly strengthened the work. I am satisfied with their manuscript at this stage.

Version 2:

Reviewer comments:

Reviewer #1

(Remarks to the Author)

The manuscript has much improved with this round of revisions. All our concerns and comments were considered and responded to. We thank the authors for all their efforts and great work that went into the preparation of this manuscript.

Two tiny remarks the authors may want to consider before publication:

1. L. 336: "the simple tuning-only model overestimated the increase in response amplitude by neurons with orientation tuning that was far away from the central 0° orientation". Wouldn't that imply that the yellow curve in Fig 3d is above the neural response median for neurons tuned to -45 and 45 deg? But it is below.
2. Equation in l. 1280: It wasn't obvious how the sum of sums equal the fraction (0.01622/0.1627). The fraction is well explained in the text but we couldn't relate it to the sum of sums (maybe it should be a fraction of sums?). Also, the definition of $a(o,b)$ is missing?

Dr Sylvia Schroeder, Dr Liad J. Baruchin
University of Sussex, UK

Reviewer #2

(Remarks to the Author)

I appreciate the changes introduced by the authors. I do not have any further comments beyond my last review.

made.

Reviewer 1

Summary

The authors employ visual stimuli of medium complexity, i.e., containing a larger spectrum of orientation and spatial frequency features than commonly used stimuli, to test neural response statistics in mouse primary visual cortex. They find that stimuli made up of a broader range of orientations drive more neurons and increase neural response amplitude, while stimuli made up of a broader range of spatial frequencies did not result in an overall change in neural responses. V1 population responses to different spatial frequencies could be discriminated better when stimuli contained a larger range of orientations. Similarly, mice could discriminate spatial frequencies better in broadband orientation stimuli.

Major comments

Overall, the study reveals new insights into how V1 represents visual stimuli and links these insights to behavioral performance. In particular, the increased neural and behavioural discrimination with broadband orientation stimuli are intriguing.

However, we feel that the study would greatly benefit from a more detailed comparison between single neuron responses to the motion cloud stimuli versus the classic grating stimuli (more details below). Secondly, it would help the reader if the authors go into more detail comparing their study to other studies using similar approaches or having similar aims to reveal their unique contribution better. In our opinion, an attempt to bridge the insights from using the classic stimuli versus the more complex stimuli may help in that respect.

We thank the reviewer for pointing out the general value of our work. Following their suggestions, we have strongly increased the content of the manuscript by including a large body of new experimental data and additional analysis to compare the responses of single neurons to motion clouds versus classic grating stimuli.

As detailed further below, we have added a bank of Gabors model to predict cortical responses to motion clouds, based on their tuning to simple gratings. The model results suggested that surround inhibition is a key factor to accurately predict increased response strengths, aside from orientation tuning alone. Following up on this finding, we performed new experiments to measure the orientation tuning and surround suppression of single neurons and predict their respective responses to motion clouds. In line with our theoretical results, we found that surround suppression was highly predictive of increased neural responses to motion clouds and outperformed orientation tuning as a regressor in a linear mixed model.

We have also strongly improved our writing to highlight these new findings and put them into better context to studies which have used similar approaches. Our unique contributions in respect to this earlier work are:

- 1) The differential impact of orientation and spatial frequency on neural activity, highlighting that increasing the bandwidth of different features has non-uniform impact on cortical responses. This now also includes the effects from a mixed condition which strongly drives neural activity in deeper cortical layers and the superior colliculus.**
- 2) Our new experiments and modeling work mentioned above, highlighting the importance of surround suppression for explaining the non-linear response behavior of cortical neurons to less predictable complex stimuli.**
- 3) Our neural and behavioral results, demonstrating that increasing orientation, but not spatial frequency, bandwidth enhances the separability of visual stimuli. These results extend earlier work from theoretical models, hypothesizing that complex visual stimuli might provide perceptual benefits, by experimentally demonstrating the impact of different broadband stimuli on visual perception.**

More detailed comments

1. A number of studies used similar approaches or had similar aims. Here are just some examples, some of which the authors already cite:

Walker EY, Sinz FH, Cobos E, Muhammad T, Froudarakis E, Fahey PG, Ecker AS, Reimer J, Pitkow

X, Tolias AS. 2019. Inception loops discover what excites neurons most using deep predictive models. *Nature Neuroscience* 22:2060–2065. doi:10.1038/s41593-019-0517-x

Yu Y, Stirman JN, Dorsett CR, Smith SL. 2022. Selective representations of texture and motion in mouse higher visual areas. *Current Biology* 32:2810-2820.e5. doi:10.1016/j.cub.2022.04.091

Han X, Vermaercke B, Bonin V. 2022. Diversity of spatiotemporal coding reveals specialized visual processing streams in the mouse cortex. *Nat Commun* 13:3249. doi:10.1038/s41467-022-29656-z

Bolaños F, Orlandi JG, Aoki R, Jagadeesh AV, Gardner JL, Benucci A. 2022. Efficient coding of natural images in the mouse visual cortex. *bioRxiv* 2022.09.14.507893. doi:10.1101/2022.09.14.507893

It would help to put the current study into the context of the findings of these (and possibly other) publications. What were the insights from those studies, what is the contribution of the current study? Instead it is not clear how relevant the comparison to the human study (1.297) is, given that it is about speed perception, which is not tested here.

We agree that a better link to the existing literature is important to put our study into context with earlier work and clarify the contributions of our work. In addition to performing a large amount of new experiments, we have therefore strongly expanded the introduction and discussion sections and now included the suggested references as well as several other earlier studies.

Generally, we find enhanced responses to broadband visual stimuli similar to previous studies, especially in awake monkeys and mice. We made use of parametric stimuli, so called motion clouds, that allowed us to match our visual stimuli to the broadband distribution of orientation and spatial frequency in natural scenes and characterized the different impact of these broadband features. Importantly, we find a stronger impact of broadband orientation on neuronal responses in the visual cortex compared to enriching spatial frequencies. We now added more thorough analysis, model studies and further experiments revealing that this difference is mainly explained by a difference in surround suppression of these visual stimuli instead of orientation tuning of individual neurons. Increasing the orientation bandwidth in random-phase motion cloud textures reduces the predictability within the visual stimulus leading to reduced surround suppression. The higher saliency of the broadband motion cloud stimuli also led to improved feature discrimination by V1 neurons and improved the behavioral performance of mice in a visual discrimination task. We believe that this work significantly extends previous knowledge about the impact of feature distributions in natural scenes on their perception and neural computations in the visual cortex. We have now also put these findings in context with the literature.

Our previous focus on the human study (Simoncini et al.) was motivated by their use of motion cloud stimuli to study visual perception. However, we agree that their focus on speed perception differs from our work and have therefore reduced our discussion of this particular work.

Regarding the specific studies, mentioned by the reviewer, we here provide a point by point response on how our study contributes novel insights to this existing body of work:

- Walker et al. use artificial neural networks that were trained on neuronal responses to natural stimuli to determine the most excitable stimuli at the receptive field center. They show that Gabor-like stimuli are not optimal for V1 neurons, demonstrating the importance of non-linear inputs to evoke strong neural responses.

In contrast, we explore the impact of specific non-linear stimulus features, orientation and spatial frequency bandwidth, on neuronal population dynamics and perception.

- Yu et al. focuses on the representation of natural movies in the mouse visual cortex. They then extracted different visual features, such as motion, texture or edges, and found that edge and motion features drive activity in different visual processing streams across visual cortical areas.

In our study, we deliberately focused on fully parameterizable stimuli to test the neural and perceptual impact of different low-level statistics that are still relatable to grating stimuli of specific orientation and spatial frequency. This allowed us to isolate the impact

of increasing the bandwidth of a specific or a combination of features on neural activity and found that changes in surround suppression are a key contributor to predicting neural response patterns.

- Han et al. focuses on the functional role of different visual areas and visual streams in terms of spatio-temporal processing, motion speed and spatial patterns. They also look at distinct types of neural tuning that drive responses to oriented and non-oriented spatial patterns. While they also report results from different orientation bandwidths, their stimuli are continuously rotating by 360° to also recruit neurons with different orientation tuning. Their results are therefore focused on diverse tuning to complex spatio-temporal stimulus features but do not test for the recruitment of additional neurons by broadband stimuli or the role of surround modulation.

In contrast, our main focus was to test the neural and perceptual impact when increasing the bandwidth of two well-defined low-level statistical features, namely orientation and spatial frequency. Due to our stimulus design with a constant central orientation and spatial frequency we found that broadband orientation but not spatial frequency increases neural response amplitude and recruitment. Furthermore, we added a Gabor filter bank model in order to measure the contribution of the orientation preferences of V1 neurons and found that an additional broadband-specific reduction in center-surround suppression was needed to describe broadband responses. We also confirmed this in additional experiments and now emphasize the importance of surround modulation for driving increased responses to broadband stimuli. Lastly, we found that broadband orientation stimuli enhance the neural encoding of central spatial frequencies and even improve the perceptual ability of mice to perform a spatial frequency discrimination task.

- Bolaños et al. shows that mice can discriminate between different texture types and spectrally matched artificial stimuli which are lacking higher-order features. At a neuronal level, they find that higher visual areas, such as LM, are more selective for higher-order textures statistics than V1 and other visual areas.

While they focus on the neural representation and perception of higher-order statistics by deconstructing natural textures, we instead investigated the role of specific lower level stimulus features (orientation and spatial frequency) by creating increasingly complex stimuli with broader feature distributions. Consequently, we found differences in neural response patterns in V1 and could show that increasing orientation but not spatial frequency bandwidth improves visual perception. In our new manuscript, we now also show results from widefield imaging, demonstrating that response differences are largest in V1 instead of higher visual areas (Fig. 5b). Lastly, we now also use the same stimulus analysis suite (Portilla and Simoncelli, 2000) to extract higher order features from our motion cloud stimuli, especially in the context of surround modulation.

2. It would help to have a more detailed description of motion cloud stimuli. Maybe a simplified description of how they are generated. From the results section alone, it wasn't clear that the stimuli are moving. Although this is not an important feature for the study, it seems like important information for the reader.

We agree that providing a better understanding of how the motion cloud stimuli are generated is important and have therefore improved our description in the results (lines 108-114) and methods sections (lines 900-936). In addition, we have created video examples for each motion cloud stimulus to provide a more intuitive understanding of how the constant 1 Hz temporal frequency translates into visual motion. These are now included as our new supplementary Movies 1-4.

3. A more detailed description of how AUCs were used to determine responsive neurons would help. Even better to include an example curve.

We agree with the reviewer and now provide a better description for how responsive neurons were determined. To identify responsive neurons, we compared the mean activity across all trials in the baseline to the stimulus period and computed the AUC for each neuron to obtain a normalized measure of how well the baseline and stimulus period can be separated. The AUC is bounded between 0 and 1, and responsive neurons with higher neural activity in the stimulus period have AUC values above 0.5. To obtain a statistical measure of significantly responding

neurons, we derived the AUC from the Mann–Whitney U statistic, using the formula $AUC = U / (n_0 \times n_1)$ where U is the Mann-Whitney U statistic (derived from the rank distributions of mean activity in the baseline and stimulus periods), n_0 is the number of samples in the baseline and n_1 is the number of trials in the stimulus period (both 20 in this case). This approach therefore allowed us to obtain a standardized measure of response strength for each neuron while using a one-sided Mann-Whitney U test (instead of bootstrapping in the original version) to identify significantly responsive neurons. Here, we considered neurons as responsive to a given stimulus if their median activity during the stimulus presentation was significantly higher ($p < 0.05$) than the baseline before stimulus start. This is now also described in more detail in the Methods (lines 1020-1026) and the corresponding Results section (lines 116-121).

We also use the AUC to quantify the discriminability of neural responses for two different visual stimuli (either with different orientations or with different spatial frequencies) as shown in Figure 6. Here, we computed the median response amplitude to all stimulus presentations for a given central feature (0.04 versus 0.16 cpd. for spatial frequency and 0° versus 90° for orientation) and computed the absolute AUC as a measure of response discriminability. The absolute AUC was computed as $AUC_{abs} = |AUC - 0.5|^2$ for each neuron, resulting in a normalized metric between 0 and 1 with larger values suggesting a larger response discriminability, regardless of the response preference for a given stimulus feature.

This is now described in the Methods (lines 1096-1106). To provide a better intuition for how the AUC was used to determine the separability of neural responses, we now also provide a graphical description that shows how AUC curves can be derived from the distributions of neural responses to two different visual stimuli, which is mathematically equivalent to computing the AUC from the U-statistic, as described above. These visualizations are now shown in our new supplementary Fig. S8.

4. Figure 1.f, h, i. It would help to see how single neuron responses change across conditions, i.e., how many increase responses, how many decrease responses, and by how much? You could introduce a change index, like $(R_1 - R_2) / (R_1 + R_2)$ where R_1 and R_2 are responses to 2 conditions, then do the same for R_1 and R_3 .

We thank the reviewer for suggesting this additional analysis. We have now included two additional indices to quantify how much neural responses are changing across conditions.

First, we have now included a response modulation index to quantify how strongly neural responses are changing when increasing the orientation or spatial frequency bandwidth. As suggested, the response modulation index was computed as the mean response to the mid- or broadband stimulus minus the narrow stimulus, divided by their sum. For higher orientation bandwidths, we found that more cells increase their responses with a higher positive modulation than cells that decreased their responses.

For mid-range orientation bandwidths, 57% of responsive cells increased their response with a median modulation index of 0.34 and 43% of cells reduced their response with a weaker median modulation of -0.28. For broad orientation bandwidths, 68% of responsive cells increased their responses with a median modulation index of 0.42 and 32% decreased their responses with a modulation index of -0.29. In contrast, for both mid- and broadband spatial frequency bandwidths we found a comparable amount of positively and negatively modulated neurons (54% versus 46% for mid-range and 48% versus 52% for broadband SF stimuli). Across all conditions, the distribution of modulation indices was largely continuous. These results further extend our initial results on the neural responses to broadband orientation and spatial frequency stimuli and are now shown in our new supplementary Figure S1.

Furthermore, following a related comment from reviewer 3 (see their main point #2), we also included a bandwidth selectivity index to better identify neurons that selectively respond to narrow-, mid- or broadband stimuli. The selectivity index was calculated as the mean response to the tested feature bandwidth minus the mean of the responses to the other two bands, normalized by their respective sums.

Neurons with a significant selectivity index against a shuffle control were then selected as being selectively responsive to a specific bandwidth, while neurons with significant selectivity for two bandwidths were considered as neurons with overlapping selectivity. Neurons without significant selectivity for any bandwidth were considered as common responders. In contrast to our earlier results, this more conservative approach strongly reduced the percentage of

bandwidth-selective neurons, especially for mid range bandwidths, and mostly revealed neurons with selectivity for broad- or narrow-bandwidth stimuli. In line with an increased responsiveness to broadband orientation stimuli, more neurons showed selective responses to broad orientation bandwidths.

These new results show that neural responses do not uniformly increase with higher orientation bandwidth but have more selective response patterns, with some neurons also showing a clear reduction, similar to cross-orientation suppression (new supplementary Fig. S1d). This was also seen for spatial frequency, where a similar amount of neurons (~20%) showed selective response for either narrow- or broad-bandwidth stimuli. These new results are now shown in our new supplementary Fig. S1h.

g. The diagram doesn't help to visualize quantities. Is there a way to match area to percentages? (same for Fig 2g).

To better visualize the quantities of the different fractions, we have replaced the Venn diagrams of the previous version with Euler diagrams that match the area of each ellipse to the individual percentages. Due to the reduction in bandwidth-selective neurons against our new shuffle control, mentioned above, we now show these panels in our new supplementary Figure S1d, h.

h. Figure caption does not match statements in text (l. 119).

This has now been improved in the new version of the manuscript.

5. The use of the term "tuning" is confusing. Presumably, the authors mean tuning to specific orientations or spatial frequencies but that is not obvious. E.g., l. 116: "Instead of an overall recruitment of neurons with narrow tuning, ... tuning is ... selective ...". What is the difference between narrow and selective tuning? Or l. 317: "... neural tuning cannot explain the response preference to broadband orientation stimuli."

We agree with the reviewer that the use of 'tuning' was not easy to understand in the earlier version of the manuscript. To resolve this confusion, we have now improved the writing in the text to make it clear that the term "tuning" is only used to refer to the preferred response of neurons to the orientation or spatial frequency of a classic grating stimulus. For neurons that respond most strongly to a specific feature bandwidth (as explained above), we now use the term "bandwidth-selective responses" to avoid any confusion with tuning to classic gratings. This was also the case for the sentence in previous line 116. Here, "narrow" was referring to the specific tuning to the orientation of a grating whereas "selective" was referring to a bandwidth-selective response towards a specific orientation bandwidth.

6. Figure 2.d. Do the calcium traces show responses to different repetitions of the exact same stimulus, or to different stimuli of the same class? Single trial or averages?

The example calcium traces show randomly selected single trial responses to stimuli of the same class. For each class, we showed 20 repetitions in total, where each exact same stimulus was only shown twice. The 10 example traces in the figure therefore contain a maximum of two responses to the exact same stimulus and represent half of the total amount of the responses to each class for the example neurons. As seen in the figure, the responses of example neurons 1-3 were highly bandwidth-selective and only responded to stimulus renderings of the same class. In contrast, example neuron 4 was equally responsive to all presented stimuli. We clarified the description accordingly.

7. Regarding the specificity for certain bandwidth stimuli (e.g., l. 174), it seems possible that it is caused by the specific instances of 10 stimuli of the same class, rather than the general stimulus class. It would help to see response variation of single cells to stimuli of the same class compared to stimuli from different classes. (classes == rendering I guess)

As described above, we only showed the same rendering 2 times per recording which was insufficient to obtain a detailed quantification of the response variability of single neurons to each rendering. However, to test if neural responses were driven by specific instances of each condition rather than the general stimulus class, we followed the reviewer's suggestion and compared the variability of neural response to renderings from the same class versus stimuli from different classes. For each rendering, we computed the mean response across all neurons

and correlated this response pattern to neural responses from other renderings of either the same or different classes.

In agreement with the notion that neurons were responding to the general features of a given stimulus class instead of individual renderings, we found higher correlations for neural responses across renderings from the same class (correlations between 0.13 - 0.2 for within class) and only weak or no correlations when comparing the responses across conditions for either orientation or spatial frequency enrichment (correlations of -0.001 - 0.05 for narrow versus broad bandwidth comparison). Below are the correlations of neural responses across all conditions for either orientation or spatial frequency enrichment:

Reviewer Figure 1. Different randomized renderings of the same motion cloud bandwidth stimulus evoke comparable neural responses. (a) Grid shows the mean trial-wise correlations of neuronal responses to each orientation bandwidth stimulus ($n = 19$ trials per each stimulus condition after excluding the first trial per condition; narrow, mid, broad orientation bandwidth and 0.04 cpd central spatial frequency. Responses from all 1394 neurons responding to either condition were included). The diagonal shows 'within-stimulus' correlations across 10 different random realizations of the motion cloud stimulus. Neural responses to within-stimulus realizations were generally higher compared to 'between stimulus' correlations. Pearson correlations within the stimulus: $r_{\text{narrow}} = 0.16 \pm 0.007$, $r_{\text{mid}} = 0.14 \pm 0.005$, $r_{\text{broad}} = 0.17 \pm 0.005$; mean \pm s.e.m. Pearson correlations between bandwidth stimuli: $r_{\text{narrowVSmid}} = 0.12 \pm 0.006$, $r_{\text{narrowVsbroad}} = 0.05 \pm 0.005$, $r_{\text{midVsbroad}} = 0.11 \pm 0.007$; mean \pm s.e.m. (b) Same as in panel a but for SF bandwidth. Pearson correlations within stimulus: $r_{\text{narrow}} = 0.13 \pm 0.007$, $r_{\text{mid}} = 0.17 \pm 0.007$, $r_{\text{broad}} = 0.20 \pm 0.007$; mean \pm s.e.m. Pearson correlations between bandwidth stimuli: $r_{\text{narrowVSmid}} = 0.07 \pm 0.006$, $r_{\text{narrowVsbroad}} = -0.0001 \pm 0.0005$, $r_{\text{midVsbroad}} = 0.13 \pm 0.007$; mean \pm s.e.m. Stars mark a Wilcoxon ranked sum test for difference of 'within-stimulus' and 'between-stimuli correlations' of each stimulus to zero. Significance threshold was $\alpha = 0.0167$ after Bonferroni correction for performing 3 tests.

To also show that individual renderings from the same class shared comparable stimulus features we also computed several texture statistics, based on the work from Portilla and Simoncelli (Portilla, J. & Simoncelli, E. P. A Parametric Texture Model Based on Joint Statistics of Complex Wavelet Coefficients. *Int. J. Comput. Vis.* 40, 49–70, 2000). This is now shown in our new supplementary Fig. S5. Matching our neural results, we found that stimulus features clearly cluster for stimuli in different classes (Suppl. Fig. S5a). Although individual renderings in the same class originate from separate random realizations, these results show that they share similar statistical features and remain clearly distinct from stimuli in other classes.

8. To get a better understanding of how the responses to the broadband stimuli are related to responses to more classic gratings, preferences for the broadband stimuli should be compared to orientation/spatial frequency preferences and tuning width of single neurons. E.g., it may be expected that the broadband orientation stimulus (with center orientation of 0 degrees) drives more neurons with preferences away from 0 degrees compared to the more narrowband orientation stimulus. As mentioned earlier, a detailed understanding of this relationship could really boost the impact of this

paper. It may be one step towards showing how far responses to the motion cloud stimuli can be predicted from responses to classic stimuli.

We fully agree with the reviewer that creating a closer link between the response to classic gratings and their responses to motion clouds would strongly boost the impact of the paper. We have therefore performed a large set of new analyses and additional experiments to better address this important point. In short (more detailed explanation in the next paragraph), we found that orientation tuning is indeed predictive of neural responses to broad orientation bandwidth stimuli but this relation was insufficient to explain the experimental results. Instead, a more important factor was surround suppression which was crucially required to link model predictions to our experimental data and was also a better predictor of single-neuron responses to broadband motion cloud stimuli. These new results therefore strongly improve the impact of our work and demonstrate that surround suppression is a key factor to predict neural responses to complex visual stimuli from responses to classic grating stimuli.

First, we created a simple Gabor filter-bank model to create a prediction of how the orientation tuning of idealized neurons should affect their responses to motion clouds with different orientation bandwidth (new Figure 3). In line with the reviewer's expectation, the model predicted that increasing the orientation bandwidth should increase the responses of neurons with preferences away from 0 degrees the strongest. Conversely, the responses of neurons with tuning towards 0 degrees should be strongly reduced when increasing orientation bandwidth, due to the addition of non-preferred orientations to the visual stimulus.

We then performed new two-photon recordings in which we first measured the orientation tuning of V1 neurons before presenting the motion cloud stimuli. This allowed us to directly test if the orientation tuning of individual neurons is a strong predictor of their responses to motion clouds. In agreement with the model, we found that increasing the orientation bandwidth stimuli had the strongest impact on neurons that preferred orientations further away from 0 degrees. However, the measured impact was lower than predicted by the model and increasing the orientation bandwidth did not reduce the amplitude of neurons with 0-degree tuning as theoretically expected (new Figure 3b).

To account for this mismatch, we added a surround suppression component to the model by subtracting the activity in the surround from center responses (see suppl. Fig. S3, S4). Adding surround suppression strongly improved the model fit to the data (Fig. 3b-c), suggesting that changes in surround inhibition, rather than feed-forward tuning alone, are driving neural responses to broadband motion cloud stimuli.

Following up on these results, we performed further experiments to measure the impact of broadband stimuli on surround suppression (new Figure 4). Using receptive field mapping, we first localized the receptive field center of V1 neurons and compared neural responses to center versus full-field stimulation to obtain a surround modulation index. These experiments showed that broadband orientation stimuli reduce surround suppression whereas this effect was not significant for broadband SF stimuli. A potential reason for this difference could be that broadband orientation stimuli are less predictable from other regions of the visual field.

As suggested by the reviewer in their point 10, we now quantify this difference as the Raw Coefficient Correlation for the different bandwidth motion cloud stimuli. Here, increasing orientation bandwidth reduces spatial correlations more strongly. To further reduce spatial correlations and thus stimulus predictability, we also introduced a mixed condition with broad orientation and spatial frequency stimuli and found that this further decreased the Raw Coefficient Correlation.

Aside from changes in surround suppression, broad spatial frequency and mixed stimuli elicited weaker responses in the RF center than broad orientation stimuli, suggesting that they are weaker feed-forward inputs. These new results therefore suggest that the combined impact of changes in feed-forward input and surround modulation can explain the observed differences in neural response strengths for broad orientation versus spatial frequency stimuli. We also quantitatively confirmed these results with a linear mixed effect model, containing orientation tuning, center RF responses, and surround modulation to predict broadband responses (Fig. 4g-i). Interestingly, our new electrophysiological recordings also show that the impact of stimulus predictability might be layer- and area-specific, with mixed stimuli driving much larger neural responses in deeper V1 layers and the SC than other conditions (new Figure 5).

9. L. 326: It is not clear what the authors mean by “spatial consistency” and how it is related to stimulus predictability. The argument in the following sentence (l. 329) then seems very weak as the relation to predictability is unclear, and it is not known which aspect actually improves stimulus representation.

As mentioned above, we now show the Raw Coefficient Correlation in our new Figure 4 to quantify differences in spatial consistency, which suggests a decrease in predictability of the stimulus in the RF center from other parts of the visual field. This was particularly notable in the mixed condition which drove the largest responses in deeper cortical layers and the SC. We have now added more analysis on the spatial homogeneity of our visual stimuli using the image statistics described by Portilla and Simoncelli and show the results in our new supplementary Fig. S5 (see also next point). We have also extended our introduction and discussion sections to better explain how the differences in visual stimulus homogeneity can result in stronger neuronal responses and relate these results to the framework of predictive coding.

10. To understand the qualitative difference in results between orientation and spatial frequency broadband stimuli, it may be useful to quantify further stimulus statistics. When one compares examples between Figure 1b and Figure 2b, our impression is that stimulus texture changes in Figure 1 but not Figure 2. To determine whether other image statistics change, we suggest to quantify texture parameters as presented in:

Portilla, J., Simoncelli, E.P. A Parametric Texture Model Based on Joint Statistics of Complex Wavelet Coefficients. *International Journal of Computer Vision* 40, 49–70 (2000). <https://doi.org/10.1023/A:102655361998>, In particular:

‘Raw coefficient correlation’ which represents the regularity of the image. This might help the point about spatial predictability.

‘Coefficient magnitude statistics’, which deal with structures in the image, and again might show a difference between the orientation broadband and the spatial broadband. The suggested statistics might add and strengthen to the message of the paper. For examples of analysis refer to :

Okazawa, G., Tajima, S., & Komatsu, H. (2015). Image statistics underlying natural texture selectivity of neurons in macaque V4. *Proceedings of the National Academy of Sciences*. doi.org/10.1073/pnas.141514611

Bolaños F, Orlandi JG, Aoki R, Jagadeesh AV, Gardner JL, Benucci A. 2022. Efficient coding of natural images in the mouse visual cortex. *bioRxiv* 2022.09.14.507893. doi:10.1101/2022.09.14.507893

We thank the reviewer for this important suggestion. Indeed, the stimulus texture changes when expanding the orientation (as in Fig. 1) but not the spatial frequency bandwidth (Fig. 2). We now show the result of the Raw coefficient correlation in our new Fig. 4 and also quantified the Coefficient magnitude statistics as Cross Orientation Energy and Cross Scale Energy. These results are now shown in our new supplementary Fig. S5. These metrics primarily indicated that expanding spatial frequency bandwidth reduced coarse higher order texture features, indicated by the cross-orientation energies. In addition, we added a mixed stimulus condition with broad distributions of orientations and SF. This combination of features further reduced the presence of such higher order structures on the coarsest spatial scale.

As described above, our new results in Fig. 4 can explain the observed response differences between broadband orientation and spatial frequency stimuli (Fig. 1 and 2) and also reveal an important hallmark of visual scene processing: if visual inputs can be predicted by the surrounding visual scene, neuronal responses are reduced. We now also discuss these new results in the context of the suggested and additional relevant literature.

11. Unexpectedly, Fig. 3b, f seem to suggest that fewer neurons responded to the higher spatial frequency and the 90 deg orientation stimuli. Could the authors discuss why there is a difference (if significant)?

The recruitment of responsive neurons to larger orientation bandwidth at a higher central spatial frequency (0.16 cpd) or spatial frequency bandwidth at a 90° orientation was indeed lower compared to the central spatial frequency and orientation in Fig. 1&2 (new Fig. 6, panel b and f).

This difference was significant for the higher central spatial frequency but not the difference for 0° versus 90° central orientations (Linear mixed effects comparison, $p = 4,1 \times 10^{-4}$ and $p = 0.49$).

A potential reason for this difference could be that higher spatial frequency stimuli are a less efficient driver of V1 responses which might also reduce the impact of response modulation when increasing the orientation bandwidth. This is also in line with our finding that the mixed condition with larger orientation and spatial frequency bandwidth caused weaker responses in the RF center as broad orientation bandwidth stimuli (new Fig. 4e). We now discuss this point in our revised Results section (lines 418-422).

12. It would be useful to have a more detailed description of the temporal trial structure of the behavioral task (inter-trial interval, maximum time period between stimulus onset and response) and the “mechanics” of the task (how was the reward delivered, where was the reward delivery in relation to the screen, where exactly did the mouse have to touch the screen, etc.).

We have strongly improved the description of the temporal structure and mechanics of the behavioral task in the Results (lines 590-602) and Methods sections (lines 956-998). We also added Supplementary Movie 5, showing an example of a touchscreen task trial structure. The mouse initialized the trial by triggering the lick detection mechanism and the trial was started after an interval of 500 ms. First, a visual cue (white cross on a gray screen) was presented for 700 ms to indicate the beginning of the trial. Subsequently, two stimuli were shown on the screen (always at the same positions, left and right to the white cross) and the mouse had to report its choice by touching one of the stimuli. The mouse had up to 120 seconds to make a response. If no response was made during this time window, the experiment was stopped to ensure that mice remained actively engaged in the task. After touching one of the two stimuli, the mouse received feedback if the response was correct or wrong. After a correct response a green LED indicated that the mouse could collect a 2.5 μ l water reward from a metal lick spout on the opposite side of the touchscreen chamber. For incorrect responses, the screen turned white and the mouse also had to trigger the lick detection (without obtaining a reward) to deactivate the white light and initiate the next trial.

To ensure that mice did not develop a strong direction or repetition bias, we used a bias correction algorithm (Same/Opposite balancing after: *Knutsen, PM, Pietr M, & Ahissar E (2006). Journal of Neuroscience, 26(33), 8451-8464*). In the first ten trials the correct stimulus was randomized left or right. Subsequently, the bias correction became active if a direction or repetition bias was detected and the stimulus position was chosen to counterbalance for the respective behavioral bias. If no bias was detected, the position of the correct and incorrect stimulus was chosen randomly.

Was the low spatial frequencies and the 0 degree orientation stimuli always the target or was the design balanced (The following statements may be related but were unclear: “In randomized sessions, different motion clouds with different spatial frequency bandwidths were used. Per session, there was only one spatial frequency bandwidth used to obtain the characteristics of the adaptive procedure.”). What is the definition of “performance”?

The targets were constant over all experiments. For the orientation discrimination task, we used a spatial frequency of 0.04 cpd and a 0° orientation for the spatial frequency discrimination task. This is explained in more detail in the revised Results (lines 593-602) and Methods sections (lines 969-991).

In the earlier version of the manuscript, “performance” was used in the context of the discriminability of neural responses to visual stimuli and also the accuracy of the mice in discriminating visual stimuli in the respective behavioral paradigms. To avoid any ambiguity, we now only use the term “performance” with regards to the perceptual accuracy of the mice in the discrimination tasks. The discrimination performance was defined as the percentage of correct choices out of all performed trials in the orientation or spatial frequency discrimination tasks (shown on the y-axis of new Fig. 7c-e and the Results section lines 607-614).

13. Figure 4.c,d. The absolute differences between performance levels in c and d look comparable. Is it possible that the higher number of samples in c underlie the significant outcome, in contrast to d?

While the absolute differences between the performance levels in panels c and d are indeed comparable, there was much more variability in the performance across sessions in panel d which prevented a significant difference across spatial frequency bandwidths in this case. We also took this trend in the behavioral data very seriously and considered that there might be a true reduction in discrimination performance when increasing spatial frequency bandwidth. Therefore, we performed the staircase procedure (described in more detail in our response to the next point below) to directly determine the orientation discrimination threshold for each spatial frequency bandwidth and detect potential differences with higher accuracy. As shown in the psychometric curves and orientation discrimination thresholds in our new Fig. 7e and f, this more sensitive approach also did not show a significant difference for discrimination performance with different spatial frequency bandwidths.

We therefore do not believe that the lack of a significant difference in panel d was due to an insufficient amount of behavioral sessions. We have improved the explanation of our reasoning in this regard in our revised Results section (lines 637-651).

f. How was the discrimination threshold determined? We assume that the psychometric curve was smoothed or fit?

The discrimination threshold was determined by a staircase paradigm which modulates the task difficulty until the animal reaches a predetermined target discrimination performance. The target performance is implemented by a 3-up/8-down stepping rule where each correct response decreases the orientation difference between the target and distractor stimulus by 3°, and every incorrect response increases the orientation difference by 8°. This adaptive procedure leads to a convergence of the orientation difference between the target and distractor around the orientation discrimination threshold which is defined as $p = 100\% / (1 + (\text{Steps up} / \text{Steps down})) = 72.7\%$. The target performance of 72.7% was chosen to obtain a discrimination performance close to the inflection curve of the psychometric curve and identify the orientation difference between the target and distractor stimulus in degrees that was closest to the animals perceptual threshold (new Fig. 7e). For each session, this procedure allowed us to adjust the task difficulty until the target discrimination performance was reached, yielding an estimate of the discrimination threshold (as the difference between target and distractor stimulus in degrees) for each session. These estimates are shown for all sessions and different spatial frequency bandwidths in our new Figure 7f.

We now better explain the staircase procedure and the resulting target performance and orientation discrimination threshold in the Methods section (lines 973 - 984).

Minor comments

14. L. 16 “improved feature discrimination performance”: it is not clear whether this refers to neural or behavioral performance

This referred to neural discrimination performance. We have improved our abstract and revised this sentence to now state ‘...increasing orientation bandwidth also increases the separability of neural responses to different spatial frequencies...’.

15. L. 210: “broadband stimuli” should be changed to “orientation broadband stimuli” as this statement is not true for spatial frequency broadband stimuli. The statement in l.296 is similarly vague.

Thank you, we have corrected these statements accordingly.

16. L. 395: “90 deg angle” is ambiguous. In relation to what? In which axis?

This statement referred to the central visual axis of the right animal’s eye. We have changed this sentence to state that the screen was placed orthogonal to the animal’s central line of vision from the right eye.

17. L. 440: Why is the p-value divided by 12 (rather than any other number)?

We apologize for the confusion. This sentence corresponded to an example case where 12 group comparisons were made, but due to a mistake in the subsequent writing this ended up reflecting the general description of the Bonferroni correction. We have accordingly fixed this

mistake and further detailed the description of the statistic in the methods section “Data analysis and statistics”.

18. Minor typo: Line 234 should read custom-built

Thank you for finding this mistake. We have corrected the writing accordingly.

Reviewer #2 (Remarks to the Author):

This work explores the relationship between statistical properties of naturalistic visual stimuli (motion clouds), population-level responses in the mouse primary visual cortex and perceptual discrimination. There are two, related central claims of this article: i) increasing the bandwidth of stimulus orientation engages V1 population more strongly than increasing the frequency bandwidth, and ii) perceptual discrimination improves with increasing orientation bandwidth and not with increasing frequency bandwidth.

The paper addresses an interesting and relevant question about the sensitivity of V1 populations and perception to higher order statistics of natural scenes. It simultaneously links neural coding and behavior - which is its definite strength. The writing is quite lucid, and I particularly appreciate very clear figures with a nice color scheme (blue-orientation, purple-frequency), which greatly helps parsing the results.

At the same time, I have some major concerns about the relevance of the study, its relationship to previously published results and the methodology. It is my opinion that these concerns should be addressed prior to the publication. I'm, however, not certain if they can be readily addressed at this stage of the project.

We thank the reviewer for their valuable comments on the strengths and weaknesses of our study. We have taken their feedback very seriously and performed a series of additional experiments, data analysis and modelling work to fully address the concerns raised by the reviewer. This has resulted in a major increase in the overall content of the study, including 3 new main figures and 9 new supplementary figures.

We also revised the introduction and discussion sections to better include earlier work on parameterizable and naturalistic stimuli. As already mentioned in our response to reviewer 1, our unique contributions in respect to this earlier work are:

1) The differential impact of orientation and spatial frequency on neural activity, highlighting that increasing the bandwidth of different features has non-uniform impact on cortical responses. This now also includes the effects from a mixed condition that combines broadband orientation and spatial frequency and strongly drives neural activity in deeper cortical layers and the superior colliculus. Our new imaging results now also show that response differences to broadband stimuli are largest in V1.

2) Our new experiments and modelling work, highlighting the importance of surround suppression for explaining the non-linear response behaviour of cortical neurons to less predictable complex stimuli.

3) Our neural and behavioural results, demonstrating that increasing orientation, but not spatial frequency, bandwidth enhances the separability of visual stimuli. These results extend earlier work from theoretical models, hypothesizing that complex visual stimuli might provide perceptual benefits, by experimentally demonstrating the impact of different broadband stimuli on visual perception.

Major concerns:

1. The article neglects a large field devoted to developing parameterizable, naturalistic stimuli which interpolate between simple, artificial signals and fully natural images. To mention some examples: Portilla and Simoncelli 2000, Freeman et al. 2013, Hermundstad et al. 2014, Yu, Schmid and Victor. 2015, Clark, Fitzgerald et al, 2014, Banyai et al. 2019 (and many more). The motion-cloud approach is presented as a singular solution to stimulus design, which is not the case. It is therefore hard to judge which specific aspects of neural coding will be addressed by choosing this specific stimulus parametrization over others.

To better address previous work on parametrized natural stimuli, we have now extended the introduction and discussion including the suggested references and several others. We also better motivate our approach of using random-phase motion cloud textures to create complex but parameterizable visual stimuli. See also our response to main point 1 of reviewer 1 who had similar concerns.

Our main motivation was to address how selectively increasing the bandwidth of visual features that are also used in regular gratings (i.e. orientation and spatial frequency bandwidth) affect neural response patterns and visual perception. Earlier work had already found that broadband stimuli, such as contrast-modulated noise movies (Niell and Stryker 2008, *Journal of Neuroscience*, 26(33), 8451-8464), increase neural responses in V1 and hypothesized that this could be due to the additional recruitment of neurons with diverse feature tuning. However, the functional impact of increasing the bandwidth of different visual features, such as orientation or spatial frequency, remained unclear. Furthermore, earlier work with noise stimuli did not compare the impact of different feature bandwidths that were matched to distributions in natural images.

We focused on motion clouds, because they allowed us to separately modulate the bandwidth of orientation and spatial frequency to compare their respective impact on neural processing and perception. In our first version of the manuscript, we found that, surprisingly, only increasing orientation but not spatial frequency bandwidth resulted in increased neural responses. Moreover, we found that broadband orientation stimuli enhance feature discrimination for individual V1 neurons and can improve visual perception of task performing animals. Such a perceptual benefit of broadband responses has been theorized before (Goris et al., 2015, *Neuron*, 88(4), 819-831), but not shown experimentally.

Our new modelling and experimental work now also directly tackle the important question if broadband stimulus responses of individual neurons are well-explained by their respective orientation tuning. While we found that some of the individual neural responses to broadband orientation stimuli were indeed related to their orientation tuning (new Fig. 3), the best predictor of single cell motion cloud responses was center surround modulation (new Fig. 4). This demonstrates that neural responses to broadband stimuli are not just explained by their tuning to simple features but are strongly shaped by network effects, such as surround modulation. Moreover, we also performed a more detailed analysis of higher order statistics in motion cloud stimuli, using Raw Coefficient Correlation and Coefficient Magnitude Statistics (Portilla, J. & Simoncelli, E. P. A Parametric Texture Model Based on Joint Statistics of Complex Wavelet Coefficients. *Int. J. Comput. Vis.* 40, 49–70, 2000). In combination with a new mixed condition that combines broad orientation and spatial frequency bandwidth, we found that reducing the overall predictability of the RF center from the surround reduces surround inhibition, which likely promotes the processing of unpredictable stimulus information and can improve visual perception. Our new widefield and electrophysiological experiments now also show that neural responses to the mixed condition (with broad orientation and spatial frequency bandwidths) are particularly strong in the deeper layers of V1 and the superior colliculus (new Fig. 5), demonstrating that the impact of less-predictable broadband stimuli on visual processing extends beyond cortical circuits.

2. Related to the point above - overall it is not quite clear what is the central, specific motivation of the article beyond application of motion-clouds. The Introduction begins with a very broad claim that the visual system is adapted to stimulus statistics. It then switches the tone and starts to read almost like an accompanying study of the motion cloud paper [20], aiming to test sensitivity of V1 to properties of this particular stimulus (lines 49-50 and 58-60). It is true that V1 is adapted to stimulus statistics, but this has been a subject of research for a long time now. The article should be much more precise in stating what specific question it is addressing, except for exposing the animal to this specific class of stimuli, even matched in some sense to natural statistics.

We agree with the reviewer that the overall motivation of the study was not clear enough in the first version of the manuscript and have therefore strongly improved the introduction and discussion sections. As described above, our main motivation was to test how increasing the bandwidth of specific low-level stimulus features, orientation and spatial frequency, affects neural processing and visual perception. A number of studies have used natural scenes or complex textures to categorize neuronal responses according to different higher-order features. In contrast, our aim was to focus on two of the most important features of natural scenes as well as regular grating stimuli and directly compare how changes in their respective feature

distributions affect neural responses. While we and others have observed before that neural responses in V1 are increased by broadband visual stimulation, a systematic comparison of different feature bandwidths has been missing and the perceptual impact of such broadband stimuli was still unclear. Our detailed comparison of the neural responses to broadband orientation and spatial frequency stimuli in combination with our behavioural results on visual perception fills this knowledge gap and our new results have significantly increased the quality and impact of our work.

As described above, our new modelling and experimental results in particular extend our initial motivation further by testing if increased neural responses to broadband stimuli are due to the additional recruitment of diversely tuned neurons, as commonly assumed. Our finding that the modulation of surround suppression, depending on stimulus predictability, is an important driver of broadband responses is hereby an important extension of earlier work.

3. A key claim is that ranges of motion cloud parameters used in experiments are matched to natural stimulus statistics. Such analysis is however not presented anywhere in the paper, and in my view this claim is weakly qualitative at best, and not substantiated at worst. Figures 1A and 2A present single images - this is not enough to justify the choice of parameters for the textures. Analyses should be performed on corpora of natural images (or one could use ranges reported in the literature, but this needs to be stated). Furthermore, I was not able to find any description of how orientation distributions or spectra were computed to generate plots in Fig 1A and 2A.

We agree with the reviewer that the analysis of natural images should be done on a large library of images to justify the choice of parameters that were subsequently used for the motion cloud textures. The single images in Figure 1A and 2A were only meant to provide examples and we had used the image library from the UPENN natural image database (1077 images) to perform the analysis. We have improved the writing of our Results (lines 105-111, 156, 220) and Methods section (lines 1114-1135) to clarify this important fact and better explain how the orientation distribution and spatial frequency spectra were computed.

The orientation distributions for each image were calculated individually and 180 bins with a bin size of 1° were selected (total range 1°-180°). For each image, a data table of measured orientation contents in the given bin of orientations was saved. We then calculated the mean and s.e.m. for each bin across all individual images to create the curves, shown in Figure 1A. The example regions to illustrate individual differences across natural scenes were computed the same way but only using sub-sections from 2 example images.

For spatial frequency spectra, we computed the 2D FFT spectrum for each image. Here, we shifted the zero-frequency component to the center and averaged energies in the frequency domain depending on their Euclidean distance to the zero frequency thereby describing the spatial frequency content of images independent of their orientation. Units of the spatial frequency spectra were then converted from cycles per image to cpd based on their presentation or acquisition. The angular size was hereby derived based on the individual camera field angle of view of each album: $FOV = 2 \times \arctan(\text{sensor dimension} / (2 \times \text{crop factor} \times \text{focal length}))$. Lastly, we calculated the mean and s.e.m. across all spectra from individual images to create the curves, shown in Figure 2A.

Beside our own analysis, our selection of feature bandwidths was based on earlier work from Girshick, Landy & Simoncelli (*Girshick, A. R., Landy, M. S. & Simoncelli, E. P. Cardinal rules: visual orientation perception reflects knowledge of environmental statistics. Nat. Neurosci. 14, 926–932 (2011)*). In this work, their natural image statistics (see their Figure 6) points towards a similar mexican hat nature of the orientation distributions with an approximately 45° optimal range around the peak. We therefore selected a bandwidth range that allowed us to capture the main points within that distribution shape.

4. The experiment studying sensitivity to frequency bandwidth is quite puzzling. The authors chose to use a single, vertical orientation of the stimulus. Given that orientation is the primary factor for which V1 cells are selective, isn't that a trivial result? One would naturally expect that limiting orientation range decreases response strengths, since only a fraction of the population responds. In fact comparing distributions in Fig 2F to the leftmost distribution in Fig 1F seems to reflect that - is that right? One potential suggestion would be to use a "null model" of V1 population tuning i.e. a bank of Gabor filters

(either parametric, or learned from natural image statistics with ICA or other algorithm). One could then simulate responses of such a filter bank to the same set of stimuli and compare to those obtained in the experiment. Any deviation of the data from such a null-model would be a signature of something non-trivial. Without analysis of this kind I find it hard to interpret the results.

We agree with the reviewer that the orientation tuning of V1 cells is likely to be an important contributor to their response to broadband stimuli and should be considered to improve the quality of our work (see also our response to main point 8 of reviewer 1). Following the reviewer's suggestion, we have therefore implemented a Gabor filter model to test how the orientation and spatial frequency tuning properties of standardized V1 neurons define their responses to broadband visual stimuli. The results from the model are shown in our new Fig. 3 and two additional supplementary figures S3 and S4.

Indeed, expanding the orientation bandwidth of our visual stimuli also increased the responses of neurons with orientation tuning that was further away from the vertical orientation (at 0 degrees) of the narrow orientation stimulus. This was observed both in the model and our experimental data (new Fig. 3b). However, while a tuning-only model captured the increased responses of neurons to broadband stimuli when their orientation tuning was further away from the central orientation, it overestimated the response reduction of neurons with 0-degree tuning (red line in new Figure 3b, c). This was because, based on orientation-tuning alone, one would expect that neurons with 0-degree tuning should be suppressed by broadband stimuli, as their preferred input from the center orientation is reduced. In contrast, although broadband modulation was weaker for neurons with 0-degree tuning, we found no consistent response reduction in our recordings.

To resolve this mismatch, we then added a surround suppression mechanism to the model. Adding surround suppression (in addition to orientation tuning) could account for the lack of response reduction of neurons with 0-degree tuning and resulted in an overall better fit to the experimental data (red line in new Figure 3b, c). This suggested that, in addition to orientation tuning, surround modulation is a key factor to predict V1 responses to broadband stimuli.

Following up on these results, we performed additional experiments to directly compare the impact of orientation tuning and surround modulation on neural responses. We presented narrow and broad bandwidth stimuli on either the full screen or only at the receptive field of the recorded neurons to compute surround modulation as the normalized response differences between center and full-field stimuli. These results are shown in our new Figure 4. As expected, the neural responses to narrow bandwidth stimuli (which largely resemble regular gratings) showed clear surround suppression with weaker responses to full-field versus center stimulation (new Fig. 4d-f). In contrast, broadband orientation stimuli reduce surround suppression whereas this effect was not significant for broadband SF stimuli. A potential reason for this difference could be that broadband orientation stimuli are less predictable from other regions of the visual field. This could be explained by the reduced spatial homogeneity within the visual stimulus as measured by the Raw Coefficient Correlation according to the algorithms described by Portilla and Simoncelli (2000) (new Fig. 4c and supplementary Fig. S5).

Aside from changes in surround suppression, broad spatial frequency and mixed stimuli elicited weaker responses in the RF center than broad orientation stimuli, suggesting that they provide weaker feed-forward input to V1 neurons. These new results therefore suggest that the combined impact of changes in feed-forward input and surround modulation can account for the observed differences in neural response strengths for broad orientation versus spatial frequency stimuli. We quantitatively confirmed these results with a linear mixed effect model, containing orientation tuning, center RF responses, and surround modulation to predict broadband responses (Fig. 4g-i). However, while all regressors were significantly related to broadband response modulation, surround modulation (inferred from responses to narrow bandwidth stimuli) was overall the strongest predictor across all stimulus conditions.

Lastly, we now also added a mixed stimulus condition with broad orientation and spatial frequency bandwidth. This allowed us to test if increasing spatial frequency bandwidth would have a stronger impact on neural responses when combined with a broader orientation bandwidth. However, comparing neural responses to the broad orientation versus the mixed condition showed no clear response increase in layer 2/3 neurons, suggesting that the lack of

response modulation by broader spatial frequency bandwidth is not explained by a limited orientation range.

Minor concerns:

1. I believe much more space should be devoted to explaining details of the analyses in the Methods section. I already mentioned the lack of explanation of natural stimulus statistics analysis. Similarly, no explanation is given how the area under the ROC curve (AUC) has been computed. What kind of classifier has been used? What was the range of threshold parameters? These details need to be included.

We agree that the explanation of the analyses and methods was insufficient and therefore have strongly expanded the Methods section in our revised manuscript. Following the above suggestions, this includes a more thorough explanation of the AUC analysis and a new supplementary Fig. S8 (see also our response to main point 3 of reviewer 1 for additional details on the improved AUC analysis to either identify responding neurons and compute the neural separability of different visual stimuli).

Importantly, we didn't use the AUC to quantify the performance of a specific type of binary classifier across a range of threshold parameters, as is commonly done to compare different models in machine learning. Instead, we directly used the mean neural response distributions for two different visual stimuli across all trials to compute the AUC as a measure of how distinct the two response distributions are. This is equivalent to creating a ROC curve by directly using the neural activity to assess if a given trial belongs to stimulus 1 or stimulus 2 while using different levels of neural activity to separate the two distributions. This approach yields a ROC curve across all activity levels in either distribution from which the AUC can be derived. This is now also illustrated in our new supplementary Fig. S8.

To calculate the AUCs, we used the Mann-Whitney U statistic and derived AUCs with the formula $AUC = U / (n_0 \times n_1)$ where U is the Mann-Whitney U statistic (derived from the rank distributions of mean activity in the baseline and stimulus periods), n_0 is the number of samples in the baseline and n_1 is the number of trials in the stimulus period (both 20 in this case). This approach is mathematically equivalent to deriving the AUC as the area under a ROC curve but is more precise and has the added advantage that we could use the results of the Mann-Whitney U test to identify significantly responsive neurons.

2. I strongly suggest including the psychometric curve for spatial frequency discrimination (as in Fig 4E). Only then one can have a full picture required to understand perceptual differences between conditions.

In previous experiments using our touchscreen chamber, mice have already been able to discriminate different orientations with high performance levels (see Wiesbrock et al. 2022). However, discrimination of different spatial frequencies has not been done before, at least to our knowledge. We therefore first tested if mice are able to perform this task. We observed reasonable discrimination performance, however only for the largest tested target-distractor difference in spatial frequencies (0.16 cpd vs 0.04 cpd, see supplementary Fig. S11). For technical reasons and size constraints of the touchscreen we could not test larger spatial frequency differences. We therefore decided to use one fixed target-distractor difference to compare the impact of different orientation bandwidths on discrimination performance. As shown in our Figure 7, we obtained an additional increase in performance using broadband compared to narrow orientation stimuli. Using the same approach for orientation discrimination, we used a target-distractor difference that leads to a similar performance as for the spatial frequency discrimination. However, since we found no significant performance difference for different spatial frequency bandwidths, we decided to broaden our experimental approach to include a wider spectrum of orientation differences. This was done to make sure that we did not miss a possible effect of the spatial frequency bandwidth on orientation discrimination. However, as shown in Figure 7, this was not the case and discrimination threshold was also similar for the three different bandwidths tested.

We would like to point out that our aim was not to compare orientation with spatial frequency discrimination performance. We rather compared the discrimination performance for each of

these features separately for different bandwidth stimuli. Also, we combined orientation bandwidth with spatial frequency discrimination and vice versa to prevent possible ambiguity if bandwidth and discriminated features overlap. These are therefore separate groups of behavioral experiments which also had to be performed in different groups of mice. In our previous experiments, we already showed that combining different behavioral tests in the same group of mice is difficult since the history of previously learned tasks and stimuli has an impact on the next test (shown in Wiesbrock et al. 2022 for comparison of cardinal vs oblique orientation discrimination). While we agree that full psychometric curves for each discrimination task would be desirable, the main issue here are technical limitations of the touchscreen chamber and the very large amount of behavioral data that is required to obtain a full curve across all bandwidth conditions in all animals. In addition, for spatial frequency discrimination, our previous test experiments have already shown a poor performance for smaller spatial frequency differences than the one we used in combination with the motion cloud stimuli.

3. It is not technically correct to say that "edge orientations in natural scenes are not statistically independent from each other" (lines 27-28). As revealed by algorithms such as ICA or sparse coding, edge filters are approximately marginally independent. They become *conditionally* dependent in presence of high-order structures (e.g. long-range contours, textures etc).

We agree with the reviewer that edge orientations are only conditionally dependent in the presence of higher-order structures. We have corrected this statement accordingly.

Reviewer #3 (Remarks to the Author):

This manuscript describes an increase in responsivity of mouse upper layer primary visual cortical neurons when the orientation bandwidth of the stimuli is increased. Surprisingly, the same is not the case for increases in the spatial frequency bandwidth used. Also surprising is the finding that some neurons appear to show bandpass tuning in the orientation bandwidth and spatial-frequency bandwidth domain, with some neurons responding preferentially to intermediate bandwidths. The authors show convincingly that the observed increases in responses to broader orientation bandwidth stimuli also lead to better perceptual discriminability of these stimuli. These results advance our understanding of primary visual cortical processing and should be of interest to visual- systems- and computational neuroscientists. In general the results are solid and well supported by the data. However, some points could perhaps have been strengthened by additional analysis or better experimental design. The methods need to be more specific and lack details.

We thank the reviewer for pointing out the overall value of our work for the visual-, systems- and computational neuroscience fields and for their very insightful review and valuable suggestions. As outlined below, we have strongly extended our analyses and performed a major amount of additional experiments to include the reviewer's advice and address their comments in full.

My main issues and suggestions, roughly in the order of decreasing importance are the following:

There are some missed opportunities: I really like their idea of increased surround suppression for the far more predictable SF bandwidth stimuli to be the reason for the lack of responsivity increases for these conditions. But it would have been very nice and straightforward to actually show it by varying SF bandwidth on wider orientation bandwidth stimuli. This could either, according to their argumentation, increase responsivity also in the SF BW domain because of even lower predictability of these stimuli. Alternatively, it could reveal that SF BW never influences responsivity of V1 populations.

We thank the reviewer for this excellent suggestion. We have now performed several additional experiments to address this possibility by combining broadband orientation and spatial frequency stimuli in a new mixed condition. In this condition, both SF and orientation bandwidth are increased, leading to a clear reduction in center predictability from the surround. This was confirmed when computing the Raw Coefficient Correlation (Portilla, J. & Simoncelli, E. P. A Parametric Texture Model Based on Joint Statistics of Complex Wavelet Coefficients. *Int. J. Comput. Vis.* 40, 49–70, 2000) of the visual stimulus, showing the lowest predictability in the mixed condition (see our new Fig. 4c and supplementary Fig. S5). Mixed stimuli also induced stronger neural responses as with narrow bandwidth stimuli but, surprisingly, did not exceed neural responses to broad orientation bandwidth stimuli. This therefore suggests that SF BW does not further increase the responsivity of the superficial V1 population, even when paired with broader orientation BW. These results are now shown in our new Figure 4.

Following up further on this point, we performed additional experiments to directly measure the impact of broadband stimuli on surround suppression. We presented narrow and broad bandwidth stimuli on either the full screen or only at the receptive field of the recorded neurons and computed center surround modulation as the normalized response differences between center and full-field stimuli. As expected, the neural responses to more predictable narrow bandwidth stimuli showed clear surround suppression with weaker responses to full-field versus center stimulation (new Fig. 4d-f). In contrast, surround suppression was lower for broadband stimuli, especially for the less-predictable mixed condition. A potential reason why the reduced surround suppression in the mixed condition did not elicit stronger responses compared to broad orientation bandwidth stimuli could be that increasing spatial frequency bandwidth reduced neural responses in the receptive field center (new Fig. 4e), suggesting that increasing the spatial frequency bandwidth also reduced the feed-forward inputs to V1 neurons. This reduction in feed-forward inputs might have counteracted a potential increase in responsiveness to the mixed condition due to lower surround inhibition (new Fig. 4g-i).

In addition, we performed additional electrophysiology experiments to test if neural responses to different orientation and SF bandwidth are similar across cortical layers. Interestingly, we found that the mixed condition induced a clear increase in neural responses but only in the deeper infragranular cortical layers. These new results therefore demonstrate that combining

broad SF and orientation bandwidth can indeed increase neural responses, potentially due to their lower overall predictability, but this effect is only seen in deeper cortical layers. Moving beyond the visual cortex, we also recorded neural responses in the superior colliculus and also found a very clear increase in neural responses in the mixed condition. Together, these new results demonstrate that increasing the complexity of the visual stimulus can strongly recruit neural response patterns in specific parts of the brain and open the door for follow-up studies to further investigate the role of this effect in visual processing. These results are now shown in our new Figure 5.

One of the main points is that they find tuning to specific orientation and SF bandwidths. Showing the percentages of cells responding relatively specifically to one or two bandwidths is good to see, however, this point of BW selectivity could perhaps be made stronger. The authors could for example consider showing the distribution of some selectivity measure. For example $(r_s - r_n) / (r_s + r_n)$ with r_s being the response to the preferred orientation bandwidth (or SF BW) and r_n the average of the other two. This would allow judging the overall level of selectivity for particular bandwidths across the population much better than the percentages (1 and 2 g) which could come from very weak differences in dF/F. If there are many units with strong tuning to intermediate bandwidths, this would make that point stronger.

We thank the reviewer for this valuable suggestion. Following this idea and also the suggestion of reviewer 1 (see their main point #4), we now computed the BW selectivity index for all cells as the response difference between the preferred orientation bandwidth and the mean of the other conditions, normalized by their sum. Furthermore, we did a shuffle control in which identities for individual stimulus presentations were shuffled 1000 times for each cell. Neurons with a bandwidth selectivity index higher than the top 2.5th percentile of the distribution of the shuffle indices for a broadband stimulus were then considered as bandwidth selective neurons (new supplementary Fig. 1d, flanks of the Euler diagram). Neurons with significant responses to all three bandwidth stimuli but no significant bandwidth selectivity index were considered as common responders (new supplementary Fig. 1d, center of the Euler diagram, n = 252 neurons). Lastly, neurons that significantly responded to two out of three bandwidth stimuli and showed a bandwidth selectivity index lower than the bottom 2.5th percentile of the distribution of the shuffle indices for the remaining stimulus were considered overlapping neurons between two bandwidth stimuli (supplementary Fig. 1d, overlapping sections).

In contrast to our earlier results, this more conservative approach strongly reduced the number of bandwidth-selective neurons, especially for mid range bandwidths, and mostly revealed neurons with selectivity for broad- or narrow-bandwidth stimuli. In line with an increased responsiveness to broadband orientation stimuli (new Fig. 1e), more neurons showed selective responses to broad orientation bandwidths.

These new results show that neural responses do not uniformly increase with higher orientation bandwidth but have more selective response patterns, with some neurons also showing a clear reduction, similar to cross-orientation suppression (new Figure 1h). This was also seen for spatial frequency. However, here we found a similar number of neurons (~20%) that showed selective responses for either narrow- or broad-bandwidth stimuli. These new results are now shown in our new supplementary Fig. 1d, h.

Similarly, with the data from supplemental figures 2 and 3, there might be an opportunity to strengthen this point: The authors could directly relate the bandwidth tuning of each neuron in the two conditions. If, for example, a neuron prefers the intermediate bandwidth for both orientations, this would be a much stronger point for this neuron being tuned to a specific orientation bandwidth, independent of the actual orientation. This could also give hints to potential mechanisms for this sort of tuning of V1 cells. To do this, the authors could, for example, simply calculate a correlation between the dF/F of each bandwidth condition in the two orientation conditions and show the distribution of the r-values. If many are high, this would point to true, strong bandwidth selectivity, independent of absolute orientation or SF. If, however, many neurons switch bandwidth-preference between the different orientations, e.g. low for vertical but broad for horizontal, this would argue more for the simpler recruitment theory. All of this would of course also be helped by a broader range of bandwidths being used.

We computed the correlation coefficient for neural responses with the two different central orientations or spatial frequencies. The results of this are now shown in supplementary Fig. S9.

Following the reviewers suggestion, we performed the analysis for all neurons with significant responses to specific BW stimuli and also the smaller set of neurons that were found to be selective after the more conservative shuffle control. In both cases, the distribution of correlation values was not significantly above zero, suggesting that bandwidth-preference depends on the central spatial frequency or orientation.

Line 176ff: The entire attempt to directly compare tuning widths between spatial frequency and orientation appears flawed. I understand that attempts to show the tuning widths between orientation and SF are not different and that the differences in increased responsiveness are not due to broader tuning to SF. However,

1. It is entirely unclear how the five SFs and five orientations and especially the step size was chosen for this control experiment and how they relate to the bandwidth experiment. These choices seem somewhat arbitrary and they are not as line 177 states the orientations and SFs they used in the motion cloud stimuli. They are an arbitrary subsample of them.

2. Importantly, there is no correspondence between an octave different spatial frequency and a 22.5° orientation shift. These are simply parameter choices. SF and orientation are entirely different measures and cannot be compared directly to say the tuning to one is "broader" than the other (regarding Figure S1).

We understand the point of the reviewer and agree that a direct comparison of the tuning widths for spatial frequency and orientation is hindered by the fact that there is no direct correspondence between specific changes in orientation and spatial frequency as they are operating on different scales. As a result, the notion that the tuning for one feature is broader than the tuning for another is incorrect and we have changed our description of these results to avoid such a direct comparison. Instead, we have improved the writing to better explain the purpose of these experiments and their implications for the interpretation of our broadband experiments.

Rather than being arbitrary subsamples, the reason why we selected the specific orientations and spatial frequencies was to obtain samples across the range of features that were most prominent within the natural images library and that were correspondingly used for the motion cloud stimuli. For orientations, we used evenly spaced bins between -45 and 45 degrees to capture samples for the orientations that were also the most prominent in broad orientation BW stimuli. For spatial frequencies, we used an exponentially increasing step size to also capture the steeper falloff and longer tail of the natural images and motion cloud distributions. This is now also indicated as gray dashed lines in our new panels a and b in our new Figures 1&2 and better explained in the figure caption.

The main goal of this experiment was to assess the link between the width of orientation and spatial frequency tuning relative to the range of the feature distributions that were used in the motion cloud stimuli. A potential reason why changing the bandwidth of spatial frequencies did not affect neural responses could have been a gross mismatch between the range in which neurons selectively respond to spatial frequencies and the narrow, intermediate and broad spatial frequency bandwidths that were used for the motion clouds. For example, if the difference between narrow and broad spatial frequency bandwidth would be too small to recruit additional neurons that are tuned to higher spatial frequencies, this could explain why broad spatial frequency bandwidth stimuli did not further increase the amount of responding neurons or their response amplitude. It was therefore important to show that the neural tuning for orientation and spatial frequency in V1 is approximately equally broad compared to the feature distributions that were used in the motion cloud stimuli. In other words, rather than claiming that neurons have equally broad tuning to orientation and spatial frequency directly, we wanted to ensure that their respective tuning width (relative to the motion cloud stimuli) was such that increasing the feature bandwidth should be within the tuning range of different V1 neurons and therefore result in an increased recruitment (if their tuning to gratings was indeed the determining factor for increasing the number of neurons and response amplitude).

We have improved our description of this point in the results section (lines 105-114, 195-203) and now clarify that the goal of this analysis was not to directly compare the neural tuning width of orientation and spatial frequency in V1 but rather to ensure that the feature distributions that were used in our motion cloud stimuli were generally appropriate to recruit differently tuned neurons in both cases. However, as described above, our new results strongly suggest that changes in center-surround suppression, rather than the recruitment of differently-tuned neurons, are the main reason for the observed responses to broadband stimuli.

It is also unclear what the Wilcoxon signed rank test compares. The tuning width for each unit? What is n? Also, if they are different, why does line 180 and the discussion state they were similar?

The Wilcoxon signed rank test was used to compare the tuning width of spatial frequency and orientation across all neurons. The n was for 971 responding neurons in total. The reason why they were described to be similar was because we wanted to emphasize that their tuning width was comparably broad relative to the distribution range of the motion cloud stimuli. We have improved both the caption of supplementary Fig. S1 (now, Fig. S2) and the corresponding results and discussion sections to clarify this point. Rather than claiming that tuning width was 'similar' for both features we now state that 'the responses of individual neurons covered a similar range of our stimulus bandwidth for both features (supplementary Fig. S2).'

There is great value in testing the orientation and SF tuning of each unit. Especially if they would be the same units measured for the bandwidth experiment (it is unclear if they are). That way there could be a direct comparison of each unit's tuning to bandwidth stimuli, as well as their tuning width for orientation and spatial frequency. However, the experiment as is, especially if performed on different units/sessions, does not prove or disprove "broader" tuning for orientation and thus their argument of broader SF tuning not being the reason for the absent increase in responsive units for broader SF bandwidths is futile.

We agree with the reviewer that testing the tuning range to regular gratings for each neuron in combination with the bandwidth experiments would be highly valuable. In the initial version of the manuscript, this was not possible because the measurements of orientation and spatial frequency tuning were done in separate imaging sessions due to the long experimental time that was required to test all 25 combinations of grating orientations and spatial frequencies shown in our new supplementary Fig. S2. To directly compare the tuning properties of the recorded neurons to their response to broadband stimuli, we therefore performed new combined experiments in which we first tested the orientation tuning of V1 neurons and then immediately followed with the presentation of different motion clouds. Since orientation tuning is a strong driver of cortical responses and we observed a stronger effect of enriching orientation bandwidth over spatial frequency, we focused on this aspect here due to recording time limitations. While we could confirm that broadband orientation stimuli recruited more neurons with an orientation preference that was further away from the central orientation, this effect was insufficient to account for the observed neural responses to broadband stimuli. Instead, our new modelling and analyses described above (new Figures 3 and 4) suggest that the difference in neural responses to broadband orientation versus spatial frequency stimuli is largely due to a reduction in surround suppression and reduced feed-forward inputs to cortical neurons (Fig. 4d-i).

The methods need some work. A lot of details are missing. One important thing would be to know whether the spatial frequency and orientation bandwidth tuning was performed in the same nine mice or whether this was separate cohorts of mice. Was it the same sessions or different sessions? In other words, do we know both the SF and orientation bandwidth tuning for the same set of cells or not?

We agree with the reviewer and have strongly improved the methods section in the new version of the manuscript. The spatial frequency and orientation bandwidth tuning experiments were performed with the same mice on different recording days. This was due to the long duration of each protocol which prevented combined recordings of both protocols. However, to obtain neural responses to broad SF and orientation bandwidth from the same neurons, we performed new experiments with new animals where we omitted the mid-range bandwidths to combine both stimulation protocols and obtain stimulus responses to all conditions from the same neurons. This is now true for all shown in our new Figures 3 and 4.

Why are there different numbers of sessions for the two stimulus sets - 16 and 18 sessions?

The difference in sessions was due to tissue regrowth underneath the cranial window making the imaging results from two sessions non-usable.

The description of the visual stimuli is lacking the size of the stimulus/screen. Was the monitor run at 30Hz? 150 frames for 5 seconds?

The monitor was a 24" BenQ XL2420T with an absolute screen size of 56,9 x 33,8 cm. The monitor was running at a frame rate of 60 Hz, the number of frames for the 5 second stimuli was therefore 300 frames. We have corrected this error in the revised version of the manuscript.

For the touchscreen task, there is no mention of how the patterns were arranged, at what size they were presented

The stimuli were presented with a circular aperture with a total diameter of 7.5 cm. Both stimuli were presented next to each other, with a distance of 20 cm between the central points.

Were there gray ISIs?

Since the stimuli were presented next to each other there was no ISI, but an ITI of 500 ms (mean gray screen) after a new trial was initialised.

How was the task timed?

The mouse initialized the trial by triggering the lick detection, which was followed by an ITI of 500 ms, where a grey screen is shown, followed by a visual cue (white cross) of 700 ms. Afterwards, the two stimuli were shown next to each other and the mouse had to touch one of them. After the touch, the mouse received feedback if the response was correct or wrong and initialized the next trial by collecting a reward or triggering the lick detection without a reward.

Was the position randomised?

In the initial ten trials the position of the correct stimulus was randomized. After ten trials, the bias correction got active to correct for direction and repetition bias. In case a bias was detected, the stimulus position was chosen to counterbalance for the bias. In case, no bias was detected, the position of the correct and incorrect stimulus was chosen randomly.

What was the make and model of the screen? What was its size?

The screen in the touchscreen chamber was an Elo Touch 1002L with an infrared touch screen frame in front. The size of the screen is 11 inch, giving a retinal arc of ~114° at a viewing distance of 2 cm when mice were close to the monitor.

Why were the target SFs and orientations not randomized but the same for all mice?

In each behavioral session, we used a fixed target SF or orientation to obtain sufficient trial counts for within-session comparison of different motion cloud bandwidths for that target feature. Since mice had to be trained on either discriminating SF or orientations, we could not alternate both stimuli within the same session or within the same mice. We could, however, have used different target SF or orientations by reversing target and distractor in an additional set of mice. This would have basically doubled the amount of experiments and mice used for this study. From our previous experience with visual discrimination experiments in touchscreen chambers, we did not expect a different outcome from these additional reversed target experiments, and therefore discarded this option.

Did all mice pass the initial criteria?

Yes, all mice passed the initial criteria.

Perhaps a video would help to imagine what the stimuli actually looked like, including the temporal component. If the TF is always 1Hz, then the different spatial frequencies would all have different speeds. Or was the pattern kept fixed and simply moved at 1Hz of the basis SF?

Thank you for this suggestion. We now include example videos for each of the 4 stimulus conditions to better show how each of them is changing over time. The pattern was dynamic with a fixed TF of 1Hz, leading to different speeds for different spatial frequencies. Due to the additional complexity when including spatiotemporal dynamics in the analysis of neural responses, we could not further investigate if the dependency of speed on different spatial frequencies might be a contributing factor for our results. However, we now mention in our new discussion section that this would be an important feature for further studies (lines 726-731).

The figures are aesthetically very nice! However, the single dots in the lightest purple are very low contrast and fairly hard to see (especially figure 4d and f). On the spectra (1&2 a&b) some x-ticks would have been nice. Especially at the spatial frequencies tested in S1. The example images in figure 3 are too small. You have to zoom to see anything and in the printed version this will only look gray.

We thank the reviewer for pointing this out. We have included all the suggested changes in the corresponding figures, by increasing the visibility of the data points, adding tick marks, and increasing the size of the example images in our new Figure 6 (former Figure 3).

Dyballa/Stryker PNAS 2018 has a similar point to more complex stimuli showing a more varied response landscape in V1 and mixed and non-linear selectivity of neurons. This paper should probably be discussed.

We thank the reviewer for suggesting this reference. We have now extended the discussion including a comparison of our results with those from the Dyballa study using visual flow patterns (lines 731-734). Their results suggest an additional mode of perception for global motion patterns which might be related to our observation of increased responses to broad bandwidth motion clouds, since visual tuning might be more complex than just explained by simple gratings.

Figure 1i shows responses well up to 16-17dF/F, but these responses are somehow not plotted in 1f, where responses max out at about 8dF/F. Which values are actually plotted and which omitted and is 1h and i not just a subset of the cells in f? Same applies to figure 2.

The reviewer is correct in assuming that panels h and i in Figures 1 and 2 show a subset of the neurons in panel f. The reason for the mismatch between the panels was that we restricted the population to neurons above the 1st and below the 99th percentile of the distribution when creating the raincloud plots. This was required to prevent the plots from being dominated by the tails of the distribution which made it much harder to see the median differences between groups. Because this was done for each plot separately, some neurons that were not shown in panel f were visible in panel i, leading to the mismatch in magnitude.

We have now corrected this issue by applying the same outlier criteria from panel f to all subsequent panels, thereby ensuring that the same neurons are shown throughout each figure. We also now describe this aspect of the data visualization directly in the figure caption as well as the Methods to avoid any confusion for the reader. Please note that this was purely a visualization choice that was needed to prevent outliers from reducing the legibility of the figure. All neurons were still used in the statistical analyses and the shown median and data distribution were not affected by restricting the length of the distribution tail.

line 65 increasing frequency bandwidth only weakly affected neural responses - this is not quite true, no? It did affect single cells quite dramatically. Just the overall responsivity wasn't enhanced as in the case for orientation bandwidth increases.

That is correct. We have adjusted this sentence to now state that 'increasing spatial frequency bandwidth modulated the responses of individual neurons but did not increase the overall responsivity of the neural population as a whole'.

line 66 it should say "spatial frequency" and not simply "frequency". Twice this line!

We have adjusted to writing to now consistently say 'spatial frequency'.

line 69: "narrow gratings" could be misinterpreted

To avoid any confusion to the reader, we have removed this statement from the sentence.

line 70: maybe should say "more closely match", as these stimuli definitely also do not fully match the statistics of natural sensory inputs.

We agree and have changed the sentence accordingly.

line 83 it is unclear here how AUC determines which neurons respond, so the shuffling procedure should be mentioned here.

To simplify the identification of responsive neurons, we have changed this approach to determine significant responses with a Mann-Whitney-U test. Here, we compare the response to baseline versus the stimulus period across all trials, instead of the previously used shuffling procedure. This is now mentioned in this sentence as well as our new Methods section. Moreover, we explain the AUC approach and identification of responsive neurons in our new supplementary Fig. S8.

line 87: should just say "neurons" as it is never tested whether these neurons are orientation tuned.

We have changed the sentence accordingly.

Figure 1b it is a bit confusing that 0° in a) seems to be horizontal but vertical in b)

We thank the reviewer for pointing this out. The selected image examples in a) are now chosen to be more consistent with the motion cloud images in b).

While discussing Figure 1 it would be good to also know about the motion component of the motion cloud stimuli. Is it drifting? homogeneously? drifting left or right? This should be clear and not have to be looked up in the methods (where it is also not reported except for the 1Hz).

The motion cloud stimuli were homogeneously drifting to the right (nasal to temporal) at a temporal frequency of 1Hz. We have now included this information to the results section when discussing the figure and extended our description of the stimuli in the methods section.

line 85/86 and 140/141, 195 it might be better to report the percentage of visually responsive units per session and not absolute numbers which will depend on FOV, expression level, etc. and thus be better comparable from animal to animal.

We agree with the reviewer and have changed the corresponding numbers to report the percentage of visually responsive neurons instead of the absolute number of neurons per session.

There seems to be a problem with the responsive neuron masks in 1d. I am assuming they should all be solid and of the same color per panel as in figure 2. Instead multiple different shades of blue show up in at least the middle panel and the masks are not solid but sometimes just an outline or parts of the cell body are coloured. Or do different levels of blue signify a difference?

The different levels of blue do not indicate additional differences and we have corrected figure 1d to now show similar shading to what is shown in figure 2d. The shown neuron masks are now solid and all in the same color for each condition.

111ff with increased neural responsiveness here, do they mean increased number of responsive cells or larger response amplitude (or both)? The wording implies increased amplitude, but the argumentation following sounds more like an increased number of cells.

The wording was meant to refer to both the increase in number in responsive cells and their larger response amplitude. We have corrected this sentence to now state 'To test if the increase in the number of responsive cells and their larger response amplitude to broader orientation bandwidth is caused by...'

Perhaps some bootstrap/shuffling procedure would be good, to show that these distributions of overlapping responsiveness are not to be expected by chance alone.

As discussed above, we have now addressed this point by computing the selectivity index to test the overall selectivity for each bandwidth across the population of neurons and then followed up with a shuffle control whereby the stimulus trial identities were shuffled 1000 times for each cell. Neurons are now selected as bandwidth selective neurons if their bandwidth selectivity index was higher than the top 2.5th percentile of the distribution of the shuffle indices for a broadband stimulus (supplementary Fig. S1d, h). See also our response to main point 4 of reviewer 1.

117 they only test for bandwidth tuning and not orientation tuning so this should just read "tuning", without orientation.

We have changed the sentence accordingly.

136 to say that the two distributions in 2a and b are matching is a vast overstatement. They are more similar than the narrow perhaps, but far from matching. Perhaps the poorer match of the SF spectrum as compared to the orientation bandwidth distribution should be discussed.

We have now improved our analysis of the orientation and spatial frequency bandwidth distribution of the natural image library to show that the spatial frequency distribution in the broad SF BW stimuli are closer to the average distribution of natural images as it appeared in the original version of Fig. 2a. However, we agree with the reviewer that the two distributions are not matching and have therefore adjusted the writing to weaken this claim. We now state that the motion broad motion cloud stimuli 'approximate' the bandwidth distributions of natural scenes.

Which spatial frequency do they use to increase the bandwidth from? Is it 0.04 as in the table in the methods? In figure 2b it looks as if the center is 0 cpd (but hard to tell without x-ticks). Or is this the difference to 0.04? Why then is it one-sided?

The central SF is 0.04 cpd. for both protocols (Fig.1 and Fig. 2) and the central orientation is also consistently 0° to maintain a fair comparison between both protocols. Our intention with figure 2b was to visualize the spatial frequency bandwidth for the narrow, mid and broad different motion cloud stimuli. The peak SF is therefore placed at 0 and only positive spatial frequencies are shown. We have now generated the panel again, and included additional input images to the analysis, added ticks in the x axis and also explained in more detail how these panels (a and b) were obtained (Methods line 1114 to line 1135).

In figure 1c and 2c, the stimulus does not appear vertical but slanted. But the text states it was vertical.

We thank the reviewer for spotting this point. We have corrected the figures to now show vertical stimuli, as they were actually presented on the screen

Is the percent responsiveness for 1 and 2d similar? Or in other words, how do all the normalized to 1 values from 1 and 2e compare? It appears to respond less to the SF BW variant. Is it because

orientation bandwidth is so narrow for these stimuli? This also relates to the earlier point of perhaps testing whether altering SF on more broad band orientation stimuli would change the lack of increase with broader distributions?

The narrow range conditions in figures 1 and 2 are indeed the same stimulus, with an orientation bandwidth of 5° and a spatial frequency bandwidth of 0.004 cpd. Only the mid- and broad-range conditions are increased for either orientation or spatial frequency bandwidth. The percentage of responding neurons to the narrow conditions across all recordings is therefore not significantly different. We now clarify this important point in the text and have changed figures 1 and 2 to report both the orientation and spatial frequency bandwidth for each stimulus condition in panel d.

As mentioned above, we now also include a mixed condition in which we combine broad spatial frequency and orientation bandwidth to address the possibility that broad spatial frequency bandwidth has a stronger impact on neural responses when combined with broad orientation bandwidth. While this does not appear to be the case for superficial V1 neurons, neural responses in deeper cortical layers and the superior colliculus are indeed much more strongly activated by the mixed condition. As described in our new discussion section, this is most likely due to the decrease in predictability with these more complex stimuli, resulting in a reduction in center-surround suppression.

Why is the target-performance for the staircase 72.7% correct?

The 72.7% target performance in the staircase paradigm is due to a 3-up/8-down stepping rule that was implemented to determine the discrimination threshold of the mice: every correct response decreased the orientation difference between the target and distractor stimulus by 3°, and every incorrect response increased the orientation difference by 8°. This adaptive procedure leads to a convergence of the orientation difference between the target and distractor around the orientation discrimination threshold of the animal, which should be between 65-75%. To achieve this, we chose the 3-up/8-down staircase rule, fixing the discrimination performance of the animal at $p = 1 / (1 + (\text{Steps up} / \text{Steps down})) = 72.7\%$. We now better explain the staircase procedure and the resulting target-performance in our new results (lines 640-648) and methods section (lines 975 - 984).

293ff it should be made clear already here that only increasing orientation bandwidths increased responses while the other had no effect on response strength or responsive population. Again in 296: only enrichment of orientation bandwidth. This distinction is discussed, but only at the end of the next paragraph.

We agree with the reviewer and changed the writing in lines 292 and 295 to make it clear that only enrichment of the orientation bandwidth induced stronger neural responses and recruited more responsive neurons in superficial V1.

186 specify orientation bandwidth
234 and 404 should be custom-built
304 should be "revealed"
305 increased sensory perception - rephrase
375 positioned and? attached
379 "with" missing
389 should be "driven" not "drive"
391 should be "were" not "when"
395 90° from what angle?
408 should be "weighed daily"
425 should be "sufficiently"
426 "to" should be "two"

Supplement
figure 1a the y axis should probably be reversed
line 8: n = XX should be a number
line 13 should be "signed rank test"

We thank the reviewer for their effort in spotting these errors. We have corrected the writing accordingly.

Reviewer #4 (Remarks to the Author):

The study by Balla et al. addresses a very interesting question that is surprisingly understudied in the mouse (despite huge literature in other species) - what types of stimuli best drive V1 neurons? They use a motion cloud tuned to different parameter spaces and measure responses in V1 neurons via 2p imaging. While the question is of broad interest to the community, a number of methodological issues substantially reduce the interpretability of the findings (see below). The overall consequence is that the results do not present a compelling, novel set of insights into stimulus representations by visual cortex.

We thank the reviewer for their thoughtful review, valuable suggestions, and pointing out that the general question is of broad interest to the community. We understand the reviewer's valid concerns regarding the interpretability of the findings in the initial version of the manuscript and have therefore performed a significant amount of additional work, including new widefield, electrophysiology and additional 2-photon experiments, new analyses and a gabor model of V1 responses to fully address the reviewer's concerns. Our new results, including 3 new main and 9 new supplementary figures, have substantially improved the quality and rigor of our work and now highlights the importance of center surround suppression as an important driver for understanding neural responses to broadband stimuli.

Major concerns:

1. The stimulus size is not specified in the methods, but appears to be whole-screen from Figure 1. V1 neurons exhibit robust surround suppression whose magnitude may be strongly dependent on the stimulus properties of the center and surround regions. Thus, it is impossible with the present data to understand which response properties are due directly to the features of the stimulus present in the classical receptive field of each cell versus the stimulation of its surround.

As assumed by the reviewer, all stimuli in the initial version of the manuscript were shown on the entire screen. However, as correctly stated by the reviewer, this does not allow us to understand if the observed effects are directly driven by the stimulus features within the receptive field or rather due to the stimulation of its surround. We agree that this is a very important question and have therefore performed a series of additional experiments and modeling analyses to identify if the observed effects are driven by the stimulus features within the center or the surround of the receptive field.

First, we created a Gabor filter-bank model, consisting of Gabors with different orientations, spatial frequency tuning, and a center receptive field to test if this simple model can account for the neural responses that we measured experimentally. The modeled responses to broadband stimuli were calculated as the filtered response in the RF center, minus a weighted response of the surround. We then fitted a sigmoid function to match the filter responses to the measured neural response amplitudes. The model achieved a good fit with the neural data, showing an increase in neural response amplitude and the number of recruited neurons when increasing the orientation bandwidth. These modeling results strongly suggest that the tuning of individual neurons is insufficient to predict their response to broadband stimuli. Instead, surround suppression was crucially required to explain the experimentally observed neural responses. The results of this new work are now shown in our new main Figure 3.

Second, we performed new 2-photon experiments to directly test the role of center-surround modulation on neural responses to broadband motion cloud stimuli in V1. We used sparse noise stimuli to perform receptive field mapping and subsequently presented a 15-degree wide stimulus in the center of the receptive fields of responsive neurons. By comparing the neural response to center-only versus whole-screen stimulation we could then assess if neural responses were primarily driven by features within the center or the surround of the receptive field. Whole-screen stimulation induced the same effects as in our initial experiments (broad orientation bandwidth inducing stronger neural responses than narrow bandwidth stimuli) but center-only stimulation showed clear differences for narrow bandwidth stimuli. Here, center-only induced much stronger responses than whole-screen stimulation, suggesting that responses to whole-screen narrow bandwidth stimuli are strongly affected by center-surround suppression. In contrast, surround suppression was reduced for broad orientation bandwidth

stimuli, suggesting that differences in surround suppression are an important driver for the observed increase in neural responses. We hypothesized that this effect was due to lower predictability of the RF center from the surround with broadband orientation stimuli and now also quantify this effect by computing the Raw Coefficient Correlation and Coefficient Magnitude Statistics (Portilla, J. & Simoncelli, E. P. A Parametric Texture Model Based on Joint Statistics of Complex Wavelet Coefficients. *Int. J. Comput. Vis.* 40, 49–70, 2000). Moreover, we introduced a new mixed condition with high orientation and spatial frequency bandwidth and found that this resulted in the lowest surround suppression across all conditions. Differences in surround suppression are therefore an important driver for the observed increase in neural responses with broad bandwidth stimuli and are most likely due to their less predictable spatial stimulus structure. These results were also in agreement with our modeling results and are now shown in our new main Figure 4.

Together, our new results show that increasing the feature bandwidth of visual stimuli not only affects neural responses by recruiting neurons with different orientation or spatial frequency tuning but can strongly reduce center-surround suppression and therefore increase the magnitude and number of responsive neurons.

2. There is no description at all of the animal's position during imaging. Figure 1 suggests the animal is in a tube, though awake or anesthetized is not specified. If anesthetized, it is hard to know if the present results generalize to awake mice. If awake, there is no discussion of how behavioral state or arousal level is monitored and included in analyses. As arousal can have profound impact on visual responsiveness, it would be necessary at a minimum to ensure this variable did not change across or within sessions. Overall, it is hard to interpret the present data without this critical information.

We fully agree with the reviewer that the animal's position and behavioral state in the setup are highly important for the interpretation of the results. Most importantly, the mice were awake in all experiments. This was mentioned in the first paragraph of the results section (old line 81) but is now stated more clearly for the different experiments and is also better explained in our improved methods section.

In our initial recordings, the animals were indeed placed in a small tube and largely sat quietly during the 2-photon imaging. However, the recorded video data in these experiments was insufficient to perform deeper analyses of the animals' arousal levels. To directly address this important point, we therefore improved the design of the setup and performed new 2-photon imaging experiments. In our new recordings, the mice were placed on a circular running wheel and we also captured video data of the animal's face and body. The results of these experiments are included in our new main Figure 4 and supplementary Fig. S7.

To identify if changes in arousal have a strong impact on our results, we used locomotion (detected by wheel movements) as a well-established indicator of arousal. While some mice were running continuously, suggesting a fairly constant arousal level throughout the experiment, other animals were largely resting and sometimes transitioned between running and quiescent behavioral states (supplementary Fig. S7b). This allowed us to isolate visual responses to motion clouds during quiet resting or arousal states and assess the impact of behavioral state fluctuations. In agreement with earlier work, we found that increased arousal during running increased visual response amplitudes compared to non-running trials (supplementary Fig. S7d). However, the increase in neural response amplitudes to broad versus narrow orientation bandwidth was equally observed in running and non-running trials, suggesting that this effect does not strongly depend on a specific behavioral state (supplementary Fig. S7e, f). Similarly, we also found that increasing spatial frequency bandwidth did not increase neural response amplitudes in either running or non-running trials and the running behavior of the animals was the same in both narrow versus broad motion cloud stimulus trials.

3. The statistical analyses are highly problematic and appear (with only minimal description) as being carried out with a "by-cell" comparison. Even with "Bonferroni Correction" (exactly what is being corrected or what the magnitude of the correction is does not appear in the text), cells within a single animal are not independent variables and cannot be analyzed as such. Most simply, analyses should

be carried out by-animal. However, a mixed-effects model where both animal and stimulus are considered could also be constructed. Again, without proper statistics, it is impossible to fully appreciate the results for which many effect sizes appear to be quite small.

We thank the reviewer for pointing out this important issue. To strengthen and better illustrate the statistical analyses, we have strongly improved our methods section and now provide a complete description of the statistical methodology and report the corresponding test statistics for each comparison in the results section.

The Bonferroni correction was done to correct for multiple comparisons when testing for significant differences between multiple stimulus pairs, such as the 3 different orientation bandwidths in Figure 1. In this case, this would mean performing 3 separate tests and correspondingly adjusting the alpha value to $0.05/3 = 0.0167$ to reach significance. We have added this information to the methods and the respective figure captions.

We also agree that pooled cells from different animals are not independent samples and therefore significant differences between stimulus conditions can not be accurately assessed with a standard Wilcoxon test. While by-animal comparisons would indeed be desirable, the neurophysiological and behavioral experiments are too time- and labor-intensive to reach the required large cohort sizes for robust statistical effects across animals. Moreover, sticking to by-animal comparisons would prevent the accurate identification of many of the cell-specific tuning effects, such as the different motion cloud responses of the subset of cortical neurons that we describe in Fig. 1 and 2 or the regression model in our new Fig. 4. This is a common problem for studies with chronic 2-photon imaging and behavior and we would like to point out that the number of animals in our 2-photon experiments (14 mice on total) is larger compared to other recent studies in *Nature Communications* (e.g. Han et al. 2022 (doi: 10.1038/s41467-022-29656-z) used 10 mice in total and performed their analyses across all recorded neurons from different mice).

However, to resolve this important point, we followed the reviewer's suggestion and implemented a linear mixed-effects model to identify significant differences between conditions while controlling for animal identity when comparing effects across pooled neurons throughout the manuscript. For the model comparison, we considered stimulus and animal identity as the fixed and random variables, respectively. The resulting t-statistics for the stimulus regressor was then used to obtain a p-value and identify significant differences between stimulus conditions. Using this approach throughout the manuscript yielded very similar results as in our initial submission and now provides a more rigorous statistical foundation to support the main conclusions of our work.

Moreover, for our main 2-photon results, where we show by-cell comparisons in the main figures, we have now expanded our analysis to individual animals, demonstrating that the same effects are also reproducible across individual mice. These new results are shown in our new supplementary Fig. S1.

4. Broadly speaking, GCaMP population imaging (6 for sure, but probably true for all) does not robustly or reliably capture small changes in firing rate (this has been clearly shown in recent Allen Institute publications). For many studies, this is only a minor issue. However, at present where the goal is to say something about the sensitivity of cells to stimulus features, it may be a major difficulty. For example, if GCaMP imaging only reports bursts of spikes, the data here may show that certain stimulus parameters increase bursting. However, much of the information in neuronal activity might be carried in the addition or subtraction of a single AP (particularly in sparsely firing Layer 2/3 cells). I think any claim of "coding principles" by these cells must include some amount of ground truth electrophysiology data.

We agree with the reviewer that GCaMP signals in imaging studies can generally be confounded by a bias of the calcium indicator toward neurons with high firing rates or bursting behavior while subtler, but equally reliable, changes in spiking responses might be overlooked. To address this important concern, we, therefore, performed additional electrophysiological experiments where we recorded the neural activity in V1 of awake mice with high-density Neuropixels probes. The results of these experiments are now presented in our new main Figure 5.

First, we used widefield imaging to reliably identify the location of V1 and the surrounding higher visual areas and confirmed that stimuli with broad orientation bandwidth elicited the strongest neural responses in V1 (new Fig. 5a-c). We then recorded V1 neural activity across all cortical layers in response to visual stimuli with narrow orientation and spatial frequency bandwidth ('narrow'), broad orientation or spatial frequency bandwidth ('ORI' and 'SF', respectively), or a combination of broad orientation and spatial frequency bandwidth ('mixed'). Across cortical layers, we found that broad orientation stimuli evoked stronger spiking responses as with broad spatial frequency bandwidth (Fig. 5f-g). To quantify this effect, we computed the AUC for each stimulus condition, comparing baseline spiking activity to the stimulus period. Since the AUC is a measure of separability between spiking in the baseline versus the stimulus condition, this ensures that our results would not be dominated by neurons with higher firing rates and should also reveal more subtle changes in sparsely firing neurons. We have also largely expanded our explanation of our implementation of the AUC and show a visualization in our new supplementary Fig. S8 (see also our response to main point 3 of reviewer 1 for more details on the AUC implementation). In agreement with our imaging results, we found that broad orientation bandwidth induced stronger responses as broad spatial frequency bandwidth in both cases (Fig. 5f-h and supplementary Fig. S8).

These new results and analysis therefore provide electrophysiological confirmation that the increase in neural responses with broad orientation bandwidth is not explained by a bias of the calcium imaging and extends our results to different cortical layers and another visual brain region.

Less major concerns:

5. Please report responsive cells as a fraction of total cells imaged, not only the absolute number.

Following this suggestion, we now include both the absolute number of cells as well as the fraction of total cells imaged throughout the manuscript.

6. There are no numeric data presented anywhere in the text (means, SEMs, quartiles, stat values, etc. should all be present in the Results text).

The results section of the initial submission already included numeric data (mostly reporting mean \pm s.e.m and sample numbers; see for example, old lines 121-124, etc). We assume that the reviewer is asking for a more thorough description of the numeric data in the main text and a better explanation of the statistical results. Following this important suggestion, we have improved the results section and now consistently report the means, SEMs, quartiles, test statistics, and sample numbers for all discussed findings throughout the manuscript.

7. The volume and titre of virus injected and the time after injection for imaging sessions are not specified.

We thank the reviewer for pointing out this omission. We now include this information in our improved methods section (see lines 784, 788).

AUTHOR RESPONSES TO REVIEWER COMMENTS

Reviewer #1

(Remarks to the Author):

The current revision of the manuscript has greatly improved the impact of the study. The authors have placed their specific research question and their results in context of previous studies on similar topics. By using a model of V1 neural responses, by relating responses for complex (motion cloud) and simple (grating) stimuli, and by recording responses to small versus fullfield stimuli, the manuscript now provides much clearer insights into circuit mechanisms behind their findings. The additional electrophysiology data from V1 adds a strong confirmation of the two-photon data and provides new insights on response differences across layers of V1. Finally, the provided movies showing the stimuli are very helpful.

The authors have also added new data on higher visual areas (HVAs) and the superior colliculus (SC). Although we think that the data are of high quality, it is our impression that they distract from the main story of this manuscript, which focuses on V1 and provides a comprehensive narrative on how neurons in V1 respond to narrow versus broadband stimuli. The additional data on other visual areas is very limited in comparison and raised several questions (see Major comments). Although they provide an interesting incentive to study broadband responses in other visual areas, we would recommend to leave these to a separate publication and/or put the current data into Supplementary figures. However, we just wanted to share our impression. The decision should be made by the authors.

We thank the reviewer for their insightful comments and pointing out that the major revision of the manuscript greatly improved the impact of our study. In our new version, we have done our best to further improve the work and address all their additional comments in full.

Our main motivation for performing the additional widefield experiments and including HVAs beyond V1 was to demonstrate that V1 is the cortical area with the strongest response modulation by our broadband stimuli (see also our new Fig. 5c). This was important because different HVAs have been shown to respond to higher-order stimulus features which might suggest that broadband responses are much stronger in areas outside of V1. Our new data shows that this is not the case, and V1 is indeed a key region to study broadband response modulation. We fully agree with the reviewer that a deeper focus on HVAs is beyond the scope of our study and have therefore improved the description of this result to keep the focus on V1.

The additional results from SC were included due to the opportunity to also record broadband stimulus responses in a non-cortical visual area in our electrophysiological experiments. The pronounced broadband modulation in the SC stood out to us as a very interesting extension of our results, which is why we initially included them in Fig. 5. However, we agree with the reviewer that the SC results are not the main focus of the study and might distract from the much more exhaustive V1 results. We have, therefore, decided to move these results to our new Supplementary Fig. S11 and improved the text in the results section to avoid this issue.

Major comments

*More details about the Gabor model need to be provided, together with a better explanation of Figure S4a. Please describe which Gabors contribute to the surround modulation of one cell. What is their size, their preferred orientation, and their spatial frequency/spatial aspect ratio? A cartoon showing the spatial layout of all Gabors relative to the stimulus would be helpful as well. Which data were used to fit the Gabor models (tuning only, and surround suppression)? Specifically, for the fit mentioned in l. 1225? **We agree that the previous illustrations of the model were a bit overloaded, making it difficult to get a clear picture of the implementation. To enhance clarity, we now simplified the model visualization in Figures S3 and S4, focusing on the most important components of the model. We also re-wrote the model description in the Methods to provide additional details and clarity.***

The Gabor filters covered the full range of orientations from -90 to 90 degrees in steps of 1 degree and five different spatial frequencies (0.01, 0.02, 0.04, 0.06, 0.08 cpd). We now also show

the spatial size of the different Gabor filters relative to the stimulus space in the top row of Figure S3b. The spatial aspect ratio of the Gabor filters was adjusted to match the orientation tuning width of mouse V1 neurons, leading to a Gabor size of ~15 degrees. Our Supplementary Figure S3c also shows the range of the five spatial frequencies that were used for the different Gabor filters. For the model fits of the tuning only and surround modulation models, we then pooled responses from all five spatial frequencies to represent a broad tuning range of V1 neurons.

The main motivation for the model was to explore whether the increased responses of V1 neurons to broadband orientation stimuli could be simply explained by the additional recruitment of neurons with different orientation tuning (in particular in response to the main point #4 of reviewer 2 in the previous review round). To fit the model, we therefore focused on the experimental data shown in main Fig. 3 to directly test if a simple Gabor model could explain the broadband response modulation of V1 neurons with different orientation tuning (Fig. 3d).

To derive a simple prediction from the Gabor model (tuning only model), we only used a scale factor w_t to fit the average response magnitudes of the Center Gabors to the average dF/F magnitudes of the experimental data (Supplementary Fig. S4). Specifically, the Gabor response magnitude was scaled to obtain the best fit to the measured responses of V1 neurons with different orientation tuning to both narrow (Fig. 3b) and broad orientation bandwidth stimuli (Fig. 3c). The results of this fit are now shown as yellow lines in our new Fig. 3b and c. Most importantly, the best fit of the tuning-only model shows much stronger responses to the narrow versus broad orientation bandwidth for neurons with 0° tuning, which should also result in a negative broadband response modulation of these neurons (Fig. 3d). This model result also makes intuitive sense, as neurons with 0° tuning should strongly respond to narrow bandwidth stimuli at their preferred orientation but weaker to broadband stimuli with diverse orientation content. Due to this trade-off, the tuning-only model predicted no overall increase in neural responses to broadband orientation stimuli (Fig. 3d) and was, therefore, insufficient to explain our experimental results.

In the surround suppression model, we computed the surround response by averaging over all responses of the surrounding Gabor filters with the same size and orientation tuning as the center Gabor. An illustration of the spatial layout of the Gabors, which contribute to the center and surround responses relative to narrow and broad orientation bandwidth stimuli, is now shown in the revised version of Supplementary Fig. S4. The average surround response was computed in a circular surround region with an inner radius of at least 7.5° and an outer radius of up to 15° away from the center Gabor. The exact surround area was fitted in the model, resulting in an area size with an inner radius of 11,19° and an outer radius of 12,19°. We then thresholded the average response magnitude in the surround region area and applied an additional weight w_s to the surround by which the responses in the center region were divided. Aside from a much more detailed description of the model in the Methods section, we now also provide a graphical illustration and the full formula for this divisive surround-suppression model in Supplementary Fig. S4. In contrast to the tuning-only model, which only scales center responses to dF/F, the suppression model, therefore, also contains a thresholded and rescaled surround response to which the center response can be normalized. Similar to the tuning-only model, the suppression model was fitted to the experimental narrow and broad orientation bandwidth responses in Fig. 3c and d to obtain a single model fit for both stimulus sets.

We are now presenting the fits from both models to V1 responses to narrow and broad orientation bandwidth stimuli in Fig. 3b and c, matching the improved illustration of the model fitting procedure in Fig. S4. This addition greatly improves the overall presentation of the model and directly shows the main effect when adding surround suppression to the model fit. The main difference between the tuning only and the suppression model is their fit for V1 neurons with orientation tuning close to 0°. While the simple Gabor model strongly overestimated the response of these neurons to narrow orientation bandwidth stimuli, adding surround suppression enabled the model to selectively reduce these responses through divisive normalization while obtaining more accurate fits for neural responses to broadband stimuli. Aside from demonstrating that the increased V1 responses to broadband stimuli can be explained when adding surround suppression to our model, these results also demonstrate that surround suppression is expected to be stronger for simple narrow- versus broad bandwidth stimuli. We then confirmed this prediction in our subsequent experiments, shown in Fig. 4.

L. 1198 and Fig S4b-e: does the spatial frequency of the response curves (title of plots) refer to the stimulus spatial frequency or the spatial frequency of the Gabors?

The spatial frequency labeling of the response curves matched the spatial frequency of the Gabors. However, to improve the model description, we now illustrate the composition of the Gabor model in Supplementary Fig. S3. We show the model response to stimuli with different spatial frequencies in the top row of Fig. S3c (titled 0.01, 0.04, 0.08 cpd), using different Gabor-filters with spatial frequencies from 0.01 to 0.08 cpd as indicated on the y-axis in the middle row. These plots clearly show the spatial frequency tuning of the different Gabor filters in the model. For the model fit to the motion cloud stimuli shown in Supplementary Fig. S4, we used this bank of Gabor filters with a range of spatial frequencies from 0.01 to 0.08 cpd.

I. 1209: Is this statement reflected in the presented data/plots? Where?

This sentence referred to an earlier version of the model/manuscript. We thank the reviewer for catching this mistake. The entire section describing the model in the Methods section has now been improved to provide a much more detailed description of the Gabor models, including all relevant equations.

Figure S3: more details need to be added.

Do the curves in the 2nd row of plot b show the same curve as in a but for different spatial aspect ratios? If yes, please state this in the figure caption. Is the spatial frequency of the stimulus the same as in a? Are response curves in c for the tuning only or the surround suppression model?

The response curve in panel a is identical to the curve presented in panel b for the matching spatial aspect ratio of 0.55. We now clarified this figure, improved the figure caption, and clarified the correspondence with the spatial aspect ratio. In the earlier version of the manuscript, we also showed responses of the fitted suppression model to confirm similar tuning; however, we see now that this convoluted the message of this figure. Instead, the new version of Fig. S3 now shows the direct Gabor filter responses to full contrast sine-wave gratings that we used to determine suitable parameters for the Gabor filters. These Gabor filters represent the basis of both the tuning only and the suppression model that we subsequently used to fit our experimental data, as further illustrated in Fig. S4.

Figure S4: b,d: legends "narrow" versus "ori" are confusing. Better "narrow" versus "broad"

We thank the reviewer for their suggestion. We agree that these terms were confusing as we introduced them only later in the context of Fig. 4. We have now modified Fig. S4 and labeled the stimuli as "narrow" and "broad orientation bandwidth" accordingly.

Figure 3: To better compare the model predictions (Fig S4b), which show response amplitudes depending on preferred orientation, it would be very useful to have a similar plot as 3b but showing response magnitude (y-axis). These are the data used to rescale the Gabor responses (I. 1204-06). We agree with the reviewer and now show the response magnitudes directly in our new main figure panels 3b and 3c, along with the respective fits from the tuning only and surround suppression model. This makes it much easier to understand how the model was fitted to the experimental data and directly shows that surround suppression reduces neural responses to narrow bandwidth stimuli in particular. It also allows the reader to better understand how the response modulation in panel d is derived from the narrow and broad bandwidth responses.

The model predictions in 3b (yellow versus red lines) look like they could be scaled versions of each other.

We understand that presenting the response modulation without the model fits for the narrow- and broad orientation bandwidth stimuli could give this impression. However, as explained in our response to main point #1, the clear difference in the predicted response modulation by the two models is because the tuning-only model could not accurately fit the neural responses to narrow- and broad-bandwidth stimuli together. This is now clarified in the new version of Figure 3, where we show the fitted neural responses in Fig. 3b,c.

Moreover, following the suggestions from Reviewer 2, we have now also explored a divisive instead of subtractive surround suppression model and found that this approach provides an even closer fit to our neural measurements. Our new suppression model with divisive surround suppression highlights the difference of the fitted responses relative to the tuning-only model

more strongly, making the difference in the best model fits (yellow versus red lines) even more prominent.

We were wondering how dependent the scaling of these curves is on the model parameters. Are there model parameters that would lead to a downscaling of the tuning-only model (yellow curve), which would lead to a better fit for the tuning only compared to the suppression model?

As described in response to main point #1, we now better illustrate how the scaling of the tuning-only model is implemented in the new Supplementary Fig. S4.

We did not fit the response modulation curves in panel 3d with the models. Instead, we fitted both the tuning only and the suppression model on the neural responses to narrow and broad orientation bandwidth stimuli simultaneously, which are now also visualized in Fig. 3b, c. The response modulation is then derived from these fitted responses as the relative change in neural responses to narrow- versus broad orientation bandwidth stimuli.

Moreover, the tuning-only model only contains a single scaling term to translate Gabor responses to dF/F. Because the response modulation is a relative metric between narrow and broad responses, the resulting yellow curve in Fig. 3d would, therefore, always come out the same way, regardless of the scaling of the yellow curves in panels 3b and c.

Could the authors add the model predictions for plots d and e? Did the authors expect any dependence of SF response modulation and recruitment modulation on orientation preference?

The possibility that SF response and recruitment modulation could depend on orientation preference was raised by reviewer 2 in their main point 4 in the first round of the review. Here, the assumption was that orientation is the primary factor for which V1 cells are selective, which could limit neural response strengths to broadband SF stimuli. To address this possibility, we wanted to show that there is no clear dependence between SF modulation and orientation tuning in our old Fig. 3d and e. However, given that there is no relationship between orientation tuning and SF response strength, producing meaningful model fits is not possible since this would just mean fitting a straight line to the experimental data.

As explained above, our main motivation for the model (following the earlier reviewer suggestions) was to explore if larger responses to broadband orientation stimuli can be simply explained by recruiting more orientation-tuned neurons. To achieve this goal, we performed additional experiments focusing on the role of orientation tuning of individual V1 neurons, which allowed us to fit the Gabor-filter model and demonstrate that orientation tuning alone is insufficient to explain the increased neural responses to broadband orientation stimuli. Due to experimental and time constraints, we could not measure the spatial frequency tuning of the same neurons and further extend our model fitting. However, we think such an additional effort is beyond the scope of the current study since we found no increase in neural responses to broadband SF stimuli (Fig. 2), and the focus of the model was to explore the role of orientation tuning for explaining increased neural responses to broadband orientation stimuli.

As mentioned above, we included SF response modulation in our earlier Fig. 3 to demonstrate that there is no orientation dependence for completeness. However, to avoid potential confusion about the importance of this result and the actual main motivation for the model, we have moved the SF modulation and recruitment results to our new Supplementary Fig. S5.

We assume that the preferred spatial frequencies of the neurons were not measured. However, if the data exist it would be very interesting to have plots and model predictions as in plots b and c but clustering neurons according to their preferred spatial frequency.

As explained above, we could not measure the spatial frequency tuning in our earlier experiments due to the overall length of the recording session becoming too long. Instead, we focused on the role of orientation tuning for responses to broadband orientation stimuli. However, we agree with the reviewer that exploring the role of spatial frequency tuning on SF response modulation would be an interesting topic for a future follow-up study.

Figure 4: Again, to evaluate the suitability of the Gabor model, it would be very informative to include the model predictions for the data shown in plots d-f. Specifically, we would be interested in seeing whether the model predicts the population responses for the small stimuli (in e) showing reduced responses to SF and mixed stimuli.

The key prediction of the model that is also reflected in the center responses in Figure 4e is that surround suppression primarily reduces the neural responses to the narrow orientation bandwidth stimulus (Fig. 3b,c). This prediction was qualitatively confirmed in Figure 4d-f, showing that surround suppression is strongest for the narrow stimulus and thus explaining the increase in neural responses to full-field broadband orientation stimuli. This is also seen in panel 4e, where the center responses to the narrow and ORI stimuli are similar in the absence of surround suppression.

As mentioned above, the Gabor-filter-based models were only fit to the narrow and broad orientation stimuli shown in Fig. 3, and we lack additional experimental data to further extend and validate the model accuracy in predicting other response changes across spatial frequencies or the mixed condition. However, we strongly believe that such a major effort is beyond the scope of the study since we explicitly used a simple Gabor model to demonstrate that orientation tuning is insufficient to explain the increased responses to broadband orientation stimuli, and the impact of surround modulation needs to be considered.

Whether such a simple Gabor model approach would be sufficient to obtain an accurate prediction of neural responses for all interactions across orientations, spatial frequencies, and mixed conditions is unclear and was not the focus of our work. To compare the importance of surround modulation, orientation tuning, and response magnitude for explaining broadband response modulation in Fig. 4, we instead followed the suggestion by reviewer 4 and used the LME model to provide quantitative evidence that surround modulation is a key predictor for the experimental data (Fig. 4g). We strongly believe that these findings are a major advance that provide a basis for future modeling work to further explore the interactions between different stimulus features in driving neural responses to more complex broadband stimuli.

Although the illustrations of model parameters in plot g are nice, we didn't find the plot too helpful to understand the LME. It would help to add the formula that was used. Also, the term "response enrichment" is not defined; it should probably read "response modulation".

We have modified the y-axis label to now read "response modulation" and added the LME formula directly to panel g. The motivation for the illustration was to show that the LME uses surround modulation, orientation tuning, and center responses to predict broadband response modulation while controlling for random effects from individual mice. This is now also shown in the formula, $R_{mij} = \beta_0 + \beta_1 Sm_{ij} + \beta_2 Ot_{ij} + \beta_3 Cr_{ij} + u_j + \epsilon_{ij}$ where R_{mij} is the response modulation for the i th observation within the j th animal, β_0 is the fixed intercept (the total mean response modulation across all animals and conditions), β_1 , β_2 and β_3 are the fixed effect coefficients for the predictors: Sm (Surround modulation), Ot (Orientation tuning) and Cr (Center response). u_j is the random intercept for the j th animal, accounting for the variation between different animals, and ϵ_{ij} is the residual error term for the i th observation within the j th animal, accounting for the variability that is not explained by the fixed effects or the random intercepts. We have also added this information to our description of the LME in the Methods section.

Effect of surround modulation: The authors state that they matched the small stimulus (without surround) to the RFs of the neurons. As surround modulation is highly dependent on the placement of the stimulus on the RF center, it is crucial that we can see the RFs of the neurons in relation to the stimulus. Please, state the size of the center stimulus in the main text and compare it to the average size of the neural RFs.

To better illustrate our choice of the center stimulus position based on the RF mapping results, we added a panel to our new Supplementary Fig. S6c. Here, we show the average receptive fields of the recorded neurons in an example session, superimposed with the chosen location of the center stimulus. We used sparse noise RF mapping for the population of recorded neurons in our imaging field of view and placed the center stimulus at the position preferred by over 60% of the neurons. The average RF size across all imaging sessions was $30.45^\circ \pm 2.04^\circ$, and the center stimulus was 15° in size. The center stimulus size was chosen to match the previously reported¹ maximally stimulating stimulus size for V1 pyramidal neurons in L2/3. The center stimulus size was also shown in Figure 4b and mentioned in the corresponding figure legend. To better address this important point, we have added more details about the center stimulus in the main text and now discuss its relation to the average RF size of V1 neurons in the literature and our two-photon data (lines 371-375, 421-422, 1086-1089).

A related point is that eye movements could easily move the neural RFs away from the small stimulus. Can the authors exclude this possibility?

To address this important point, we further analyzed video data of the animals' eye that was acquired during two-photon imaging and extracted the average position of the pupil using DeepLabCut. We then analyzed the pupil position during RF mapping and the full field and center stimulus presentation. The results are shown in our new Supplementary Fig. S6. As shown in Figures S6a and b, we only observed very rare and small pupil movements without any notable difference in pupil position across stimulus presentation conditions. These results are also in agreement with earlier work^{2,3}, showing only minimal pupil movements during passive stimulation in head-fixed mice. Lastly, significant responses to at least one of the center stimuli were an additional criterion for selecting neurons as part of the population of "center responding cells".

Currently, it is not clear what the widefield data in Fig 5 a-c add to the manuscript. Widefield imaging was introduced to study HVAs, but no quantitative data (like for V1 in Fig 5c) are presented. The authors state that (based on Fig 5b) broadband stimuli decrease responses in HVAs, which seems surprising but is not further discussed.

As mentioned above, our main motivation for the additional widefield experiments was to demonstrate that V1 is the cortical area with the strongest broadband response modulation. It has been shown before that different HVAs can be tuned to higher spatial frequencies or motion speed or preferably respond to textures over scrambled images. We, therefore, wanted to explore if HVAs outside of V1 show similar or even stronger neural responses to broadband visual stimuli.

Our widefield results show this is not the case, and V1 is a key region to study broadband response modulation. To clarify this important point, we now also show quantification of the response differences between broad- and narrow-bandwidth orientation stimuli across V1 and six different HVAs in our new Fig. 5c, demonstrating that broadband response modulation is strongest in V1. These increased responses in V1 were specific to ORI and mixed and did not occur with SF stimuli (Fig. 5d), thus matching our two-photon imaging results in Figures 1,2 and 4. The widefield results, therefore, underline that the observed increase in neural responses to broadband orientation stimuli is indeed most pronounced in V1, and we did not miss out on larger or different effects in other HVAs. We now better describe this point and our new quantification in the main text (lines 492-496) and have also extended the text in the discussion accordingly (lines 737-742).

In addition, it is surprising that the V1 activity ipsilateral to visual stimulation is comparable in size to the contralateral activity although the stimulus was only presented to one visual hemifield. On the other hand, broadband modulation is much smaller on the ipsilateral side. Can the authors explain/discuss this?

We agree that the comparable size of the ipsilateral V1 response is a bit surprising. Most of the activity was concentrated in the lateral binocular region, which also receives information from the ipsilateral eye and could, therefore, result from visual stimulation. However, it might also be that some of the ipsilateral responses were caused by diffuse reflections within the setup from the large contralateral screen. This would also explain the lack of broadband modulation on the ipsilateral side because diffuse reflections are unlikely to convey such stimulus-specific information. We now mention this possibility in the Results section and emphasize that our main conclusions are based on visual responses in the contralateral hemisphere (lines 469 - 473).

The widefield and the LFP data (Fig 5c-e) show that responses to the narrow stimuli are smallest compared to other stimuli, which is different from the single-neuron data in Fig 4d. Furthermore, mixed stimuli evoke large activity in the widefield data (V1), but relatively low activity in the LFP data in superficial layers of V1, while for single-neuron data mixed stimuli evoked responses comparable to narrow and SF data. The single-neuron spike data then is very similar to the single-neuron 2P data, which is a great confirmation. However, we think that the above-mentioned differences across datasets and recording modalities should be discussed.

We agree with the reviewer that including a discussion on the differences between different recording modalities would further enhance the quality of the manuscript. Since the LFP and widefield are population signals that do not just represent the somatic spiking of superficial neurons but also subthreshold potentials as well as dendritic and axonal signaling within a local

area, the response differences between narrow and SF stimuli in the LFP and widefield are likely explained by differences in sub-threshold signals to broadband stimuli that are insufficient to drive spiking in superficial neurons. Similarly, the stronger responses to mixed stimuli in our widefield recordings compared to the superficial LFP are likely due to the stronger contribution of dendritic signals from layer five neurons, which are also reflected in the widefield signal. Earlier work⁴ has shown that widefield signals represent a mixture of superficial and deep cortical activity and are particularly correlated to the activity of deep cortical neurons (see their Fig. S4). Since we found that these deeper neurons respond more strongly to mixed stimuli, this could explain the corresponding increase in the widefield responses. We have extended the discussion in the revised manuscript to now address these differences between the different recording modalities (lines 742-752)

Spike sorting: please add the criteria that were used to sort the units and evaluate commonly used quality metrics of the chosen units (if these were not used for sorting).

The main criteria for the manual curation of the clusters from Kilosort 2.5 were well-separated and consistent spike waveforms across at least three recording channels, stable spike amplitudes across the recording, and a visible gap around the center of the auto-correlogram. To further enforce low refractory period violations in the selected clusters, we also computed the contamination rate using this method⁵ and used an additional cutoff of 0.5 to remove contaminated clusters (following the guidelines on ISI violations from the Allen Brain Institute⁶). We now describe the manual sorting procedure in more detail in the Methods section and also provide an overview of commonly used quality metrics for the selected units in our new Supplementary Fig. S11e. The overview shows consistent results with well-isolated units, such as a high presence ratio, d-prime, and nearest-neighbor hit rate, with a low amplitude cutoff.

Regarding the spiking data from the SC, more information needs to be added. Specifically, where are the RFs of these neurons? How was the border between the cortex and the SC determined, and in which layers were the neurons recorded? Are there differences across the depth of the SC as observed in V1?

Due to time limitations during the Neuropixels recordings, we could not map the RFs of the SC neurons in these experiments. We generally targeted the center of V1 (at 4 mm posterior and 2.5 mm lateral from bregma) and the underlying SC but did not attempt to record from neurons with a similar RF in both regions. We now mention this limitation in the discussion (lines 749-751). However, since the narrow and broadband stimuli were presented as full-field stimuli, we do not think that this represents a major confound for our general conclusions.

As suggested by the reviewer, we now moved the results from the SC recordings to our new Supplementary Fig. S11 to not distract from the main V1 results of the study. In Fig. S11, we show an example brain slice to better demonstrate the probe placement in V1 and SC (Fig. S11a) and also include LFP responses to narrow and broadband stimuli across the different SC layers (Fig. S11b). Consistent with the spiking responses, mixed stimulation induced the largest visual responses in the superficial optical SC layers (Fig. S11c). However, in contrast to V1, there were no general changes in the response preference to the different broadband stimuli across the depth of the SC. The border between cortex and SC was determined by a clear gap in visual responses after roughly 1 mm of recording depth, and we used the depth of the first channel with visual responses after this gap to determine the surface of the optical SC. In agreement with our histological results, this was usually at a depth of 1.63 ± 0.16 mm, and most visually responsive neurons were recorded within the top 0.3 mm of the SC. We have added this information to the Methods section (lines 944-947) and the caption of Supplementary Fig. S11.

Minor comments

• *Figure 1: The use of color in panel h is misleading. In all other plots, color reflects the stimulus feature while in h color reflects neural tuning.*

We thank the reviewer for pointing out this potentially confusing point and have changed the colors in panel h of Figures 1 and 2 to use a different color schema as in the preceding panels.

• *L. 255-57: Again, difficult to read. Not clear how this statement follows from the previous sentence. What is "diverse response modulation" (up and down modulation?) and where do we see that?*

The statement 'diverse response modulation' indeed refers to both the up and down modulation of individual neurons in response to the broadband SF stimulus. This is seen in Fig. S1h, showing that ~20% of the neurons were either narrow- or broad-range selective, respectively. So, the lack of increased responses to broadband SF stimuli is not due to a general lack of spatial frequency bandwidth specificity (with neurons responding equally to all stimuli) but rather an equal amount of up and down modulation when increasing SF bandwidth. We have improved the writing in this section to clarify this point.

• L. 569: *delta AUC was not defined. Should it read absolute AUC?*

We thank the reviewer for pointing this out. Delta AUC should indeed read absolute AUC, as it is also defined as AUC_{abs} in the Methods section. We have corrected this mistake and now consistently refer to this metric as AUC_{abs} throughout the manuscript. Moreover, we further reference our new Supplementary Fig. S13, where we illustrate how the AUC and AUC_{abs} are computed.

• L. 694-705: *The argument could be clarified by stating the assumption: if neurons were broadly tuned to spatial frequency, they would be expected to respond to all spatial frequency bandwidth stimuli. The data show that they are not tuned broadly relative to the tested spatial frequencies. We thank the reviewers for this valuable suggestion and have added this assumption to the paragraph.*

• *Figure S8b: an explanation/cartoon of how the curves are constructed from the data in a would help readers unfamiliar with ROCs; our impression is that the green curve is not consistent with the data in a (right side) maybe because of the smoothing by the violin plot.*

This issue was indeed due to the smoothing of the violin plot and the ROC curve. To improve the clarity of this figure (now Supplementary Fig. S13), we now show explicit response histograms and unsmoothed ROC curves for each example cell and better illustrate how the ROC results from the response differences for two separate stimuli. We now show markers in the response distributions and corresponding ROC curves that indicate possible positions of a discrimination threshold to separate the two stimulus distributions in light and dark orange. As the threshold is placed at different positions between the two response distributions, the true positive and false positive rates change accordingly, giving rise to the ROC curve. The AUC is also shown as the area under the ROC curve to provide an intuitive understanding of how high AUC values relate to improved stimulus discriminability. Lastly, we also added a detailed description of this process in the figure caption of Supplementary Fig. S13.

Figure S9a: Correlation measure is not clear. Is each correlation for a single neuron? But in response to how many different stimuli and how many trials? This correlation seems to measure how similar the response of a neuron is to different stimuli, not the overlap of populations responding to different stimuli. We agree that the correlation in the earlier version was not clear enough and have improved the figure (now Supplementary Fig. S12) and the caption accordingly. We performed this analysis in response to Reviewer #3's suggestion to compute the correlation for neural responses across bandwidth conditions between the two central features to test for bandwidth selectivity that is independent of the central feature.

Each correlation was indeed for a single neuron, and we tested all neurons that responded to at least one stimulus condition. For each neuron, we concatenated the average responses to all stimulus presentations with different feature bandwidths (e.g., orientation bandwidth) and the same central feature (e.g., low SF) condition together (19 trials per bandwidth, 57 responses in total) and then computed the correlation of this response vector to the stimulus responses with the same feature bandwidths but a different central feature. Our logic here was that neurons that selectively respond to a specific stimulus bandwidth, regardless of the central stimulus feature, should have low responses to other bandwidth stimuli but strongly respond to stimuli with that bandwidth in both central feature conditions. To illustrate this, we now show an example cell in the top part of our new Supplementary Fig. S12 that showed a high correlation and preferentially responded to broadband orientation stimuli with a central SF of either 0.04 or 0.16 cpd. However, across all neurons, the correlations were very small for either different orientation bandwidth or SF bandwidth stimuli (Fig. S12a,c). Therefore, there appears to be no large systematic response preference to stimuli with a specific feature bandwidth that is independent of the central stimulus feature across all neurons.

To further make this point, we also reported that the overlap between neural populations that responded to the same bandwidth stimuli with different central features is fairly low. However, although this result was reported in the same figure caption, it is not based on the correlation analysis but from using the same selection criteria as for Supplementary Fig. S1 d,h. We have kept this statement in the caption of Fig. S12 for completeness but now explicitly state that this is an additional result to extend the interpretation of the correlation analysis.

L. 156: "(a)" is missing

L. 252-255: Sentence is difficult to understand. Consider shortening and separating comparison between spatial frequency and orientation results from narrow and broad-range selectivity. As there are many parameters/features discussed, please specify each time which features are compared. For example, "... more common for broadband spatial frequency stimuli compared to broadband orientation stimuli".

- L. 500: better terms for "excitatory or inhibitory responses" may be "enhanced or suppressed responses"

- L. 523-25: Description could be clarified in this sentence stating that spatial frequencies will be discriminated while orientation bandwidth is increased, and vice versa.

- L. 603ff.: Please add to the main text that stimuli of different bandwidths were grouped into blocks, not intermixed within a block.

- L. 1233: Fig S4g, h don't exist.

We thank the reviewer for spotting these mistakes. All these points have now been corrected in the new version of the manuscript.

Reviewer #2

First, I would like to thank the authors for their effort in addressing mine and other reviewers' comments. The new manuscript has been substantially modified - just the rebuttal letter is an impressive 30 pages.

I believe that the manuscript has much improved. Comparisons to the baseline Gabor filter model show which effects are expected by default and which reveal more complex properties of V1 populations. Below I list suggestions that I believe should be incorporated prior to publication. I leave the final assessment of the fit of this work to Nature Communications to the editors. I can say, however, that I find this manuscript a thorough, rigorous and systematic contribution to our understanding of population coding in the rodent visual cortex.

We thank the reviewer for their insightful feedback and positive evaluation of our work.

Minor comments:

1. Please add the equations describing the surround suppression model. I can't find it in the main text. The mechanics of the model should be clearly explained and motivated. In particular please explain why did you choose subtractive (?) and not divisive normalization.

We agree with the reviewer that it is important to clearly explain the mechanics and motivation for the Gabor model approach. As described in our response to main point #1 of reviewer 1, the main motivation for the model was to explore whether the increased responses of V1 neurons to broadband orientation stimuli can be simply explained by the additional recruitment of neurons with different orientation tuning. This is now clarified in the main text, and we have strongly improved the explanation of the models, including the equations for both the tuning only and the surround suppression model in the Methods section and Supplementary Fig. S3 and S4. We believe that our improved descriptions and illustrations have strongly improved the quality of the manuscript and made it much clearer for the reader to understand the model motivation and implementation.

Given the existing literature on this topic, we also thank the reviewer for pointing out that divisive normalization would be more logical to use for the surround suppression model. When implementing the initial model, we thought of the surround response as an inhibitory signal that is subtracted from center responses to normalize neural responses to simple grating stimuli.

However, following the reviewer's suggestion, we modified our model approach to employ divisive instead of subtractive normalization and found that this approach is even better suited to fit our experimental data. As illustrated in our new Supplementary Fig. S4, we therefore implemented a divisive surround suppression mechanism in the model, which is better in line with previous work and further improves the model performance.

2. Parameters of the model were fitted to data - while this does not affect the interpretation, it should be perhaps explained whether it is a prediction or a fit.

We agree that it is important to clarify which parameters of the model were fitted to the data and which results are a prediction or a fit. We have, therefore, improved our description of the model in the Results and Methods sections and included a new illustration of the tuning-only and surround suppression model in Supplementary Fig. S4. The new illustration makes it much clearer that the model parameters were fitted directly to the neural responses to the narrow and broad orientation bandwidth stimuli. We now also show the resulting fits of the model directly in panels b and c of our new main Figure 3.

As the response modulation is a relative index, it is independent of the response scaling in the tuning-only model. Therefore, the deviations from the response modulation are direct predictions from the Gabor filters. The center Gabors predicted a strong reduction for neurons with 0° tuning in response to broadband stimulation, which did not agree with the experimental data. In contrast, the surround suppression model obtained a much better fit for neural response with narrow orientation bandwidth and accurately predicted the experimentally observed broadband response modulation. The main qualitative prediction from the model is, therefore, that surround suppression should be strongest for neural responses to narrow orientation bandwidth, while broadband stimuli should reduce such surround suppression. This is now explicitly stated in the Results section (see lines 355-358) and formed the central hypothesis for our follow-experiments in Fig. 4

Typos:

l. 1176 - envelop -> envelope

Thank you, this has been corrected.

Reviewer #3

The authors have significantly improved on the original version of the manuscript by adding new data, new analyses, and a model to substantiate their findings. They have improved the writing and methods and completely clarified my previous questions.

We thank the reviewer for all their effort and for confirming that our revisions have significantly improved the value of our work.

The only remaining very minor issue for me is in the results for Figures 1 and 2, they report the results of the Wilcoxon signed-rank tests. They test the mid vs. the narrow bandwidth, but it is unclear what the test is for the broad bandwidth (against mid or narrow). This is the case for both orientation and SF bandwidths, as well as for amplitude and percentage of responsive cells. Perhaps a Kruskal-Wallis ANOVA would be more appropriate here.

We agree with the reviewer that this was not fully clear in the earlier version of the manuscript. In all cases for Figures 1 and 2, we tested both mid- and broad-bandwidth responses against the narrow-bandwidth responses. We reasoned that responses to narrow stimuli provide a baseline against which we could compare responses to mid- or broad-bandwidth stimuli. We have made this clearer in the new version of the Results section and now consistently specify that the tests were performed against the narrow bandwidth condition.

Reviewer #4

The authors have done a great job responding to both my concerns and those of the other reviewers. The addition of copious new data sets and additional modifications to the statistical approaches greatly strengthened the work. I am satisfied with their manuscript at this stage.

We thank the reviewer for their effort in improving the manuscript and confirming that our new additions to the work have greatly strengthened the scientific value of the manuscript.

References

1. Keller, A. J., Roth, M. M. & Scanziani, M. Feedback generates a second receptive field in neurons of the visual cortex. *Nature* 582, 545–549 (2020)
2. Sakatani T, Isa T. Quantitative analysis of spontaneous saccade-like rapid eye movements in C57BL/6 mice. *Neurosci Res.* 2007 Jul;58(3):324-31.
3. Meyer A.F., Keefe, O.J., Poort J., Two Distinct Types of Eye-Head Coupling in Freely Moving Mice. *Current Biology* Volume 30, Issue 11, 2116 - 2130.e6.
4. Peters, A.J., Fabre, J.M.J., Steinmetz, N.A. *et al.* Striatal activity topographically reflects cortical activity. *Nature* 591, 420–425 (2021).
5. Hill, N. D., Mehta B. S., Kleinfeld D., Quality Metrics to Accompany Spike Sorting of Extracellular Signals *Journal of Neuroscience* 15 June 2011, 31 (24) 8699-8705;
6. https://allensdk.readthedocs.io/en/latest/static/examples/nb/ecephys_quality_metrics.html#ISI-violations

Broadband visual stimuli improve neuronal representation and sensory perception

Elisabeta Balla, Gerion Nabbefeld, Christopher Wiesbrock, Jenice Linde, Severin Graff,
Simon Musall, Björn M. Kampa

Response to the reviewers' comments

III. Third round of reviews:

Reviewer

#1

(Minor final points):

1. L. 336: "the simple tuning-only model overestimated the increase in response amplitude by neurons with orientation tuning that was far away from the central 0° orientation". Wouldn't that imply that the yellow curve in Fig 3d is above the neural response median for neurons tuned to -45 and 45 deg? But it is below.

We thank the reviewers for spotting this error. While we initially found both effects described, the final model presented in our manuscript no longer predicts this increase in response amplitudes for neuron with +45° orientation tuning. We have corrected this statement accordingly:

"In contrast, the simple tuning-only model predicted a stronger decrease in neural response amplitude for neurons with 0° tuning that was not seen in the measured responses (Fig. 3d, yellow line $R^2_{\text{tuning only}} = -1.67$)."

2. Equation in l. 1280: It wasn't obvious how the sum of sums equal the fraction (0.01622/0.1627). The fraction is well explained in the text but we couldn't relate it to the sum of sums (maybe it should be a fraction of sums?). Also, the definition of a(o,b) is missing?

Response: We thank the reviewers for their suggestion. We did in fact compute this factor as a fraction of sums. However, we tried to simplify this equation to improve readability, only now realizing that this was not an equivalent rearrangement of the equation. In addition, we had previously included a sum over spatial frequency tuning (F) here. However, we realize now that this sum was already included in the definition of c(o, b). We removed this obsolete summation and express the equation as the fraction of the individual sums now, the way we computed it:

$$w_t = \frac{1}{n_o n_B} \times \frac{\sum_{b \in B} (\sum_{o \in O} a(o, b))}{\sum_{b \in B} (\sum_{o \in O} c(o, b))} = \frac{0.01622 \Delta F / F}{0.1627 AU} = 0.0997$$

Further, we did introduced a in Line 1282. However, we understand that this might have been unclear and we declare $a(o, b)$ more explicitly now.

"Here, w_t represents the scaling factor from Gabor response magnitudes to match the measured neural responses to different stimulus conditions, based on their orientation tuning ($a(o, b)$; analogous to $c(o, b)$) in units of $\Delta F / F$ (Fig. 3b, c)."